# THE CENTRAL SPANNING TREE PROBLEM

## ABSTRACT

Spanning trees are an important primitive in many data analysis tasks, when a data set needs to be summarized in terms of its "skeleton", or when a tree-shaped graph over all observations is required for downstream processing. Popular definitions of spanning trees include the minimum spanning tree and the optimum distance spanning tree, a.k.a. the minimum routing cost tree. When searching for the shortest spanning tree but admitting additional branching points, even shorter spanning trees can be realized: Steiner trees. Unfortunately, both minimum spanning and Steiner trees are not robust with respect to noise in the observations; that is, small perturbations of the original data set often lead to drastic changes in the associated spanning trees. In response, we make two contributions when the data lies in a Euclidean space: on the theoretical side, we introduce a new optimization problem, the "(branched) central spanning tree", which subsumes all previously mentioned definitions as special cases. On the practical side, we show empirically that the (branched) central spanning tree is more robust to noise in the data, and as such is better suited to summarize a data set in terms of its skeleton. We also propose a heuristic to address the NP-hard optimization problem, and illustrate its use on single cell RNA expression data from biology and 3D point clouds of plants.

## 1 INTRODUCTION

Many data analysis tasks call for the summary of a data set in terms of a spanning tree, or use tree representations for downstream processing. Examples include the inference of trajectories in developmental biology (Saelens et al., 2019; Chizat et al., 2022), generative modeling in chemistry (Ahn et al., 2022), network design (Wong, 1980) or skeletonization in image analysis (Bai et al., 2023; Wang et al., 2019). The problem is akin to, but more complicated than, the estimation of principal curves because good recent methods such as (Lyu et al., 2019) cannot account for branched topologies. For a spanning tree representation to be meaningful, it is of paramount importance that the tree structure be robust to minor perturbations of the data, e.g. by measurement noise. In this work, we address the geometric stability of spanning trees over points lying in an Euclidean space.

The minimum spanning tree (mST) is surely the most popular spanning tree, owing to its conceptual simplicity and ease of computation. For a graph $G = (V, E)$ with edge costs, the mST is a tree that spans $G$ while minimizing the total sum of edge costs. It prioritizes shorter edges that connect closely located nodes enhancing data faithfulness. Unfortunately, its greedy nature makes the mST susceptible to small data perturbations that may lead to drastic changes in its structure, see Figure 2. An alternative, the minimum routing cost tree (MRCT), minimizes the sum of pairwise shortest path distances (Masone et al., 2019). Unlike the mST, solving the MRCT is NP-hard (Wu and Chao, 2004). Despite this, the MRCT exhibits a more stable geometric structure compared to the mST, as it tends to be more "star-shaped" (see Figure 2). Nevertheless, this star-shaped tendency inclines towards connecting nodes that are spatially distant, introducing a risk of information loss and compromised data fidelity. This effect becomes particularly pronounced in high-dimensional spaces, potentially rendering the MRCT approach unusable (see Figure 12, Appendix B). Achieving a balance between data fidelity and geometric robustness is crucial for an effective spanning tree.

**Central spanning trees (CST)** In this paper, we propose a novel parameterized family of spanning trees that interpolate and generalize all the aforementioned ones. Unless otherwise stated, we will assume a complete Euclidean graph, where the nodes are embedded in the Euclidean space with coordinates $X_V = \{x_1, \ldots, x_{|V|}\} \subset \mathbb{R}^n$ and the edge costs are given by the distances between the

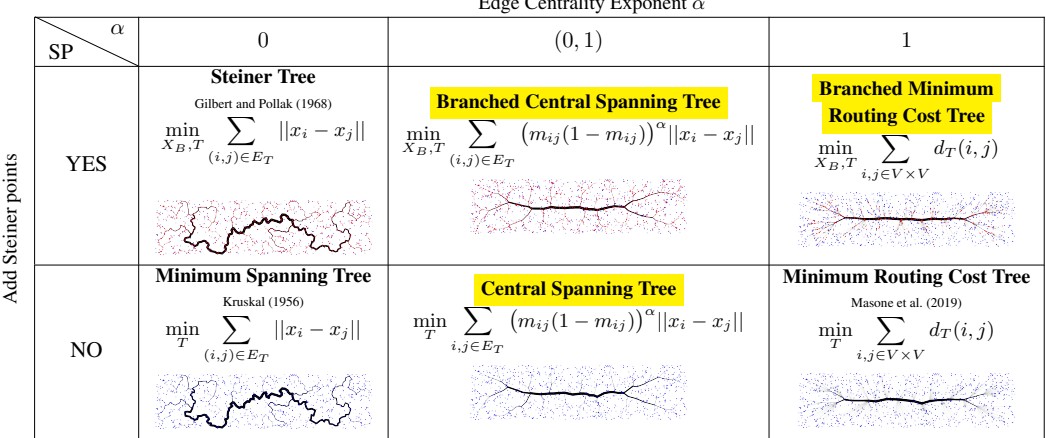

Figure 1: **Family of Euclidean central spanning tree problems with and without Steiner points**. The central spanning tree weighs the costs of the edges, given by node distances, with the centrality of the edges, $m_{ij}(1 - m_{ij})$. The influence of the centrality is regulated by the parameter $\alpha \in [0, 1]$. For lower $\alpha$ values the centrality becomes insignificant, and the tree tends to contain short edges. For higher $\alpha$ values, the tree encourages central edges of low cost, at the expense of peripheral edges of higher cost. We study central spanning trees with and without additional Steiner points (shown in red). The widths of all edges are proportional to their centrality. The central spanning tree problems includes well-known and novel spanning tree problems, the latter highlighted in yellow.

vertices, $c_{ij} = ||x_i - x_j||$. We define the CST as the spanning tree of $G$ that minimizes the objective

$$\text{CST} \coloneqq \arg\min_{T \in S} \sum_{e \in E_T} \left(m_e(1 - m_e)\right)^{\alpha} c_e = \arg\min_{T \in S} \sum_{e=(i,j)\in E_T} \left(m_{ij}(1 - m_{ij})\right)^{\alpha} ||x_i - x_j|| \quad (1)$$

where $m_e$ and $(1 - m_e)$ are the normalized cardinalities of the components resulting from the removal of the edge $e$ from $T$. Thus, $m_e(1 - m_e)$ is the product of the number of nodes on both sides of the edge divided by $|V|^2$. Because this value is proportional to the "edge betweeness centrality"[1] of $e$ in $T$ (Brandes, 2008), we call the problem the Central Spanning Tree (CST) problem. The exponent $\alpha$ is the interpolating parameter that modulates the effect of the edge centrality. For $\alpha$ close to 0, the centrality weight becomes insignificant, so the tree tends to contain lower cost edges overall. For $\alpha = 0$ we retrieve the mST. On the other hand, for higher values of $\alpha$, the centrality becomes more relevant and the tree is more inclined to minimize the cost of the most central edges of the tree. For $\alpha = 1$, the resulting expression is proportional to the MRCT since each edge cost is multiplied by the number of shortest paths it belongs to. In that case, the total sum retrieves the sum of shortest path distances (see Appendix D). Although we here focus on the $\alpha$-range $[0, 1]$, it is worth mentioning the limiting cases when $\alpha$ tends to $\pm\infty$. At one extreme, when $\alpha \to \infty$, the CST tends to a star graph centered on the medoid of the graph, i.e. the node that minimizes the distance to the rest of nodes. At the other extreme, when $\alpha \to -\infty$, the CST tends towards the path graph that minimizes the CST objective. We refer to Appendix E for further details.

As will be seen in Section 2, the $\alpha$ parameter has an effect on the geometric stability of the spanning tree, with higher $\alpha$ resulting in greater robustness. The robustness of spanning trees has been explored in various ways, such as investigating mST cost robustness under edge weight uncertainty (Kasperski and Zieliński, 2011) and studying robustness against node or edge failure in networks (Liu et al., 2019). The central tree problem (Bezrukov et al., 1996), related by name to ours, focuses on computing a tree that minimizes the maximal distance to a set of given trees. To our knowledge, we are the first to delve into the robustness of the geometric structure of a spanning tree in response to data perturbations like noise.

Finally, note that the Minimum Concave Cost Network Flow (MCCNF) problem (Zangwill, 1968; Guisewite and Pardalos, 1990), which aims to minimize the cost of distributing a certain commodity

---

[1]The edge betweenness centrality measures an edge's frequency in shortest paths between nodes, with more traversed edges being deemed more central. In trees, it's the product of nodes on opposite sides of the edge.

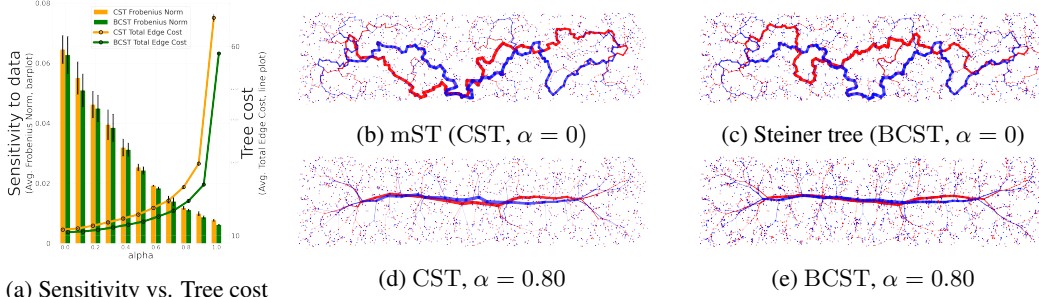

(a) Sensitivity vs. Tree cost

(b) mST (CST, $\alpha = 0$)

(c) Steiner tree (BCST, $\alpha = 0$)

(d) CST, $\alpha = 0.80$

(e) BCST, $\alpha = 0.80$

Figure 2: (B)CST for $\alpha > 0$ are more robust to noise and adhere to large scale structure in the data better than the mST and Steiner tree. Left: When increasing $\alpha$, the sensitivity to random density fluctuations in the data decreases (good). At the same time, the total length of the tree increases (bad). This tradeoff can be adjusted with a single hyperparameter $\alpha$. More details in Section 2.1. Right: CST and BCST of two samples, red and blue, drawn from the same distribution. The tree backbone reflects the global structure more accurately for $\alpha > 0$ than for $\alpha = 0$. Edge widths are proportional to their centrality. All trees except for the mST were computed using the heuristic proposed in Section 4.2. See Appendix A for more examples.

from source to sink nodes, is also connected to the CST problem. Such a problem models the cost of an edge as a concave function of the flow. In our case, the term $m_e$ in equation 1 can be interpreted as a flow, and the function $(m_e(1 - m_e))^{\alpha} c_e$ is concave with respect to it for $\alpha \in ]0, 1]$. A more detailed discussion of the interpretation of the CST as a MCCNF problem is offered in Appendix C.

**Branched central spanning trees (BCST)** Inspired by (Lippmann et al., 2022), we also study the variant of the CST problem which allows for the introduction of additional nodes, known in the literature as branching points or Steiner points (SPs). Formally, we distinguish between two types of nodes. On the one hand, we have the terminal nodes with fixed coordinates given by $X_V$. On the other hand, we allow for an extra set of points, $B$, whose coordinates $X_B$ must be jointly optimized with the topology $T$. In this case, $T$ is a spanning tree defined over the nodes $V \cup B$. Accordingly, the objective of the CST problem becomes

$$\min_{T \in S_{V \cup B}, X_B} \sum_{(i,j) \in E_T} \left( m_{ij}(1 - m_{ij}) \right)^{\alpha} ||x_i - x_j|| \qquad (2)$$

In this generalization, which we refer to as the branched CST (BCST), the well-known Steiner tree problem (Hwang and Richards, 1992; Warme et al., 2000) arises when $\alpha = 0$. Figure 1 summarizes (B)CST and its limiting cases, only some of which have been studied in the literature so far.

**Contributions.** 1) We present the novel (B)CST problem and provide empirical evidence for its greater stability on toy and real-world data; 2) We propose an iterative heuristic to estimate the optimum of the BCST problem. By exploiting the connection between the branched and unbranched versions of the CST problem, we are able to use the heuristic defined for the BCST to also approximate the Euclidean CST without Steiner points. 3) We show analytically that if the terminal points lie on a plane then for $\alpha \in [0, 0.5] \cup \{1\}$ the Steiner points of the optimal solution can have up to degree 3, and we provide evidence that this holds also for $\alpha \in ]0.5, 1[$.

## 2  STABILITY OF THE CST PROBLEM

### 2.1  TOY DATA

We explore the robustness of the (B)CST problem against data perturbations by comparing the (B)CST topologies as small noise is introduced. In a toy example, three sets of 1000 points, uniformly sampled from a rectangle, are perturbed with Gaussian noise, yielding five perturbed sets per original set. We then compute the CST and BCST for $\alpha \in \{0, 0.1, \ldots, 0.9, 1\}$, deeming a tree robust if minor data perturbations lead to minor structural changes in the tree. Formally, we consider a method $\delta$-robust, if, for any sets of points $P_1$ and $P_2$ and their respective tree $T_1$ and $T_2$, $d_T(T_1, T_2) \leq \delta d_P(P_1, P_2)$, where $d_T$ and $d_P$ measure tree and set distances, respectively. $d_P$

quantifies the perturbations we aim to withstand, where lower $d_P$ values correspond to sets that are similar based on specific criteria. In our example, we define $d_P$ as the average distance between points and their perturbed counterparts. Since we apply the same noise to each point, the average distance between points approximates the Gaussian noise's standard deviation, making it nearly constant. To quantify structural tree changes, we set $d_T$ equal to the Frobenius norm of the shortest path distance matrices between the original and perturbed (B)CST trees.

Figure 2a shows the average Frobenius norm between the original and corresponding perturbed samples across various $\alpha$ values. It is evident that as $\alpha$ increases, there is a noticeable decrease in the Frobenius norm. Since our $d_P$ is fairly constant, showcasing that the Frobenius norm decreases implies a reduction in $\delta$ as $\alpha$ rises, i.e. the trees become more robust. However, we also plot the average cost of the trees (sum of the individual edge costs), which increases with $\alpha$. Thus, the improvement in robustness comes at the expense of adding longer edges. This pattern is expected because, as $\alpha$ increases, the (B)CST tends to a medoid-centered star graph (see Appendix E). This graph will have long edges but will also exhibit robustness to noise due to the medoid's inherent stability. According to our definition of $\delta$-robustness, the $\alpha \to \infty$ (B)CST limiting case, which always outputs a star-graph, will be deemed robust despite its undesirability for describing the data structure. We associate the data structure with the graph node interconnectivity, wherein shorter edges preserve it better. Thus, $\alpha$ serves as a parameter trading off stability vs. data fidelity. Indeed, the mST and Steiner tree ($\alpha = 0$) on the right side of Figure 2 are highly sensitive to minor data changes due to their greedy nature, prioritizing shorter edges. Conversely, the (B)CST solutions at $\alpha = 0.8$ are more stable, faithfully representing the data's overall layout, albeit with longer edges.

## 2.2 REAL-WORLD DATA

The ability to summarize the structure of data without excessive sensitivity to random jitter is important in many applications. In this section we briefly showcase some potential applications where the (B)CST can be beneficial (see Appendix B for more details):

**Trajectory inference of single cell data:** The high dimensional single cell RNA-sequencing data can be used to model the trajectories defined by the cell differentiation process. We show results on mouse bone marrow data (Paul et al., 2015), here denoted as Paul dataset. To study robustness we perturb the data by removing half of the samples and then compare how the backbone of different spanning trees is affected by this perturbation. For visualization, the data is embedded in 2D using PAGA (Wolf et al., 2019). If a spanning tree aligns well with the 2D embedding, this is an indication that the tree approximates the trajectory well. Figure 3 shows the embedded trees. The mST misses the highlighted bifurcation and it is more sensitive to the noise. The CST and BCST are robust to the perturbation and align well with the PAGA embedding, though the CST may not reconstruct well the finer details. The addition of SPs enables the BCST to follow the trajectory more closely. The case $\alpha = 1$, resulting in a star graph, is presented in Appendix B.

**3D Plant skeletonization:** The skeletonization of plants is relevant for comprehending plant structure, with applications in subsequent tasks. The skeletons of a 3D point cloud of the surface of a plant (Schunck et al., 2021), obtained using the BCST for various $\alpha$ values, is shown in Figure 4. Intermediate $\alpha$ values are able to represent the plant's structure more accurately than extreme ones, and produce skeletons that are on pair with dedicated handcrafted pipelines (Magistri et al., 2020).

## 3 CORRESPONDENCE BETWEEN THE BCST AND CST TOPOLOGIES

Both the CST and the BCST problems have to be optimized over the set of feasible spanning tree topologies. This optimization is combinatorial in nature and turns out to be NP-hard in both cases (see Remark $C.1$ Appendix C). For the CST case, Cayley's formula tells us that the number of feasible topologies is equal to $N^{N-2}$ (Cayley, 1878), which grows super-exponentially with the number of nodes $N$. For the BCST case, w.l.o.g., we can represent any feasible topology as a full tree topology, i.e. as a tree with $N-2$ Steiner points, each of degree 3, and with leaves corresponding to the terminals. This representation is justified by the fact that any other feasible topology with SPs of degree higher than 3 can be represented by a full tree topology by collapsing two or more SPs, that is, when two or more SPs have the same coordinates. We prove in Appendix I that if the terminal points lie in a plane, then SPs with degree higher than 3 are not realized in optimal

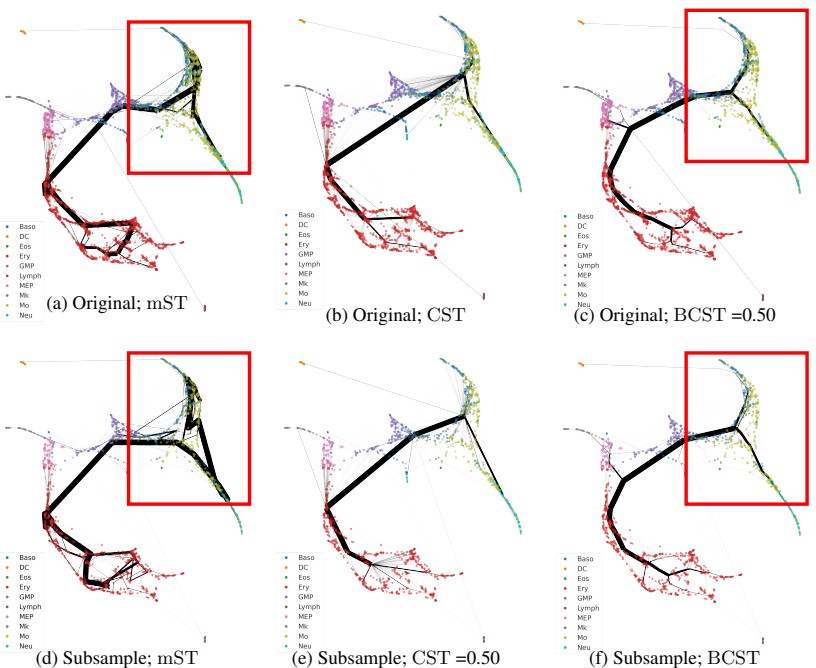

Figure 3: **mST, CST and BCST of the Paul dataset Paul et al. (2015)**. We applied the algorithms to both the original data (top) and a perturbed version with half of the points randomly removed (bottom). PAGA was used for 2D visualization, while the trees were computed in a 50-dimensional PCA projection. Colors represent different cell populations. The width of the edges is proportional to their centralities. In the original data, the mST does not faithfully model the trajectory bifurcation highlighted by the rectangle (3a). Moreover, the trajectory changes drastically at this point once a subset of the samples is removed (3d). The CST and BCST are able to detect the bifurcation, and preserve the main backbone of the tree after the data has been perturbed (3b, 3e, 3c, 3f)

solutions for $\alpha \in [0, 0.5] \cup \{1\}$ unless they collapse with a terminal. For $\alpha \in [0, 0.5]$ this fact follows from the subadditivity of the function $m_e(1 - m_e)^{2\alpha}$. For $\alpha \in ]0.5, 1[$ we provide empirical evidence that the statement also holds in that case. This hypothesis is strengthened by the fact that we can prove it for $\alpha = 1$. For $N$ terminals, the number of possible full topologies is equal to $(2N - 5)!! = (2N - 5) \cdot (2N - 7) \cdots 3 \cdot 1$ (Schröder, 1870), which also scales super-exponentially, but at a lower rate than the number of topologies of the CST. Consequently, an exhaustive search through all trees is not feasible.

The heuristic presented in Section 4 exploits the correspondence between the feasible topologies of the BCST and the CST problems. Given a full tree topology $T_{\text{BCST}}$, we say that a topology $T_{\text{CST}}$ of the CST problem can be derived from $T_{\text{BCST}}$ if: 1) we can collapse the SPs of $T_{\text{BCST}}$ with

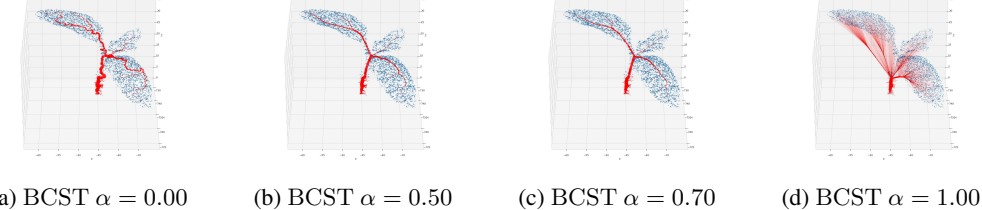

(a) BCST $\alpha = 0.00$     (b) BCST $\alpha = 0.50$     (c) BCST $\alpha = 0.70$     (d) BCST $\alpha = 1.00$

Figure 4: BCST for different $\alpha$ values of a 3D point cloud with 5000 samples capturing the surface of a plant. With $\alpha = 0.00$, the tree branches exhibit greater irregularity, while at $\alpha = 1.00$, finer details are obscured. Intermediate $\alpha$ values offer a more faithful representation of the plant's structure.

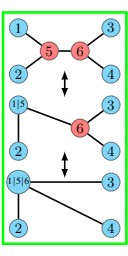 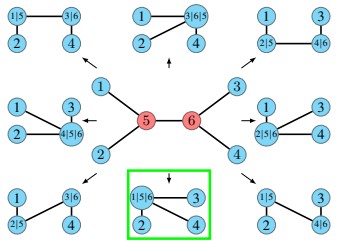 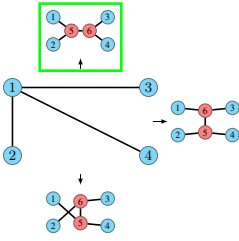

(a) Steps $T_{\mathrm{CST}} \leftrightarrow T_{\mathrm{BCST}}$     (b) Topologies derived from $T_{\mathrm{BCST}}$     (c) Topologies derived from $T_{\mathrm{CST}}$

Figure 5: **Correspondence between the topologies of the CST and BCST problems.** 5a) Mapping from a BCST topology, $T_{\mathrm{BCST}}$, to a CST one, $T_{\mathrm{CST}}$ and vice versa. Terminal nodes and SPs are represented in blue and red, respectively. From top to bottom it is shown how the SPs couple to different terminals. From bottom to top, the SPs are spawned by different pairs of adjacent neighbors. First by the pair 3 and 4 and later by 2 and 6. 5b) $T_{\mathrm{CST}}$ feasible topologies derived from a single $T_{\mathrm{BCST}}$. 5c) $T_{\mathrm{BCST}}$ feasible topologies derived from a single $T_{\mathrm{CST}}$.

the terminals such that the resulting topology is $T_{\mathrm{CST}}$, and 2) for any SP $s$ that is collapsed with terminal $t$, then all SPs along the path connecting $s$ to $t$ must also collapse with $t$. In other words, a SP cannot overtake any other SP in the collapse process. Figure 5a (from top to bottom) shows the steps to transform a topology $T_{\mathrm{BCST}}$ into a topology $T_{\mathrm{CST}}$ by iteratively collapsing the SPs. Analogously, we can derive a topology $T_{\mathrm{BCST}}$ from $T_{\mathrm{CST}}$ by spawning SPs from the terminals in $T_{\mathrm{CST}}$, i.e., introducing SPs connected to the terminals. Since in a full tree topology SPs have degree 3 and terminals have degree 1, we add one SP per each pair of nodes adjacent to a common terminal node, so that the SP is connected to the triple of nodes.

The correspondence mapping between $T_{\mathrm{CST}}$ and $T_{\mathrm{BCST}}$, as shown in Figures 5b and 5c, is not unique. Multiple $T_{\mathrm{CST}}$ can be derived from a single $T_{\mathrm{BCST}}$ and vice versa. Specifically, any $T_{\mathrm{BCST}}$ can generate approximately $O(3^{N-2})$ $T_{\mathrm{CST}}$ topologies because each SP locally has 3 nodes to merge with. Similarly, the number of $T_{\mathrm{BCST}}$ topologies derived from $T_{\mathrm{CST}}$ depends on the degree of the terminal nodes. Higher-degree terminals can generate more topologies as they can spawn more pairs of nodes. For more details on the cardinalities of derivable topologies, see Appendix F.

Despite the mapping ambiguity between the topologies, we can reduce the number of trees to explore in the CST/BCST given a BCST/CST topology. Although the optimum of one problem is not guaranteed to be derived from the optimum of the other (see Appendix G), we show empirically that the heuristic proposed in Section 4 can exploit the positions of the SPs together with the correspondence between the sets of topologies of both problems to produce competitive results.

## 4 CST AND BCST OPTIMIZATION ALGORITHM

This section details the proposed heuristic for optimizing the BCST and CST problems. We will first focus on the BCST. The heuristic iterates over two steps: First, given a fixed topology, the algorithm finds the geometric positions of the Steiner points (SPs) that exactly minimize the cost conditioned on the topology. Given the optimal coordinates of the SPs, we then update the topology of the tree by computing an mST over the terminals and SPs. This procedure is iterated until convergence or until some stopping criterion is met.

### 4.1 GEOMETRY OPTIMIZATION

The BCST problem is similar in nature to "branched" or "ramified" optimal transport (BOT) (Gilbert, 1967; Bernot et al., 2008). The main difference lies in the weight factors that multiply the distances in the objective function equation 2, which in our case are the edge betweeness centralities, and in BOT are flows matching supply to demand. Appendix C.1 elaborates on the commonalities and differences between the BOT and BCST problems.

The BCST problem can be divided into two subproblems: combinatorial optimization of the tree topology and geometric optimization of the coordinates of the SPs, $X_B$. When conditioning on a

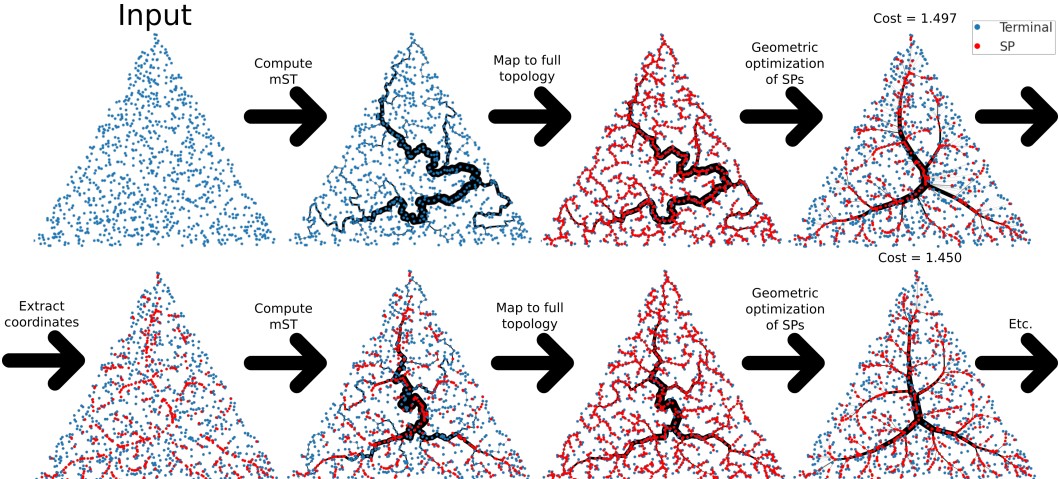

Figure 6: **The mSTreg heuristic iteratively transforms an mST to an approximate BCST.** Given a set of points, the heuristic first computes the mST over all points and transforms it into a full tree topology by adding Steiner points (SPs). Next, the optimal positions of the SPs are computed using iteratively reweighted least squares. Given the updated SPs coordinates, the heuristic recomputes the mST over the union of the terminals nodes and the previous SPs. This mST produces a new topology, which is again transformed into a full tree topology by adding SPs whose coordinates are optimized. This process is repeated until some stopping criterion is satisfied.

topology $T$, the BCST objective equation 2 is a convex problem w.r.t. $X_B$. Despite its convexity, the objective is not everywhere differentiable. We build on the iteratively reweighted least squares (IRLS) approach from Smith (1992); Lippmann et al. (2022) to efficiently find the positions of the SPs. In Appendix J, we show that the algorithm is agnostic to the weighting factors that multiply the distances, and can therefore be applied mutatis mutandis to our problem.

We emphasize that Gilbert (1967) provided an analytical formula for the branching angles around the SPs for the BOT problem. This derivation carries over to the BCST case, as we show in Appendix H. This formula can help to generalize some properties to non-Euclidean manifolds (Lippmann et al., 2022). One of these properties is the infeasibility of degree 4 SPs in the plane, which we prove in Appendix I for $\alpha \in [0, 0.5] \cup \{1\}$.

### 4.2 HEURISTIC OPTIMIZER FOR THE (B)CST PROBLEM

We now present a heuristic which alternates between the SP geometric coordinate optimization (convex) and a topology update (combinatorial). The heuristic's main characteristic is how it exploits the location of the branching points given an initial topology guess. The heuristic's underlying assumption is that the optimum position of the branching points may suggest a more desirable topology.

Unless otherwise stated, the heuristic we propose starts from the mST over all terminal nodes. At this point, the mST does not contain any SPs and is therefore not a full tree topology. Thus, we need to transform the mST into a full tree topology. As mentioned in Section 3, and highlighted in Figure 5, this process is not unambiguous. In particular, for each terminal node $v$ with degree $d_v \geq 2$, we have to add $d_v - 1$ SPs. Consequently, there are $(2d_v - 3)!!$ ways to connect these SPs to the neighbors of $v$. Among all possible subtopologies connecting the SPs with $v$ and its neighbors, we choose the one given by the dendrogram defined by the hierarchical single linkage clustering algorithm applied to $v$ and its neighbors. In practice, this choice tends to work relatively well since nearby terminals are also closer in the subtopology.

Once we have a full tree topology, we can apply the geometry optimization step to obtain the optimal coordinates of the SPs. We assume that the optimal positions of the SPs indicate which connections between nodes might be more desirable, since they may be biased to move closer to other nodes than the ones to which they are connected. Therefore, we propose to recompute an mST over the terminals together with the SPs. This new mST defines a new topology that needs to be transformed

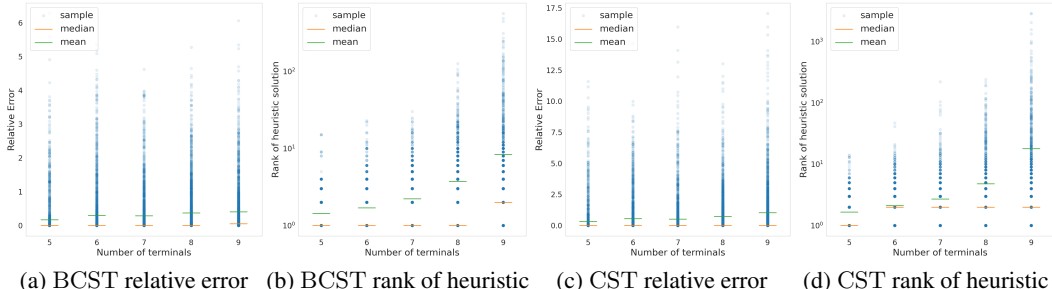

(a) BCST relative error  (b) BCST rank of heuristic  (c) CST relative error  (d) CST rank of heuristic

Figure 7: Relative cost errors between the mSTreg heuristic and optimal solutions; and sorted position of the heuristic tree for different number of terminals, $N$. For each $N$, we uniformly sampled 200 different terminal configuration and solved them for different $\alpha$ values. Most runs ended up close to the global optimum, though the heuristic is slightly better for the BCST problem.

into a full tree topology for the geometry optimization. Once we have a valid full tree topology, we recompute the optimal positions of the SPs. This process is repeated iteratively until convergence or until some stopping criterion is met. We refer to this algorithm as the mST regularization (mSTreg) heuristic. The steps of the algorithm are visualized in Figure 6, while its pseudocode and complexity analysis can be found in Appendixes K and L. We remark that the mSTreg heuristic is independent of the weighting factors that multiply the distances, thus it can also be used to approximate other problems as well, for instance a generalized version of the optimum communication tree with SPs.

Optionally, before the mST step is computed over the terminals and previous SPs, we can add intermediate points along the edges of the output generated by the geometry optimization step. These additional points will allow the mST to more reliably follow the edges of the geometry-optimized tree from previous step. Moreover, in case the initial topology was poor, these extra points may help to detect and correct edge crossings, which are known to be suboptimal. An illustration of the effect of these extra points can be found in Appendix M.

The heuristic designed for the BCST problem can also be applied to the CST problem by transforming BCST topologies at each iteration into CST topologies. While this transformation isn't unique, we found that iteratively collapsing one SP at a time with the neighbor that leads the smallest increase in cost produces compelling results. Additionally, when collapsing SPs together, centering the new node at the weighted geometric median of its new neighbors improves results slightly. Further details can be found in Appendix N.

## 4.3 BENCHMARK

To assess the quality of the mSTreg heuristic, we compare the cost of the trees computed by the mSTreg algorithm with the globally optimal solutions of configurations with up to nine terminals, obtained by brute force. We generate 200 instances with $N \in \{5, 6, 7, 8, 9\}$ terminals sampled from a unit square. Both CST and BCST problems are solved for each $\alpha \in \{0, 0.1, \ldots, 0.9, 1\}$. In Figure 7, the relative error, calculated as $100(c_h - c_o)/c_o$ where $c_h$ and $c_o$ are heuristic and optimal costs, is shown for different $N$. We also show how the heuristic solution ranks, once the costs of all topologies are sorted. The heuristic attains the optimum in the majority of cases, with slightly better performance in the BCST problem than in the CST one. Appendix O provides $\alpha$-based results.

In addition, we evaluate mSTreg on bigger datasets from the OR library[2] (Beasley, 1992) for the Steiner tree ($\alpha$=0) and the MRCT ($\alpha$=1) problems. This dataset includes exact solutions of Steiner problem instances of up to 100 nodes randomly distributed in a unit square. The used instances are labeled as e$n.k$, where $n$ denotes the number of terminals, and $k$ represents the instance number. Figure 8a compares the cost of our heuristic with the optimal cost. We also provide for reference the costs of the mST and the topology obtained by transforming the mST into a full tree topology with its SP coordinates optimized (referred to as mST_fulltopo). Though our heuristic does not reach the optimal cost, it produces good topologies. The average relative error is lower than 1%.

---

[2]http://people.brunel.ac.uk/~mastjjb/jeb/orlib/esteininfo.html

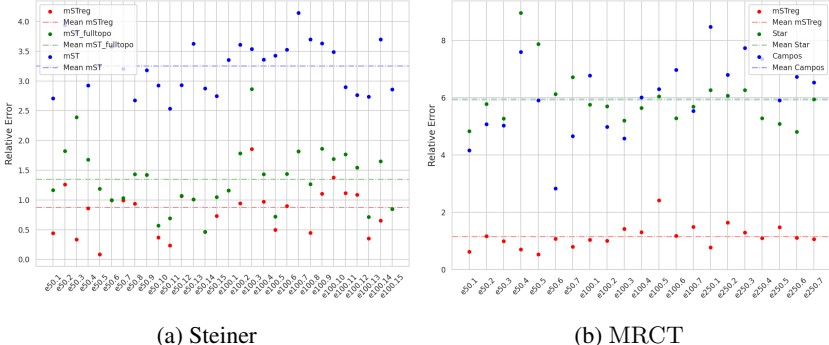

(a) Steiner        (b) MRCT

Figure 8: Relative cost error with respect to a reference cost for the Steiner and MRCT problems for different instances and different methods (lower is better). Left: For the Steiner problem, the reference cost is the optimal cost. mSTreg finds good solutions and improves over mST_fulltopo. Right: For MRCT the reference cost is given by the GRASP_PR algorithm. The heuristic beats all other methods, but is still sightly worse than GRASP_PR algorithm.

For the MRCT, we compare our heuristic with the Campos (Campos and Ricardo, 2008) and GRASP_PR (Sattari and Didehvar, 2015) algorithms. Campos modifies Prim's algorithm with heuristic rules, while GRASP_PR conducts local search by exchanging one edge at a time. We test the algorithms on the OR library datasets for problem instances with 50, 100 and 250 terminals. Figure 8b shows the relative errors. In this case, we do not have access to the optimal cost, therefore we use GRASP_PR costs cited from Sattari and Didehvar (2015) as reference. Campos cost are obtained from our own implementation. For reference, we also show the 2-approximation (Wong, 1980) given by the star graph centered at the data centroid. While mSTreg proves competitive (surpassing Campos but falling short of GRASP_PR by a modest average relative error of 1.16%), it is worth noting that GRASP_PR relies on a time-consuming local search. Leveraging the competitive solution provided by mSTreg as an initial step can enhance the performance and convergence of local search based algorithms, such as GRASP_PR. Appendix P ratifies this by showing GRASP_PR's improvement when initialized with mSTreg and provides additional comparisons across various $\alpha$.

## 5 CONCLUSIONS

We have introduced a novel problem, the (branched) central spanning tree, which encompasses the minimum spanning, the Steiner and minimum routing cost trees as limiting cases. The CST is a tree with low total communication cost between all nodes, thanks to it multiplying all edge costs with their centralities. The family is parameterized by $\alpha$, which regulates the influence of the centralities. We have focused on the Euclidean version of the problem, where the nodes are embedded in Euclidean space. Moreover, we presented a variant of the problem allowing the addition of extra nodes, referred to as Steiner points (SPs). In addition, we provided empirical evidence for the robustness of the (B)CST tree structure to perturbations in the data, which increases for higher $\alpha$. We contrast its noise resilience with mST using toy and single-cell transcriptomics datasets.

For the BCST problem, we give in Appendix H closed formulae for the branching angles at the Steiner points. Moreover, we showed that if the terminal nodes lie in the plane, then for $\alpha \in [0, 0.5] \cup \{1\}$ the optimal Steiner points have degree 3 at most, and we give evidence to support that this is also the case for $\alpha \in ]0.5, 1[$ (see Appendix I). It is an open question for which $\alpha > 1$ a phase transition for the feasibility of degree-4 SP occurs.

Based on an efficient algorithm to compute the optimal locations of the SPs, we have proposed the mSTreg heuristic, which exploits the optimal position of the SPs and the correspondence between the CST and BCST topologies to find approximate solutions for either. We benchmark this algorithm and show its competitiveness. Since the proposed heuristic is agnostic to the weighting factors that multiply the distances, we leave as future work to test whether it is equally competitive for other problems, like the general optimum communication tree. Another open question is whether the algorithm can be adapted to perform well on non-complete or non-Euclidean graphs.

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

OVERVIEW OF APPENDIX CONTENTS

The appendix is organized into the following sections:

## A  STABILITY EXAMPLES

In this section, we show how stable the BCST and CST for different $\alpha$ values are. We sample 1000 points uniformly from uniform distributions over different supports and perturb them by adding zero centered Gaussian noise. We generate two perturbations and show how the tree structure evolves

across different values. See Figures 9 10 and 11. As we increase the value of $\alpha$, the trees exhibit a more pronounced "star-shaped" pattern and enhanced stability. The parameter $\alpha$ provides a trade-off mechanism between preserving the structure of the data and ensuring the stability of the resulting tree.

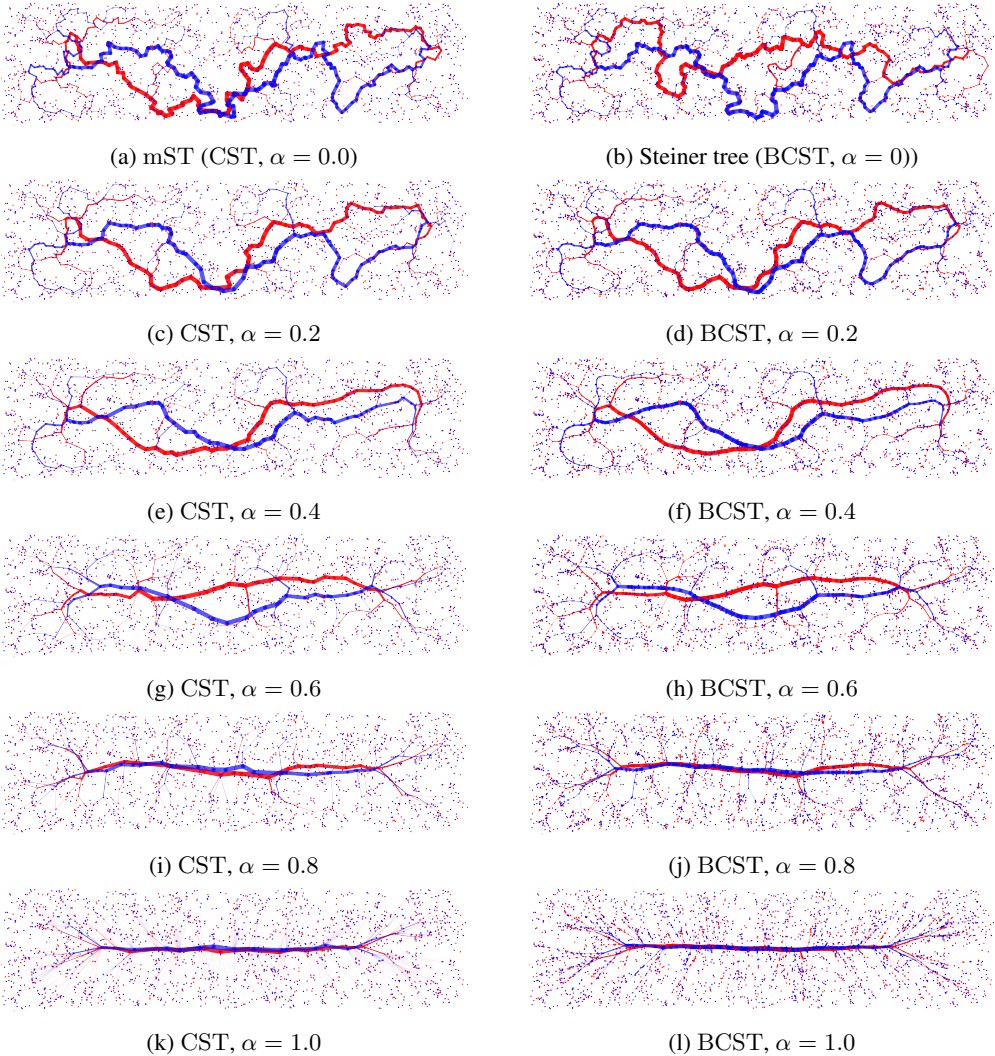

(a) mST (CST, $\alpha = 0.0$)   (b) Steiner tree (BCST, $\alpha = 0$))

(c) CST, $\alpha = 0.2$   (d) BCST, $\alpha = 0.2$

(e) CST, $\alpha = 0.4$   (f) BCST, $\alpha = 0.4$

(g) CST, $\alpha = 0.6$   (h) BCST, $\alpha = 0.6$

(i) CST, $\alpha = 0.8$   (j) BCST, $\alpha = 0.8$

(k) CST, $\alpha = 1.0$   (l) BCST, $\alpha = 1.0$

Figure 9: CST and BCST are computed for two perturbed instances generated by adding zero-centered Gaussian noise to points derived from a common sample uniformly taken within a rectangle. (B)CST for higher $\alpha$ values are more robust to noise and adhere to large scale structure in the data better. The width of each edge is proportional to its centrality. All trees except for the mST were computed using the heuristic proposed in Section 4.2.

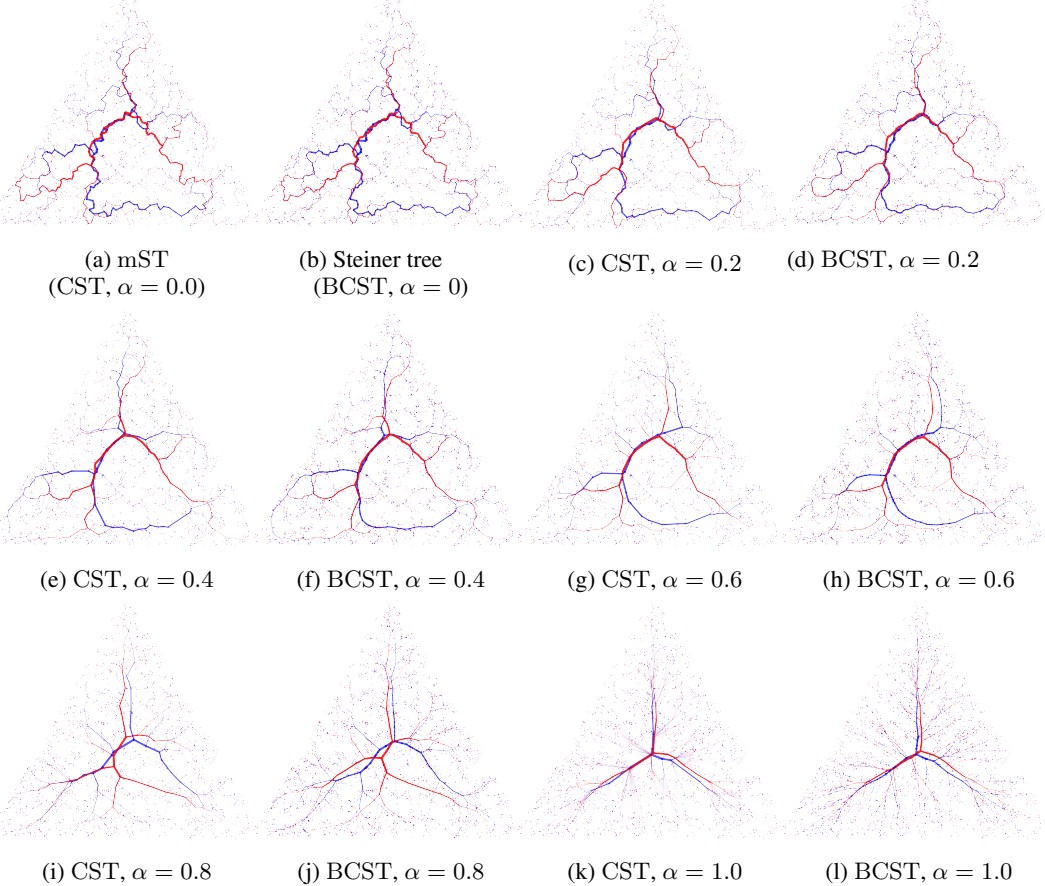

(a) mST
(CST, $\alpha = 0.0$)

(b) Steiner tree
(BCST, $\alpha = 0$)

(c) CST, $\alpha = 0.2$

(d) BCST, $\alpha = 0.2$

(e) CST, $\alpha = 0.4$

(f) BCST, $\alpha = 0.4$

(g) CST, $\alpha = 0.6$

(h) BCST, $\alpha = 0.6$

(i) CST, $\alpha = 0.8$

(j) BCST, $\alpha = 0.8$

(k) CST, $\alpha = 1.0$

(l) BCST, $\alpha = 1.0$

Figure 10: CST and BCST are computed for two perturbed instances generated by adding zero-centered Gaussian noise to points derived from a common sample uniformly taken within a triangle. (B)CST for higher $\alpha$ values are more robust to noise and adhere to large scale structure in the data better. The width of each edge is proportional to its centrality. All trees except for the mST were computed using the heuristic proposed in Section 4.2.

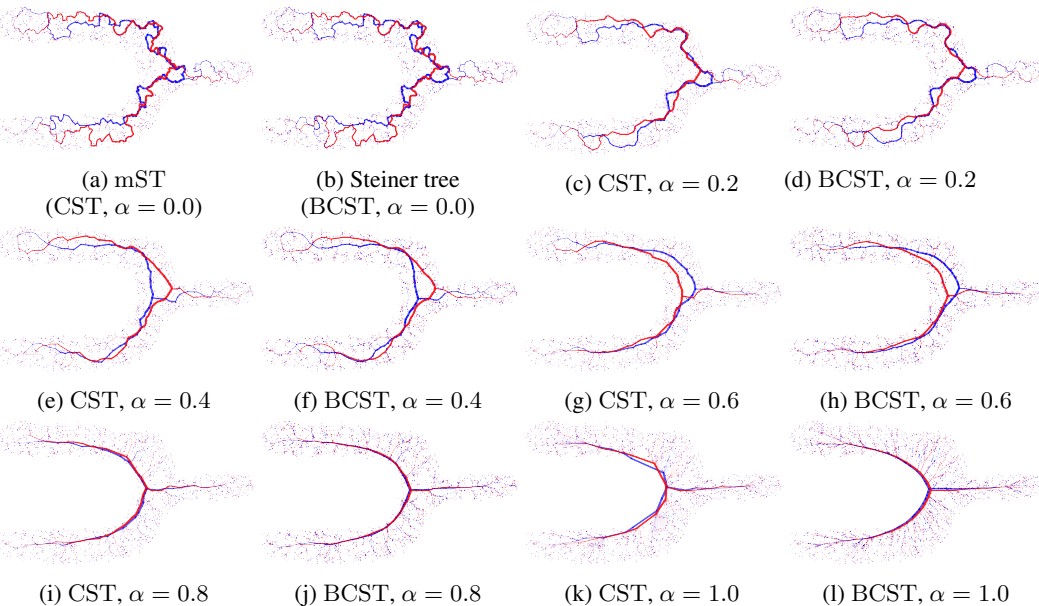

(a) mST
(CST, $\alpha = 0.0$)

(b) Steiner tree
(BCST, $\alpha = 0.0$)

(c) CST, $\alpha = 0.2$

(d) BCST, $\alpha = 0.2$

(e) CST, $\alpha = 0.4$

(f) BCST, $\alpha = 0.4$

(g) CST, $\alpha = 0.6$

(h) BCST, $\alpha = 0.6$

(i) CST, $\alpha = 0.8$

(j) BCST, $\alpha = 0.8$

(k) CST, $\alpha = 1.0$

(l) BCST, $\alpha = 1.0$

Figure 11: CST and BCST are computed for two perturbed instances generated by adding zero-centered Gaussian noise to points derived from a common sample uniformly taken within a non convex shape. (B)CST for higher $\alpha$ values are more robust to noise and adhere to large scale structure in the data better. The width of each edge is proportional to its centrality. All trees except for the mST were computed using the heuristic proposed in Section 4.2.

## B  APPLICATIONS

### B.1  SINGLE CELL TRANSCRIPTOMIC DATA

Single-cell transcriptomics analyzes the gene expression levels of individual cells in a particular population by counting the RNA transcripts of genes at a given time. The high dimensional single cell RNA-sequencing data can be used to model the gene expression dynamics of a cell population as well as the cell differentiation process. The reconstruction of these trajectories can help discover which genes are critical to understand the underlying biological process. It is often assumed that these trajectories can be represented as trees (Saelens et al., 2019; Street et al., 2018), and therefore the (B)CST can be applied to model such trajectories.

In this section we will give additional details on the example shown in the main paper regarding the Paul dataset. In addition, we show another example of the performance of the (B)CST with a different dataset, namely the Setty dataset (Setty et al., 2019)

**Paul dataset**   The Paul datset consists of gene expressions measurements of cells of mouse bone marrow Paul et al. (2015). The original dataset is formed by 2730 cells each with 3451 gene measurements. The data is preprocessed using the recipe described in Zheng et al. (2017), which reduces the dimensionality to 1000 by selecting the most relevant genes. We further reduce the dimensionality of the data, by applying PCA with 50 principal components. Finally, we apply the corresponding spanning tree algorithm. For visualization purposes, we used the PAGA algorithm (Wolf et al., 2019), one of the best algorithms for single cell trajectory inference (Saelens et al., 2019). PAGA was designed to faithfully represent the trajectories. Thus, if a spanning tree aligns well with the embedding, this is an indication that the tree approximates the trajectory well. In the main paper we showed the mST and CST, BCST (with $\alpha = 0.5$) of the original sample and a perturbed sample with 50% of the cells randomly sampled. In addition, here we include the CST and BCST with $\alpha$ set equal to 1, in order to be also able to compare against the MRCT (i.e CST with $\alpha = 1$). Figure 12 shows that the CST results in a star tree, and the BCST nearly resembles a star tree as well. In high-dimensional data, such as the Paul dataset with 50 dimensions, the star-shaped tendency

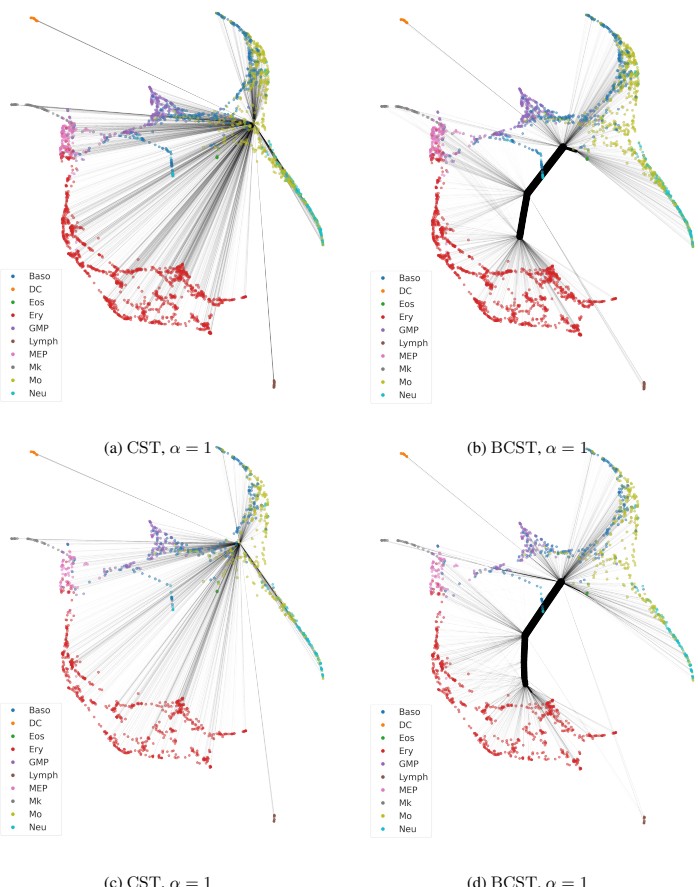

(a) CST, $\alpha = 1$        (b) BCST, $\alpha = 1$

(c) CST, $\alpha = 1$        (d) BCST, $\alpha = 1$

Figure 12: **CST and BCST with $\alpha = 1$ of the Paul dataset (Paul et al., 2015)**. We applied the algorithms on the original data (top) and perturbed version where half of the points have been randomly removed (bottom). The CST with $\alpha = 1$ becomes a star tree, and the BCST nearly approaches a star tree as well. Due to star-shape structure of the trees, both trees are robust to the perturbation but none of them model appropriately the data structure.

becomes more prominent. For $\alpha = 1$, most of the intricate structure of the data is lost. Therefore, for higher dimensions lower $\alpha$ values become more relevant.

**Setty dataset:** Extending our analysis, we applied the same experiment from Section 2.2 to a different real dataset: the human bone marrow dataset from Setty et al. (2019) referred here as the Setti dataset. After applying the same preprocessing as the one used for the Paul dataset, this dataset consists of 5780 samples within a 50-dimensional space, as opposed to the unprocessed data, which existed in a 14651-dimensional space. We computed BCST, CST, and mST on both the original dataset and a downsampled version (25% of the original), comparable in size to the Paul dataset's downsampled version. For visualization purposes we used the TSNE projection of the data provided by the scvelo library[3]. Figure 13 presents the results, showing a consistent pattern akin to the Paul dataset. BCST and CST exhibited superior robustness compared to mST, with BCST providing enhanced trajectory modeling[4]. This additional example reinforces the efficacy and robustness of our proposed methods.

---

[3]https://scvelo.readthedocs.io/en/stable/scvelo.datasets.bonemarrow/
[4]In this particular case, we have excluded the $\alpha = 1.00$ scenario. This is because, similar to the Paul dataset, both the CST and BCST algorithms yield star-shaped graphs under this condition.

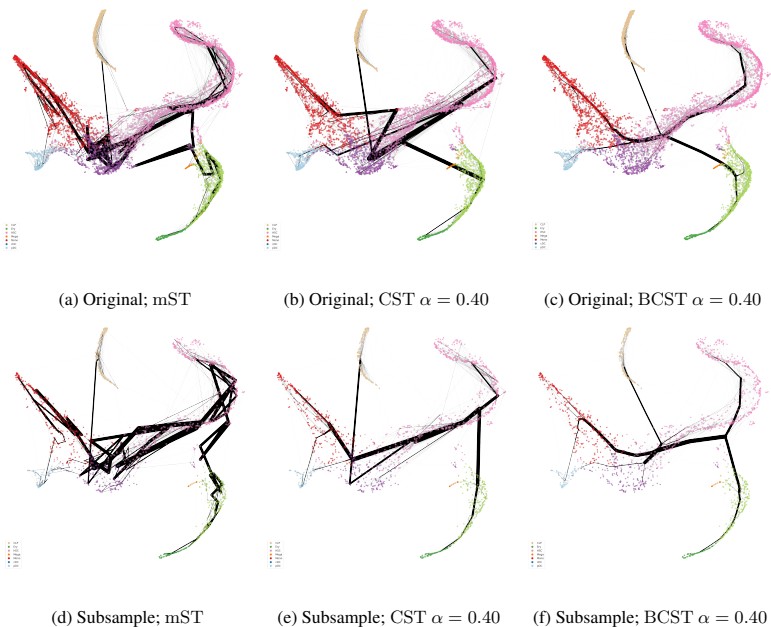

(a) Original; mST      (b) Original; CST $\alpha = 0.40$      (c) Original; BCST $\alpha = 0.40$

(d) Subsample; mST      (e) Subsample; CST $\alpha = 0.40$      (f) Subsample; BCST $\alpha = 0.40$

Figure 13: **mST, CST and BCST of the the Setty dataset (Setty et al., 2019)**. We applied the algorithms on the original data (top) and perturbed version where 25% of the points have been randomly removed (bottom). We use the TSNE algorithm to visualize the data in 2D, though the trees were computed on a 50 dimensional PCA projection of the preprocessed data. Colors correspond to different ground truth cell populations. The width of the edges is proportional to their centralities. The mST fails to accurately represent the trajectory and proves to be highly susceptible to noise. The CST and BCST are more faithful and robust, though the BCST performance is superior.

## B.2 3D Plant skeletonization

Plant skeletonization is a foundational technique for elucidating growth patterns, branching hierarchies, and responses to environmental factors in plants. It simplifies complex plant structures into skeletal representations, often described by spanning trees. In this section, we demonstrate how the BCST can model a plant's skeleton using a point cloud of its surface[5]. We provide additional examples beyond those presented in the main paper.

For the skeletonization process, we utilized the 4D Plant Registration Dataset[6]. It consists of 3D point cloud data that captures the surface of different plants at different growth stages. Specifically, our analysis focused on the point clouds of the tomato plant on days 5, 8, and 13. From each point cloud, we subsample uniformly at random 5000 points and then we compute the BCST.

Figure 14 presents the BCST results computed for different $\alpha$ values. When $\alpha = 0.00$, the tree branches exhibit greater irregularity, while at $\alpha = 1.00$, the finer details are obscured. Intermediate $\alpha$ values offer a more faithful representation of the plant's structure. However, some modeled branches deviate from the data, creating shortcut connections between points. This issue can be readily addressed by incorporating prior information about the root's position within the plant. By augmenting the point density near the root through virtual point replication (creating five times as many virtual points as there are original points at the root location), we encourage the branches to closely follow the natural branch density, resulting in a more faithful representation of the data. That is, if the original point cloud contains 5000 points, we introduce 25000 virtual points at the root location.

---

[5]Note that we do not test the CST in this context, as the ideal skeleton should closely follow the object's centerline. While the BCST has Steiner points, which naturally align with the center of the surrounding points to minimize the distance, the CST lacks this flexibility. Consequently, the backbone of the CST can not align with the centerline, given that the terminals lie on the surface.

[6]Data accessible at https://www.ipb.uni-bonn.de/data/4d-plant-registration/

Figure 15 displays the results after incorporating this prior information for the point cloud of the plant at different growth stages[7]. Notably, the model exhibits greater fidelity for intermediate $\alpha$ values when the prior is applied.

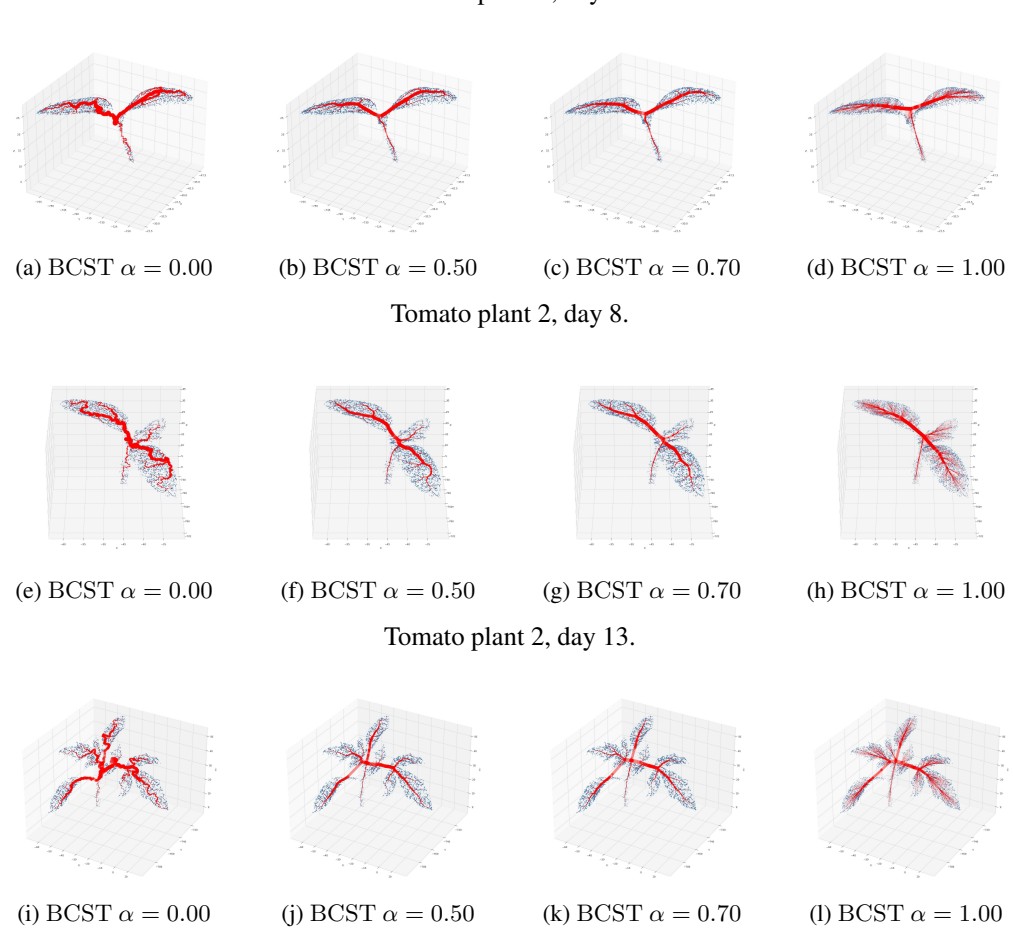

Tomato plant 2, day 5.

(a) BCST $\alpha = 0.00$     (b) BCST $\alpha = 0.50$     (c) BCST $\alpha = 0.70$     (d) BCST $\alpha = 1.00$

Tomato plant 2, day 8.

(e) BCST $\alpha = 0.00$     (f) BCST $\alpha = 0.50$     (g) BCST $\alpha = 0.70$     (h) BCST $\alpha = 1.00$

Tomato plant 2, day 13.

(i) BCST $\alpha = 0.00$     (j) BCST $\alpha = 0.50$     (k) BCST $\alpha = 0.70$     (l) BCST $\alpha = 1.00$

Figure 14: Skeletons at different $\alpha$ values of 3D point clouds of a tomato plant at different growth stages. The skeletons are modeled using the BCST with varying $\alpha$ values. With $\alpha = 0.00$, the tree branches exhibit greater irregularity, while at $\alpha = 1.00$ the finer details are missed. Intermediate $\alpha$ values offer a more faithful representation of the plant's structure. Nonetheless these may present some slight deviation, where some modeled branches do not align through the center of the actual branches (see day 13). This effect is alleviated once prior information concerning the root's location is added (see Figure 15).

## C   CST AS MCCNF

In this section, we will pose the central spanning tree (CST) problem as a minimum concave cost network flow (MCCNF) problem. The MCCNF problem minimizes the transportation cost of a commodity from sources to sinks. Here, the edge costs are modeled by concave functions that depend on the edge flow. Formally, given a demand vector $\mu \in \mathbb{R}^N$ with $\sum_{i=1}^{N} \mu_i = 0$ and a

---
[7]Day 8 of Figure 15 corresponds to Figure 4 in the main paper

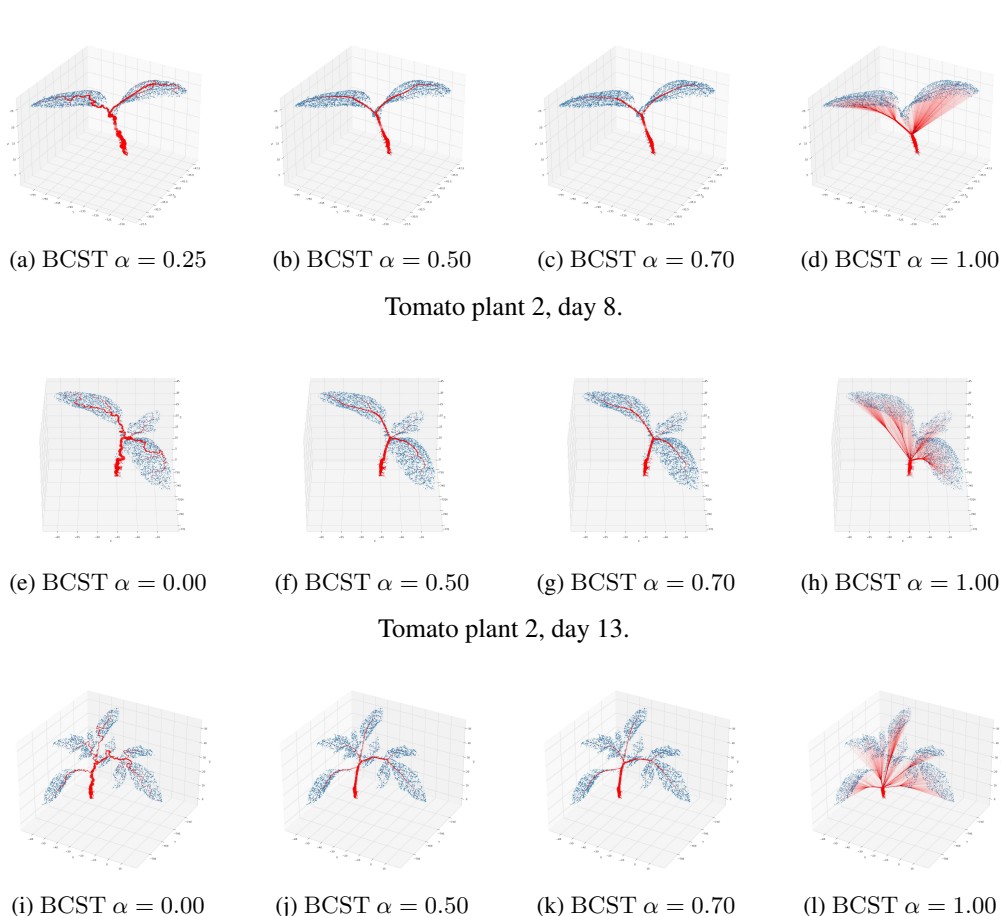

Figure 15: Skeletons at different $\alpha$ values of 3D point clouds of a tomato plant at different growth stages. The skeletons are modeled using the BCST with varying $\alpha$ values, incorporating prior information about the root's location. In contrast to Figure 14, we have enhanced the density of points at the root location, by virtually augmenting the number of points at the root's coordinates. This enhancement results in a more accurate and faithful representation of the plant's skeleton.

network $G = (V, E)$ with $N$ nodes, we define the MCCNF problem as

$$\min_f \sum_{ij \in E} C_{ij}(f_{ij}), \quad \text{subject to}$$

$$\sum_{(i,j) \in E} f_{ij} - \sum_{(j,i) \in E} f_{ji} = \mu_i, \qquad \forall i \in V \cdot \tag{3}$$

$$f_{ij} \geq 0$$

In the equation, $f_{ij}$ represent the flows associated with edge $(i, j)$ and $C_{ij}$ is a concave function dependent on $f_{ij}$ which determines the cost of the edge $(i, j)$. Note that the network defined by the flow, i.e. by the edges with $f_{ij} > 0$, does not necessarily have to be a tree. We will refer to nodes with negative demands as sources and nodes with positive demands as sinks.

To be able to represent the CST problem as an MCCNF, we need to identify the terms $m_e$ as flows. Since the function $(m_e(1 - m_e))^\alpha c_e$ is concave for $\alpha \in [0, 1]$, it will follow that the CST is an instance of the MCCNF problem.

Next, we will show how the $m_e$ can be interpreted as the flow along an edge of a particular single source flow problem. Consider a graph with $N$ nodes, where there is one source node $s$ with a mass

$(N-1)/N$ that needs to be transported to the rest of the nodes. Each sink node has a demand of $1/N$ mass. Thus, equation 3 becomes

$$\min_x \sum_{ij \in E} c_{ij} \big(f_{ij}(1-f_{ij})\big)^\alpha, \ \text{ subject to}$$
$$\sum_{(j,s) \in E} f_{js} - \sum_{(s,j) \in E} f_{sj} = \frac{N-1}{N},$$
$$\sum_{(i,j) \in E} f_{ij} - \sum_{(j,i) \in E} f_{ji} = \frac{1}{N}, \qquad \forall i \neq s$$
$$f_{ij} \geq 0$$

(4)

For single source flow problems, it is well-known that the optimal solutions can only form trees (Zangwill, 1968). We will show that for this particular problem, $f_{ij}$ is equal to $m_{ij}$ for any tree. Recall that for a given tree $T$, the value $m_e$ associated with an edge $e$ was defined as the number of nodes that lie on one of the sides of $e$ divided by the total number of nodes. Although for our purposes the chosen side of the edge is arbitrary, since the objective function is symmetrized thanks to being multiplied by $(1-m_e)$, that is not the case when we want to interpret it as a flow. However, which side to choose will be canonically determined by the flow direction.

Let $T$ be a feasible solution of equation 4, i.e. a tree. The flow $f_{ij}$ at edge $(i,j)$ of $T$ indicates the outgoing mass that is transported through the edge. This mass is equal to the sum of the demands of the nodes that lie in the side of the edge $(i,j)$ indicated by the flow. Given that each node has a demand of $1/N$, then the flow $f_{ij}$ is equal to $1/N$ multiplied by the number of nodes in the side in question, that is $f_{ij} = m_{ij}$.

**Remark C.1.** Considering the (B)CST from the perspective of an MCCNF problem, it becomes clear that it falls into the NP-hard category. Indeed, the authors of Guisewite and Pardalos (1991) showed that single-source MCCNF problems with strictly concave functions are NP-hard. Consequently, we conclude that the CST problem is NP-hard for $\alpha \in ]0,1]$ due to the strictly concave nature of the edge cost function $(m_e(1-m_e))^\alpha c_e$.

## C.1 RELATION TO THE BRANCHED OPTIMAL TRANSPORT PROBLEM

The branched or irrigation optimal transport (BOT) problem is also a particular MCCNF instance. In the BOT problem, the nodes are embedded in a Euclidean space and also allows for the inclusion of additional Steiner points. It is an extension of the optimal transport problem, distinguished by its diminishing costs which lead to a branching effect by promoting the joint transportation of mass.

Formally, the BOT problem minimizes

$$\min_{x_B, m_E} \sum_{(i,j) \in E} m_{ij}^\alpha \|x_i - x_j\|_2, \ \text{ subject to}$$
$$\sum_{(i,j) \in E} m_{ij} - \sum_{(j,i) \in E} m_{ji} = \mu_i, \qquad \forall i \in V \cup B,$$
$$m_{ij} \geq 0$$

(5)

where $m_{ij}$ is the flow transported along an edge $(i,j)$ and $B$ and $x_B$ are the set of SPs and their coordinates, respectively. As before, the vector $\mu$ represents the demands of the nodes. The demands of the SPs are set to zero.

In the scenario where there is a single source and all nodes share the same demand, the BCST and BOT problems differ only in the factors that multiply the distances. In the BOT problem, these factors correspond to $m_{ij}^\alpha$, representing the mass transported along an edge raised to the power of $\alpha$. In the BCST problem, the factors are given by $\big(m_{ij}(1-m_{ij})\big)^\alpha$, representing the centralities of the edges raised to the power of $\alpha$. It is worth mentioning, that both problems converge to the Steiner tree problem when $\alpha = 0$.

The primary distinction between the two problems lies in the selection of a source node. In the BOT problem, the selection of the source node determines the optimal topology of the network. However,

that is not the case for the BCST problem. Indeed, for the BCST problem, the specific source node chosen is irrelevant due to the symmetrization effect caused by the term $m_e(1 - m_e)$. In other words, the location of the source determines the edge orientation, which then defines the value of $m_e$. Nonetheless, this effect is nullified when multiplied by $(1 - m_e)$. This independence of the source node choice makes the BCST a more natural extension of the Steiner tree problem, since it does not require the choice of sources and sinks, unlike the BOT problem.

## D   EQUIVALENCE CST WITH $\alpha = 1$ AND MRCT

As mentioned in the main paper, the term $m_{ij}(1 - m_{ij})$ is proportional to the betweenness centrality of the edge $(i, j)$. This centrality quantifies the number of shortest paths that traverse the given edge. Thus, the multiplication of the length of each edge by its frequency in a shortest path is a rearrangement of the sum over all shortest path costs, i.e. the MRCT cost. Formally,

$$
\sum_{i,j \in V \times V} d_T(i, j) = \sum_{i,j \in V \times V} \sum_{(u,v) \in P_{ij}} ||x_u - x_v|| = \sum_{(u,v) \in T} \sum_{i,j \in V \times V} I_{(u,v) \in P_{ij}} ||x_u - x_v||
$$
$$
\propto \sum_{(u,v) \in T} m_{uv}(1 - m_{uv}) ||x_u - x_v||,
$$
(6)

where $d_T(i, j)$ is the shortest path distance in tree $T$ between $i$ and $j$ realized by the path $P_{ij}$ and $I_{(u,v) \in P_{ij}}$ is an indicator function, which equals 1 when the condition holds.

## E   TOPOLOGY ANALYSIS IN THE LIMIT CASE OF $\alpha \to \pm\infty$ FOR THE CST PROBLEM

In this section, we investigate the topologies of the limit cases of the CST as $\alpha$ approaches $\pm\infty$. We will use the following notation.

- $N$ will represent the number of terminals.
- For a given tree $T$ containing edge $(x, y)$, $m_{xy}^T$ indicates the proportion of nodes (normalized by $N$) that are reachable from $x$, once edge $(x, y)$ is removed from $T$. That is, the normalized number of nodes that lie in the side of $x$.
- For a given tree $T$, the term $\mathcal{N}_x$ denotes the set of neighbors of $x$ in $T$

### E.1   TOPOLOGY ANALYSIS IN THE LIMIT CASE OF $\alpha \to \infty$

In this case, the tree topology will tend to a star graph centered at the medoid. The medoid is defined as node that minimizes the total distance to all other nodes. When $\alpha$ is high, the CST aims to minimize the centrality of the edges, since the edge costs become relatively insignificant in comparison. Among all edges of a tree, the ones adjacent to a leave have the least centrality. Thus, any star graph is a tree that minimizes simultaneously the edge centrality of all its edges, with each of them having centrality equal to $(N - 1)/N$ (see Lemma $E.1$). In such case, we can factor out the edge centrality of the cost function in equation 1 and the problem reduces to find the star graph with minimum cost. As a result, when $\alpha$ tends to positive infinity, the optimal solution of the problem CST will be a star graph centered at the node that minimizes the total distance to all other nodes, that is, centered at the medoid. In the case of the BCST, where Steiner Points can be added, the star graph will be centered at the geometric median, as the Steiner Points will adjust to the location that minimizes the distance to all nodes.

The next Lemma formalizes why star graphs have lower cost than non-star graphs for $\alpha$ high enough in the CST problem.

**Lemma E.1.** As $\alpha$ approaches infinity ($\alpha \to \infty$), for any non-star-shaped tree $T$, there exists a star graph $T_\star$ with a lower CST cost.

*Proof.* We will demonstrate that it is always possible to increase the degree of a specific node in such a way that the CST cost of $T$ decreases. This process can be repeated until a single node with a degree of $N - 1$ is reached, resulting in the formation of a star graph.

Let $e = (u, v)$ be the most central edge of $T$. Let us assume w.l.o.g. that the number of nodes at $u$'s side of edge $e$ has more nodes than the side of $v$. Let $k \neq u \in \mathcal{N}_v$. We will show that the topology $T'$ which connects $k$ to $u$ instead of $v$ has a lower CST cost. The only edge centralities affected by this change are those associated with edges $(u, v)$, $(u, k)$, and $(k, v)$. To compare the costs of the topologies, it suffices to focus on these specific edges.

First, let's determine the values of the centralities for the edges in both trees:

- Normalized centrality of edge $(u, v)$ in $T$:

$$m_{uv}^T(1 - m_{uv}^T)$$

- Normalized centrality of edge $(u, v)$ in $T'$:

$$m_{uv}^{T'}(1 - m_{uv}^{T'}) = \left(m_{uv}^T + m_{kv}^T\right)\left(1 - m_{uv}^T - m_{kv}^T\right)$$

  The equality is a result of the fact that once $k$ becomes a neighbor of $u$, all nodes that were on the same side as $k$ will now be on the same side as $u$.

- Normalized centrality of edge $(k, v)$ in $T$ and centrality of edge $(k, u)$ in $T'$:

$$m_{kv}^T(1 - m_{kv}^T) = m_{ku}^{T'}(1 - m_{ku}^{T'})$$

  Both $u$ and $v$ lie in the same side of the edges, hence the equality of their normalized centralities.

Note that $m_{uv}^{T'}(1 - m_{uv}^{T'}) < m_{uv}^T(1 - m_{uv}^T)$, since we are adding more nodes to the side of $u$ which already had a greater number of nodes than the side of $v$. Additionally, since the side of $u$ already had more nodes than the side of $v$, we have $m_{uv}^T > 0.5$. The difference between the topologies of both costs are

$$
\begin{aligned}
\text{CST}(T) - \text{CST}(T') =& c_{uv}\left(m_{uv}^T\left(1 - m_{uv}^T\right)\right)^\alpha + c_{kv}\left(m_{kv}^T\left(1 - m_{kv}^T\right)\right)^\alpha \\
& - c_{uv}\left(\left(m_{uv}^T + m_{kv}^T\right)\left(1 - m_{uv}^T - m_{kv}^T\right)\right)^\alpha - c_{ku}\left(m_{kv}^T\left(1 - m_{kv}^T\right)\right)^\alpha \\
=& c_{uv}\left(\left(m_{uv}^T\left(1 - m_{uv}^T\right)\right)^\alpha - \left(\left(m_{uv}^T + m_{kv}^T\right)\left(1 - m_{uv}^T - m_{kv}^T\right)\right)^\alpha\right) \\
& + \left(c_{kv} - c_{ku}\right)\left(m_{kv}^T\left(1 - m_{kv}^T\right)\right)^\alpha \\
=& \frac{c_{uv}}{m_{kv}^T}\left(\frac{\left(m_{uv}^T\left(1 - m_{uv}^T\right)\right)^\alpha - \left(\left(m_{uv}^T + m_{kv}^T\right)\left(1 - m_{uv}^T - m_{kv}^T\right)\right)^\alpha}{m_{kv}^T}\right) \\
& + \left(c_{kv} - c_{ku}\right)\left(m_{kv}^T\left(1 - m_{kv}^T\right)\right)^\alpha \\
\geq & \frac{c_{uv}}{m_{kv}^T}\frac{dF}{dx}\left[m_{uv}^T\right] + \left(c_{kv} - c_{ku}\right)\left(m_{kv}^T\left(1 - m_{kv}^T\right)\right)^\alpha
\end{aligned}
\tag{7}
$$

where $F(x) = (x(1 - x))^\alpha$ and $\frac{dF}{dx}[\hat{x}]$ denotes its derivative evaluated at $\hat{x}$. In the inequality, we used the fact that the function $(F(x) - F(x + h))/h$ is decreasing with respect to $h$ when $x > 0.5$ and $x + h < 1$. As $h$ approaches 0, we recover the derivative of $F$.

By utilizing the fact that $\left(m_{kv}^T\left(1 - m_{kv}^T\right)\right) \leq \left(m_{uv}^T\left(1 - m_{uv}^T\right)\right)$ which follows from $(u, v)$ being the most central edge of $T$, we can proceed with the following calculations:

$$
\begin{aligned}
\frac{F(m_{kv}^T)}{\frac{dF}{dx}[m_{uv}^T]} &= \frac{(m_{kv}^T(1 - m_{kv}^T))^\alpha}{\alpha(m_{uv}^T(1 - m_{uv}^T))^{\alpha - 1}(2m_{uv}^T - 1)} \\
&\leq \frac{(m_{uv}^T(1 - m_{uv}^T))^\alpha}{\alpha(m_{uv}^T(1 - m_{uv}^T))^{\alpha - 1}(2m_{uv}^T - 1)} = \frac{m_{uv}^T(1 - m_{uv}^T)}{\alpha(2m_{uv}^T - 1)} \xrightarrow{\alpha \to \infty} 0
\end{aligned}
$$

Hence, for $\alpha$ high enough the first term of the last expression in equation 7 will dominate the second term resulting in a positive difference between $\text{CST}(T)$ and $\text{CST}(T')$. By repeating this process, we will eventually arrive at a star graph with a lower cost than the original tree $T$. $\qquad \square$

The next Lemma shows that in branched CST case, Steiner points tend to collapse with each other as $\alpha$ approaches infinity, such that an star-shaped graph is formed.

**Lemma E.2.** For any given topology $T$, as $\alpha$ approaches infinity ($\alpha \to \infty$), there exists a geometric arrangement in which all Steiner points are collapsed into a single node that has a lower BCST cost than any other arrangement where not all Steiner points are collapsed.

*Proof.* Consider $T$ as a topology in which not all Steiner points have been collapsed into a single point. Let $b$ represent the Steiner point that is adjacent to the edge with the highest centrality among all edges. We will demonstrate that collapsing all Steiner points with the Steiner point $b$ reduces the BCST cost.

We denote the centrality of the edge connecting nodes $u$ and $v$ as $\zeta_{uv} = m_{uv}(1 - m_{uv})$, and we represent the distance between nodes $u$ and $v$ as $d_{uv}$. Let $\zeta_* := \max \zeta_{uv}$. It is important to note that although $T$ does not collapse all Steiner points into one, the Steiner point $b$ may have already been collapsed with other Steiner points. In such cases, we consider the collapsed nodes as single one represented by the node $b$. Thus, $b$ may have degree higher than three and the distance from $b$ to all its neighbors is strictly positive, i.e. $d_{bu} > 0$ for all nodes, $u$, neighboring $b$. We denote by $T'$ the solution obtained by collapsing all Steiner points with node $b$, while preserving the topology. Note that in $T'$, only the distances have been updated, while the edge centralities remain the same as in $T$. Let $d'_{uv}$ represent the distances of $T'$. Notice that $d'_{ub} = 0$ for all Steiner points $u$ neighboring $b$, as all Steiner points have been collapsed to $b$. Now we are able to show that the cost of $T'$ is lower than the one of $T$ as $\alpha$ approaches infinity.

$$\text{BCST}(T) - \text{BCST}(T') = \sum_{(u,v) \in E_T} d_{uv} \zeta_{uv}^{\alpha} - \sum_{(u,v) \in E_T} d'_{uv} \zeta_{uv}^{\alpha} = \sum_{(u,v) \in E_T} (d_{uv} - d'_{uv}) \zeta_{uv}^{\alpha}$$

$$= \zeta_*^{\alpha} \underbrace{\left( \sum_{(u,v) \in E_T \setminus E_{\zeta_*}} (d_{uv} - d'_{uv}) \underbrace{\left( \frac{\zeta_{uv}}{\zeta_*} \right)^{\alpha}}_{<1} + \sum_{(u,v) \in E_{\zeta_*}} (d_{uv} - \underbrace{d'_{uv}}_{=0}) \right)}_{\xrightarrow{\alpha \to \infty} \sum_{(u,v) \in E_{\zeta_*}} d_{uv} > 0}$$

(8)

where $E_{\zeta_*} := \{(u,v) \in E_T : \zeta_{uv} = \zeta_*\}$. Note that we have utilized the fact that all edges in $E_{\zeta_*}$ are connected to $b$, implying that $d'_{uv} = 0$ for all $(u,v) \in E_{\zeta_*}$. Consequently, based on Equation equation 8, it follows that for a sufficiently high value of $\alpha$, the cost of $T'$ is lower than the one of $T$. This demonstrates the existence of a more optimal star-shaped arrangement of the Steiner points as $\alpha$ approaches infinity. $\square$

**Corollary E.1.** As $\alpha$ approaches infinity ($\alpha \to \infty$), the CST optimal solution is the star-shaped tree centered at the medoid of the terminals, that is, centered at the terminal which minimizes the sum of distances to all nodes. For the BCST case, the tree is centered at the geometric median of all terminals.

*Proof.* As a consequence of Lemma $E.1$, the optimal solution must be a star graph. In a star graph all edges have the same centrality, which is equal to $N - 1/N$, where $N$ is the number of terminals. Let $u$ denote the center node of a star graph, then its CST objective is equal to

$$\sum_{v \neq u} \frac{N-1}{N} c_{uv} = \frac{N-1}{N} \sum_{v \neq u} c_{uv}.$$

Thus the optimum star-graph is the one whose center minimizes $\sum_{v \neq u} c_{uv}$, i.e. the star-graph centered at the medoid. The analogous argument can be applied for the BCST case.

$\square$

## E.2 TOPOLOGY ANALYSIS IN THE LIMIT CASE OF $\alpha \to -\infty$

Negative values of $\alpha$ favour high central edges, as $(m_e(1 - m_e))^{\alpha}$ will be lower. Consequently, for sufficiently negative values of $\alpha$, the CST problem will prioritize minimizing the number of leaves

since the centrality of its adjacent edges attain the minimum centrality. The tree that minimizes the number of leaves is a path. Therefore, when $\alpha \to -\infty$ the optimum tree will be the Hamiltonian path which minimizes the CST objective function. A Hamiltonian path is a path that visits each node exactly once. The following lemma formalizes why a path will have a lower cost than any other graph for sufficiently negative values of $\alpha$.

**Lemma E.3.** In a graph where the triangle inequality holds strictly, if $\alpha$ tends to negative infinity $(\alpha \to -\infty)$, for any tree $T$ that is not a path, there exists a Hamiltonian path $T_\star$ with a lower CST cost.

*Proof.* We will show that for any node $u$ with degree higher than 2, we can always decrease its degree such that the CST cost of $T$ decreases. By repeating this process iteratively, the degrees of all nodes will eventually become at most 2, resulting in the formation of a path.

Let $v$ be a node with degree higher than 3. Let $(k, v)$ and $(u, v)$ be the two edges adjacent to $v$ with the lowest $m_{kv}^T$ and $m_{uv}^T$ values. Note that $m_{kv}^T < 0.5$ and $m_{uv}^T < 0.5$. Moreover, we can assume w.l.o.g. $m_{kv}^T \geq m_T^{uv}$ which implies

$$\left(m_{kv}^T\left(1 - m_{kv}^T\right)\right) \geq \left(m_{uv}^T\left(1 - m_{uv}^T\right)\right).$$

We will now demonstrate that the modified topology $T'$, where we connect node $k$ to node $u$ instead of node $v$, results in a lower CST cost. The only edge centralities affected by this change are those associated with edges $(u, v)$, $(u, k)$, and $(k, v)$. To compare the costs of the topologies, it suffices to focus on these specific edges.

First, we determine the values of the centralities for these edges in both trees.

- Normalized centrality of edge $(u, v)$ in tree $T$:

$$m_{uv}^T(1 - m_{uv}^T)$$

- Normalized centrality of edge $(u, v)$ in tree $T'$:

$$m_{uv}^{T'}(1 - m_{uv}^{T'}) = \left(m_{uv}^T + m_{kv}^T\right)\left(1 - m_{uv}^T - m_{kv}^T\right)$$

The equality is due to the fact that once $k$ is a neighbor of $u$, all nodes that were in the same side as $k$ will be now in the same side as $u$.

- Normalized centrality edge $k, v$ in $T$ and normalized centrality edge $k, u$ in $T'$:

$$m_{kv}^T(1 - m_{kv}^T) = m_{ku}^{T'}(1 - m_{ku}^{T'})$$

Both $u$ and $v$ lie in the same side of the edges, hence the equality of their normalized centralities.

Note that $m_{uv}^{T'}(1 - m_{uv}^{T'}) > m_{uv}^T(1 - m_{uv}^T)$. Otherwise, it would imply that

$$
\begin{aligned}
m_{uv}^{T'}(1 - m_{uv}^{T'}) &< & m_{uv}^T(1 - m_{uv}^T) &\iff \\
\left(m_{uv}^T + m_{kv}^T\right)\left(1 - m_{uv}^T - m_{kv}^T\right) &< & m_{uv}^T(1 - m_{uv}^T) &\iff \\
\min\left(m_{uv}^T + m_{kv}^T, 1 - m_{uv}^T - m_{kv}^T\right) &< & \min\left(m_{uv}^T, 1 - m_{uv}^T\right) = m_T^{uv}
\end{aligned}
$$

Trivially, $m_{uv}^T + m_{kv}^T < m_T^{uv}$ leads to a contradiction since $m_{kv}^T > 0$. Thus, the only possibility is

$$1 - m_{uv}^T - m_{kv}^T < m_T^{uv} \iff 1 - m_{kv}^T < 2m_{uv}^T \iff 1 - m_{kv}^T = m_{uv}^T + \sum_{i \in \mathcal{N}_v} m_{iv}^T < 2m_{uv}^T$$

$$\iff \sum_{i \in \mathcal{N}_v} m_{iv}^T < m_{uv}^T$$

which is also a contradition since by assumption $m_{uv}^T = \min_{i \in \mathcal{N}_v} m_{iv}^T$. Now we are able to show that the cost of $T'$ is lower than the one of $T$

$$\mathrm{CST}(T) - \mathrm{CST}(T') = c_{uv} \left( m_{uv}^T (1 - m_{uv}^T) \right)^\alpha + c_{kv} \left( m_{kv}^T (1 - m_{kv}^T) \right)^\alpha$$
$$- c_{uv} \left( \left( m_{uv}^T + m_{kv}^T \right) \left( 1 - m_{uv}^T - m_{kv}^T \right) \right)^\alpha - c_{ku} \left( m_{kv}^T (1 - m_{kv}^T) \right)^\alpha$$
$$= c_{uv} \left( \left( m_{uv}^T (1 - m_{uv}^T) \right)^\alpha - \left( \left( m_{uv}^T + m_{kv}^T \right) \left( 1 - m_{uv}^T - m_{kv}^T \right) \right)^\alpha \right) + (c_{kv} - c_{ku}) \left( m_{kv}^T (1 - m_{kv}^T) \right)^\alpha$$
$$= \left( m_{uv}^T (1 - m_{uv}^T) \right)^\alpha \left( c_{uv} - c_{uv} \underbrace{\left( \frac{\left( m_{uv}^T + m_{kv}^T \right) \left( 1 - m_{uv}^T - m_{kv}^T \right)}{\left( m_{uv}^T (1 - m_{uv}^T) \right)} \right)^\alpha}_{>1} + (c_{kv} - c_{ku}) \underbrace{\left( \frac{m_{kv}^T (1 - m_{kv}^T)}{m_{uv}^T (1 - m_{uv}^T)} \right)^\alpha}_{\geq 1} \right).$$

We can differentiate two cases. If $m_{kv}^T (1 - m_{kv}^T) = m_{uv}^T (1 - m_{uv}^T)$ then

$$\mathrm{CST}(T) - \mathrm{CST}(T') \xrightarrow{\alpha \to -\infty} \left( m_{uv}^T (1 - m_{uv}^T) \right)^\alpha (c_{uv} + c_{kv} - c_{ku}) > 0, \qquad (9)$$

where we have used the strict triangle inequality. Otherwise, the limit tends to

$$\mathrm{CST}(T) - \mathrm{CST}(T') \xrightarrow{\alpha \to -\infty} \left( m_{uv}^T (1 - m_{uv}^T) \right)^\alpha c_{uv} > 0.$$

Hence, for sufficiently negative values of $\alpha$, the difference $\mathrm{CST}(T) - CST(T')$ will be positive. By repeating this process, we can continue reducing the degree of nodes with degree higher than 2 until all nodes have degree at most 2. This process will eventually lead to the formation of a Hamiltonian path with a lower cost than the original tree $T$. $\qquad \square$

**Remark E.1.** Note that Lemma $E.3$ is only valid for graphs where the triangle inequality holds strictly, since equation equation 9 does not hold without it. Figure 16 shows an example where the strict triangle inequality is not satisfied, and therefore the Hamiltonian path is not optimal.

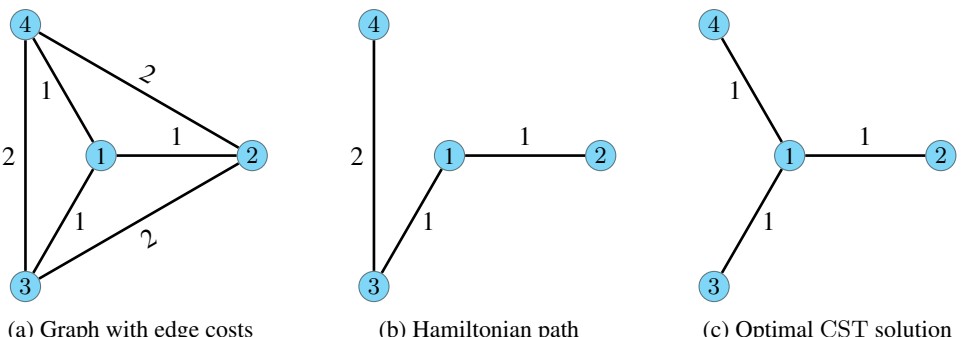

(a) Graph with edge costs          (b) Hamiltonian path          (c) Optimal CST solution

Figure 16: 16a) Graph with edge costs depicted, where the triangle inequality is not strictly satisfied. 16b) Optimal hamiltonian path with CST cost equal to $\frac{3^\alpha \cdot 3 + 4^\alpha}{16^\alpha}$. 16c) Optimal CST with cost equal to $\frac{3 \cdot 3^\alpha}{16^\alpha}$. Thus, if the triangle inequality does not strictly hold, the Hamiltonian path will not necessarily be optimal even for sufficiently negative $\alpha$ values

**Remark E.2.** Notice that Lemma $E.3$ does not generalize to the BCST case, since the Steiner Points have degree 3. For instance, consider the problem configuration where four terminals are positioned at the corners of a rectangle. If $\alpha$ approaches negative infinity, the optimal solution will connect the branching points with the terminals at 180 angles, forming an "H" shape topology.

## F    EXPLORING THE NUMBER OF DERIVABLE TOPOLOGIES FROM CST AND BCST TOPOLOGIES

### F.1    NUMBER OF BCST TOPOLOGIES DERIVABLE FROM A CST TOPOLOGY.

In this section, we explicitly determine the number of topologies of the BCST problem that can be derived from a single CST topology.

To derive a full tree topology $T_{\mathrm{BCST}}$ from a CST topology $T_{\mathrm{CST}}$ with $N$ terminals, we need to add $N - 2$ SPs. In particular, for each terminal node, $v$, with degree $d_v \geq 2$, we need to spawn $d_v - 1$ SPs. Since for $k$ terminal nodes, there exist a total $(2k-5)!!$ of full tree topologies (Schröder, 1870), there are $(2(d_v + 1) - 5)!! = (2d_v - 3)!!$ ways to connect the added SPs to the neighbors of $v$ and $v$ itself. Thus the total number of full tree topologies is equal to the number of possible combinations of subtopologies engendered per terminal neighborhood for terminals with degree higher than 2. Formally, this number is equal

$$\prod_{v \,:\, d_v \geq 2} (2d_v - 3)!!. \tag{10}$$

Note that, on the one hand, if all nodes have degree lower or equal than 2, i.e. the tree is a path, then a single full tree topology can be derived. On the other hand, if the original $T_{\mathrm{CST}}$ is a star graph, then there is a single graph with degree higher than 2, which is equal to $N - 1$. Thus, the total number of topologies derived from it is equal to $(2N - 5)!!$. This is the total number of possible full tree topologies, hence a star graph can generate any full tree topology. In general, the higher the degree of the nodes in $T_{\mathrm{CST}}$, the higher the number of derivable full tree topologies.

### F.2 NUMBER OF CST TOPOLOGIES DERIVABLE FROM A BCST TOPOLOGY.

In this case, we need to collapse each SP to a terminal. The collapse process can be carried out sequentially, where each SP is collapsed to one of its neighboring nodes, until no SPs remain. Naively, we might assume that there are $3^{N-2}$ possible topologies, given that each SP has 3 neighbors available for collapse and there are $N - 2$ SPs. However, this is not the case because some combinations may result in non-valid topologies. For instance, if all SPs collapse with neighbors that are also SPs, none of the SPs will be collapsed with a terminal node.

Before providing the formula for the number of CST topologies that can be derived from a full tree topology, let's introduce the All Minors Matrix Tree Theorem (Chaiken, 1982), which is necessary to derive the formula. The Matrix Tree Theorem (Tutte, 1984; Kirchhoff, 1847) states that the number of spanning trees of a given graph $G = (V, E)$ can be calculated from the determinant of a submatrix of its Laplacian. Recall that the Laplacian matrix of a graph $L$ is given by

$$L = D - A, \quad \text{where} \quad A_{ij} = \begin{cases} 1 & \text{if } (i, j) \in E \\ 0 & \text{otherwise} \end{cases} \quad \& \quad D_{ij} = \begin{cases} \sum_k A_{ik} & \text{if } i = j \\ 0 & \text{otherwise} \end{cases}.$$

The All Minors Matrix Tree Theorem generalizes the Matrix Tree Theorem. We state a simplified version of the theorem without providing a proof.

**Theorem F.1.** Given a graph $G = (V, E)$ and a subset $U$ of nodes in $G$, let $W = V \backslash U$. We define $L_{W,W}$ as the submatrix of the Laplacian matrix of $G$, which includes the rows and columns indexed by the nodes in $W$. In this context, the determinant of $L_{W,W}$, denoted as $\det(L_{W,W})$, provides a count of the number of spanning forests of $G$ that consist of $|U|$ disjoint trees, with the nodes in $U$ being disconnected across these trees.

*Proof.* See Chaiken (1982). $\qquad \square$

Now we are ready to present the main result of this subsection. Consider a full tree topology $T_{\mathrm{BCST}}$. The number of topologies for the CST problem that can be derived from $T_{\mathrm{BCST}}$ is given by

$$\det L_{N+1:, N+1:}, \tag{11}$$

where $L_{N+1:, N+1:}$ represents the submatrix of the Laplacian matrix $L$ of $T_{\mathrm{BCST}}$. This submatrix is formed by selecting the rows and columns associated with the SPs, which we assume, without loss of generality, are numbered from $N + 1$ to $2N - 2$. By virtue of Theorem $F.1$, equation equation 11 counts the number of forests of $T_{\mathrm{BCST}}$ which disconnect the terminal nodes. To demonstrate that this count of forests coincides with the number of topologies that can be derived from the full tree topology $T_{\mathrm{BCST}}$, we will establish a bijection.

Indeed, if we have a forest that disconnects all the terminals, each SP within the forest must belong to a component with a single terminal. In this scenario, we can unambiguously collapse each SP to its corresponding terminal. Once we have collapsed the SPs, we still need to reconnect the terminals between them to form a valid CST topology. Now, notice that in the original full tree topology

$T_{\mathrm{BCST}}$, each terminal is uniquely adjacent to a SP. We can connect the terminals between them based on the collapse process of the SPs. Specifically, a terminal $v_t$ is connected to another terminal $u_t$ if the neighboring SP of $v_t$ in the original $T_{\mathrm{BCST}}$ has been collapsed to $u_t$. Similarly, we can reverse these steps to map a CST topology back to a unique forest that disconnects the terminals. Figure 17 illustrates the individual steps of this bijection using two examples. We have proven the following theorem

**Theorem F.2.** Let $T_{\mathrm{BCST}}$ be a full tree topology with $N$ terminals. Consider the Laplacian matrix $L$ of $T_{\mathrm{BCST}}$. The number of CST topologies that can be derived from $T_{\mathrm{BCST}}$ is equal to the determinant of $L_{N+1:,N+1:}$, which is the submatrix of the Laplacian obtained by selecting the rows and columns associated with the SPs, which are numbered from $N + 1$ to $2N - 2$. Hence, the number of CST topologies derived from $T_{\mathrm{BCST}}$ can be calculated as $\det L_{N+1:,N+1:}$.

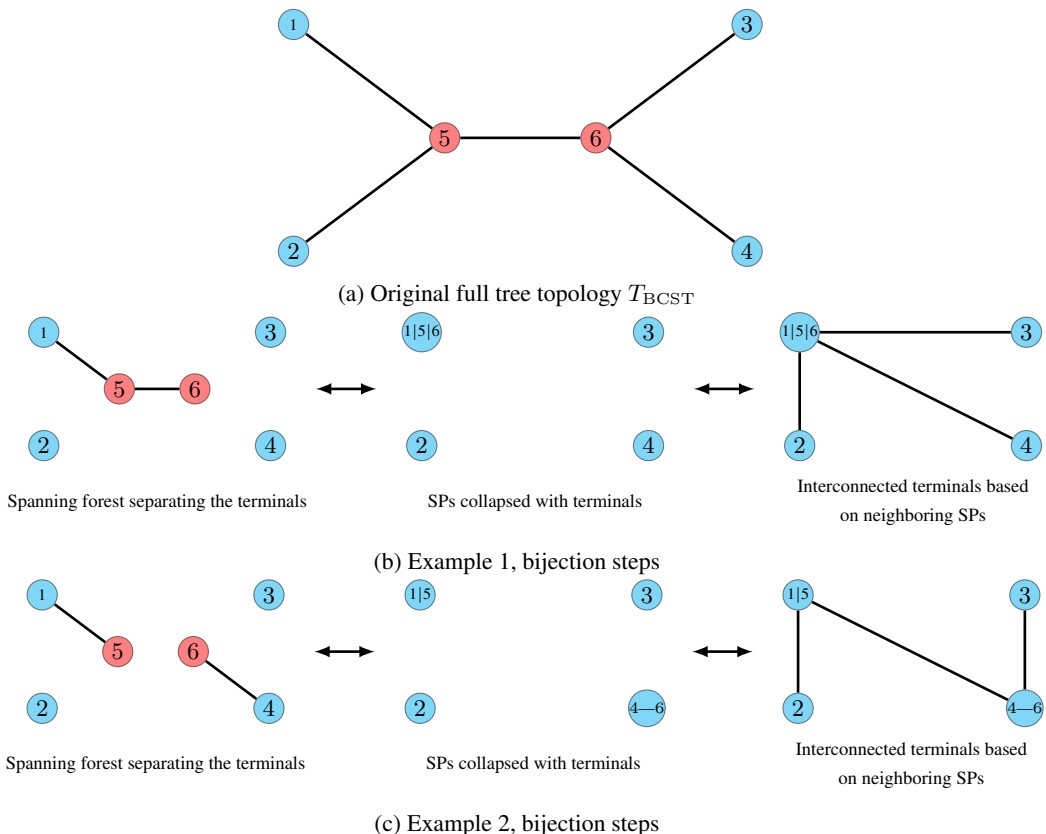

(a) Original full tree topology $T_{\mathrm{BCST}}$

Spanning forest separating the terminals        SPs collapsed with terminals        Interconnected terminals based on neighboring SPs

(b) Example 1, bijection steps

Spanning forest separating the terminals        SPs collapsed with terminals        Interconnected terminals based on neighboring SPs

(c) Example 2, bijection steps

Figure 17: The bijection between the set of CST topologies derived from a full tree topology $T_{\mathrm{BCST}}$ and the set of spanning forests of $T_{\mathrm{BCST}}$ that disconnect the terminals. Figures 17b and 17c illustrate two examples of the relationship between a spanning forest and a derived CST topology of a full tree topology depicted in Figure 17a.

# G    OPTIMAL CST AND BCST TOPOLOGIES ARE NOT GUARANTEED TO BE DERIVED FROM EACH OTHER

See Figure 18.

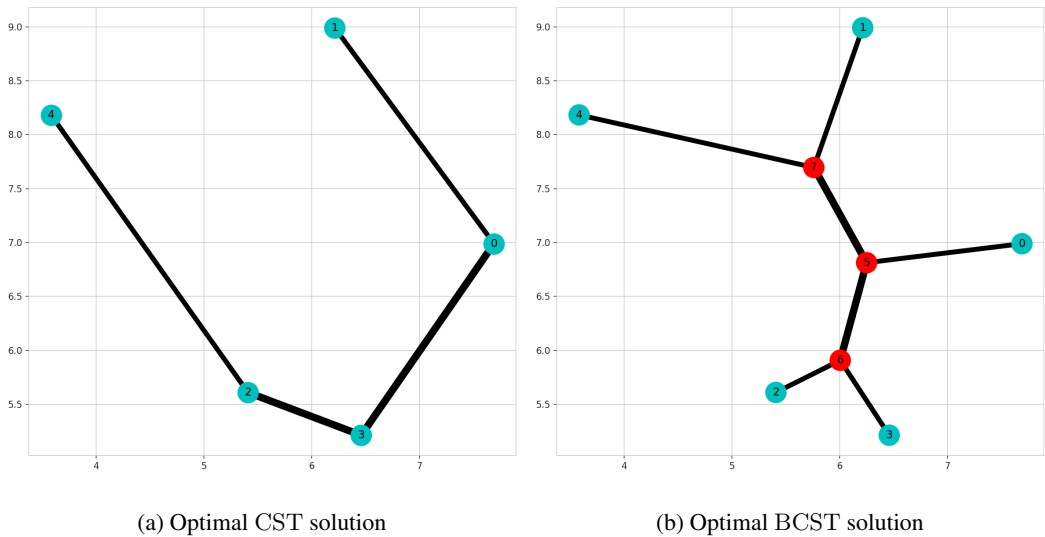

(a) Optimal CST solution                    (b) Optimal BCST solution

Figure 18: **Optimal CST and BCST topologies may not be derived from each other for the same terminal configuration.** Left: Optimal CST solution. Right: Optimal BCST solution. The CST topology cannot include nodes 4 and 1 as direct neighbors if derived from BCST, as it would result in nodes adjacent to a common neighbor. Similarly, the optimal BCST topology cannot be derived from the CST topology, as nodes 1 and 4 would not be connected to a common SP.

# H    BRANCHING ANGLES AT THE STEINER POINTS IN THE BCST PROBLEM

In this section, we formulate the branching angles in terms of the centralities of the edges for a given topology of the BCST problem.

First and foremost, we emphasize the locality characteristic of the geometric optimization of SPs of the BCST problem. Because of the convexity of the BCST objective equation 2, it can be shown that the geometric optimization of the SPs coordinates can be solved locally, meaning that the optimal position of a SP is determined by its neighbors and weighting factors. Lemma $H$.1 formalizes this statement. For a proof, we refer to Lemma 2.1 of Lippmann et al. (2022), where the same statement was shown for the branched optimal transport (BOT) problem. Since the proof is independent of the weighting factors of the distances, the result applies to the BCST problem as well.

**Lemma H.1.** Given a topology, its SPs are in optimal position w.r.t. the BCST problem if and only if any individual SP interconnects its neighbors at minimal cost. Moreover, the optimal topology of the BCST is optimal if and only if for any subset of connected nodes the corresponding subtopology solves the respective subproblem.

*Proof.* See Lemma 2.1 Lippmann et al. (2022).                    □

Recall that any feasible topology of the BCST problem can be represented as a full topology where each SP has degree 3. Thus, as a consequence of Lemma $H$.1, it is enough to study the geometric optimization of 3 nodes connected by a single SP. Consider the problem configuration depicted in Figure 19, where node $b$ represents the branching point whose coordinates need to be optimized, nodes $\{a_i\}$ are the terminals with fixed positions and $\zeta_i := m_{ba_i}(1 - m_{ba_i})$ are the centralities of

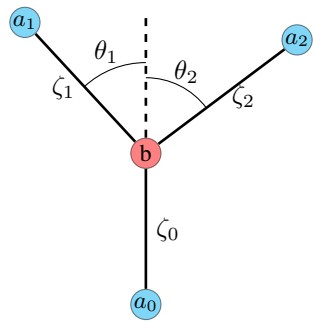

Figure 19: Branching angles at steiner point

the edges $(b, a_i)$. The objective to be minimized is

$$C(b) = \zeta_0||b - a_0|| + \zeta_1||b - a_1|| + \zeta_2||b - a_2|| \tag{12}$$

We will reproduce the arguments exposed for the BOT problem in Bernot et al. (2008) to determine the angles $\theta_1$ and $\theta_2$. We will differentiate two cases: when the SP does not coincide with any other terminal node; and when $b$ collapses with one of the terminals.

## H.1 Steiner point $b$ does not collaspe with a terminal

In this case, the function is differentiable with respect to $b$, and therefore we just need to see where the gradient of equation equation 12 is equal to zero. The formula for the gradient is as follows

$$\nabla_b C(b) = \zeta_0^\alpha n_0 + \zeta_1^\alpha n_1 + \zeta_2^\alpha n_2, \tag{13}$$

where $n_i = \frac{b - a_i}{||b - a_i||}$. By applying the dot product to $\nabla_b C(b)$ with each $n_i$ and setting it equal to zero, we derive the following equalities:

$$\langle \nabla_b C(b), n_0 \rangle = 0 \quad \rightarrow \quad \zeta_0^\alpha + \zeta_1^\alpha \underbrace{\langle n_1, n_0 \rangle}_{-\cos(\theta_1)} + \zeta_2^\alpha \underbrace{\langle n_2, n_0 \rangle}_{-\cos(\theta_2)} = 0$$

$$\langle \nabla_b C(b), n_1 \rangle = 0 \quad \rightarrow \quad \zeta_0^\alpha \underbrace{\langle n_0, n_1 \rangle}_{-\cos(\theta_1)} + \zeta_1^\alpha + \zeta_2^\alpha \langle n_2, n_1 \rangle = 0$$

$$\langle \nabla_b C(b), n_2 \rangle = 0 \quad \rightarrow \quad \zeta_0^\alpha \underbrace{\langle n_0, n_2 \rangle}_{-\cos(\theta_2)} + \zeta_1^\alpha \langle n_1, n_2 \rangle + \zeta_2^\alpha = 0$$

Solving the linear system we obtain that the angles satisfy

$$
\begin{aligned}
\cos(\theta_1) &= \frac{\zeta_0^{2\alpha} + \zeta_1^{2\alpha} - \zeta_2^{2\alpha}}{2\zeta_0^\alpha \cdot \zeta_1^\alpha} \\
\cos(\theta_2) &= \frac{\zeta_0^{2\alpha} + \zeta_2^{2\alpha} - \zeta_1^{2\alpha}}{2\zeta_0^\alpha \cdot \zeta_2^\alpha} \\
\cos(\theta_1 + \theta_2) &= \frac{\zeta_0^{2\alpha} - \zeta_1^{2\alpha} - \zeta_2^{2\alpha}}{2\zeta_1^\alpha \cdot \zeta_2^\alpha}
\end{aligned}
\tag{14}
$$

## H.2 Steiner point $b$ collapses with a terminal

In this case, in order to determine the optimality angles, we will use the subdifferential argument applied in Lippmann et al. (2022). W.l.o.g. we will assume that $b$ collapses with terminal $a_0$.

The subdifferential of a convex function $h : \mathbb{R}^n \to \mathbb{R}$ at $x$ is defined as the following set of vectors

$$\partial g(x) \coloneqq \{v \; : \; h(z) \geq h(x) + \langle v, z - x \rangle, \; \forall z \in \mathbb{R}^n\}.$$

In other words, $\partial g(x)$ comprises all vectors $v$ such that the line passing through $h(x)$ in the direction of $v$ lies below the function $h$ at all points. Each of these vectors is called a subgradient of $h$ at $x$. When a function is differentiable at $x$, the subdifferential only contains the gradient of the function at $x$.

Fermat rule states that a convex function attains its minimum at $x$ if and only if $0 \in \partial g(x)$. Furthermore, the subdifferential of two convex functions is equal to the union of the pairwise sums of their subgradients. In other words, for $g(x) = g_1(x) + g_2(x)$ then

$$\partial g(x) = \{v_1 + v_2 \; : \; v_1 \in \partial g_1(x), \; v_2 \in \partial g_2(x)\}. \tag{15}$$

We can apply Fermat's rule to determine when the minimum is attained at $b = a_0$. For the function $g(x) = w \cdot ||x - a||$, the subdifferential is given by

$$\partial g(x) = \begin{cases} \{v \; : \; ||v|| \leq w\}, & \text{if } x = a \\ \left\{ w \frac{x-a}{||x-a||} \right\}, & \text{otherwise} \end{cases}.$$

Thus, applying equation equation 15, the subdifferential of $C(b)$ at $b = a_0$ is given by

$$\partial C(a_0) = \left\{ v + \zeta_1^\alpha \frac{b - a_1}{||b - a_1||} + \zeta_2^\alpha \frac{b - a_2}{||b - a_2||} \; : \; ||v|| \leq \zeta_0^\alpha \right\}.$$

In order for $b$ to be optimal at $a_0$, zero has to belong to $\partial C(a_0)$, which is true if and only if

$$
\left\| \zeta_1^\alpha \frac{b - a_1}{||b - a_1||} + \zeta_2^\alpha \frac{b - a_2}{||b - a_2||} \right\| \leq \zeta_0^\alpha
$$
$$
\iff \left\| \frac{b - a_1}{||b - a_1||} + \frac{b - a_2}{||b - a_2||} \right\|^2 = \zeta_1^{2\alpha} + \zeta_2^{2\alpha} + 2\zeta_1^\alpha \zeta_2^\alpha \cos(\gamma) \leq \zeta_0^{2\alpha}
\tag{16}
$$

where $\gamma$ is the angle of the terminal triangle at $a_0$, that is $\gamma := \angle a_1 a_0 a_2$. Isolating $\gamma$, we get

$$\gamma \geq \arccos\left( \frac{\zeta_0^{2\alpha} - \zeta_1^{2\alpha} - \zeta_2^{2\alpha}}{2\zeta_1^\alpha \cdot \zeta_2^\alpha} \right) = \theta_1 + \theta_2. \tag{17}$$

Thus $b$ will collapse to $a_0$ if the angle $\angle a_1 a_0 a_2$ is greater than the optimal angle given by equation 14. In such cases, the resulting branching is referred to as a $V$-branching.

**Remark H.1.** It is worth noting that the reasoning presented in this section remains independent of the weighting factors, which, in our case, were set equal to the normalized edge centralities powered to $\alpha$. As a result, this finding holds true for any weights and can be used to determine an arbitrary weighted geometric median of three points. Furthermore, we emphasise that the position of the SP, $b$, depends exclusively on the angles and weighting factors and not on the distances between the terminal nodes.

## I  INFEASIBILITY OF DEGREE-4 STEINER POINTS IN THE PLANE

In this section, we will prove the infeasibility of degree 4 SPs in the optimal solution of the BCST. Specifically, we will focus on the scenario where the terminal nodes lie in the plane and the value of $\alpha$ falls within the range $\alpha \in [0, 0.5] \cup 1$. Moreover, we will provide compelling evidence to support the validity of the statement for the case where $\alpha \in \,]0.5, 1[$. We will divide the proof into two parts: one for $\alpha \in [0, 0.5]$ and the other for $\alpha = 1$. We divide it into two sections to account for the distinct approaches required for each case.

### I.1  INFEASIBILITY OF DEGREE-4 STEINER POINT FOR $\alpha \in [0, 0.5]$

The optimality of a solution in the BCST problem relies on the locality characteristic, as stated in Lemma $H.1$. Specifically, each subtopology within a connected subset must solve its respective problem for the overall solution to be optimal. Consequently, the realization of a degree 4 SP, as depicted in Figure 20, requires the collapse of $b_2$ with $b_1$. Moreover, this collapse must occur in any topology. As we have discussed in H.2, both nodes $b_1$ and $b_2$ will collapse if a $V$-branching

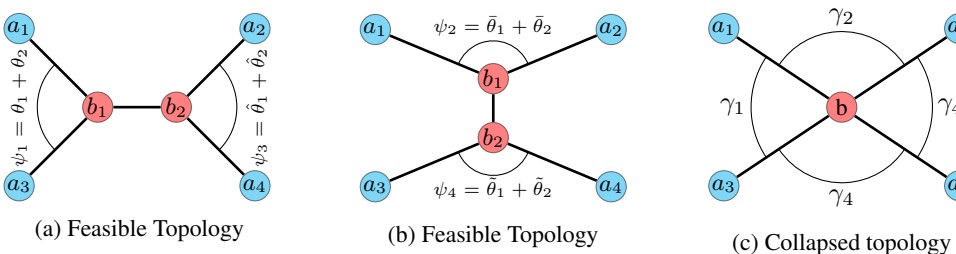

(a) Feasible Topology $\qquad$ (b) Feasible Topology $\qquad$ (c) Collapsed topology

Figure 20: Figures 20a and 20b depict two topologies, where the optimal angles given by equation equation 17 are represented by $\psi_i$. Figure 20c illustrates the collapsed solution with the corresponding angles $\gamma_i$. An essential requirement for the optimality of Figure 20c is that $\psi \leq \gamma_i$.

occurs, that is if the angle realized between the collapsed node and the two other nodes connected to it exceeds the optimal angle given by equation 17. Therefore, for the Figure 20 it follows that $\gamma_i \geq \psi$. We will demonstrate that the sum of $\sum_{i=1}^{4} \gamma_i >$ is greater than $2\pi$, rendering aSP of degree 4 infeasible. To do this we will prove that $\psi_i > \pi/2$ for all $i$.

We will utilize the notation introduced earlier in Section E for $F(x)$ and $m_{xy}^T$, but since the topology is constant, we will omit the $T$ and use $m_{xy}$ instead. W.l.o.g. let us consider $i = 1$ and denote $\psi_1$ as $\psi$. If $\cos(\psi) < 0$, the angle $\psi \in [0, \pi]$ will be greater than $\pi/2$. Based on equation 17, we can derive the following:

$$\cos(\psi) = \frac{F\left(m_{a_3 b_1} + m_{a_1 b_1}\right)^{2\alpha} - F\left(m_{a_3 b_1}\right)^{2\alpha} - F\left(m_{a_1 b_1}\right)^{2\alpha}}{2F\left(m_{a_3 b_1}\right)^{\alpha} F\left(m_{a_1 b_1}\right)^{\alpha}} \tag{18}$$

The function $F(x)^{2\alpha} = (x(1-x))^{2\alpha}$ is strictly subadditive in $\mathbb{R}^+$ for $\alpha \in [0, 0.5]$,[8] that is $F(x + y)^{2\alpha} < F(x)^{2\alpha} + F(y)^{2\alpha}$ if $x, y > 0$. Thus, the numerator of equation 18 is negative and therefore $\cos(\psi) < 0$. Consequently, $\psi > \pi/2$. This argument applies to all $\psi_i$, hence their sum will be greater than $2\pi$. Consequently, a SP with a degree of 4 cannot be part of an optimal solution. We have just shown the next theorem.

**Theorem I.1.** Let $\alpha \in [0, 0.5]$. Given a set of terminals which lie in the plane, then the SPs of the optimal solution of the BCST problem will not contain SPs of degree 4 unless this collapse with a terminal.

## I.2 INFEASIBILITY OF DEGREE-4 STEINER POINT FOR $\alpha = 1$

For higher $\alpha > 0.5$ the argument used in the previous section does not apply. Indeed, we can find values for the edge centralities for which the lower bounds of $\gamma_i$, given by equation 17, are all lower than $\pi/2$.[9] In this section we will take a more general approach that can rule out degree-4 branching for higher $\alpha$ values. In concrete, we will show the infeasibility for $\alpha = 1$.

The optimal position of the SPs is continuously dependent on the terminal positions and solely relies on the branching angles, as demonstrated in Section H. Consequently, assuming that there exists a configuration such that the SPs collapse, it is possible to find terminal positions that lead to an unstable collapse of the SPs. Here, instability refers to a configuration where an infinitesimal translation of the terminals results in the splitting of the SPs. This scenario is depicted in Figure 21. In such cases, the angles realized by the terminals and the SPs will reach the upper bounds specified by equation 17. Therefore, the angles depicted in Figure 21a fulfill the condition

$$\gamma = \pi - \theta_1 - \theta_2, \tag{19}$$

---

[8]In fact, all concave functions, $f$, with $f(0) \geq 0$ are subadditive on the positive domain. It is easy to see that $(x(1-x))^{2\alpha}$ is concave for $\alpha \in [0, 0.5]$.

[9]For instance for $\alpha = 1$, $m_{a_3 b_1} = m_{a_1 b_1} = 0.2$ and $m_{a_2 b_2} = m_{a_4 b_2} = 0.3$ all $\psi_i$ angles are acute. Hence their sum is also lower than $2\pi$.

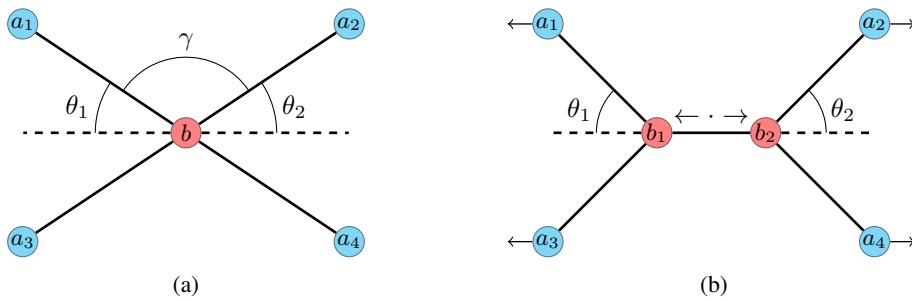

(a)  (b)

Figure 21: Figure 21a depicts the collapsed solution of a 4-terminal configuration. Figure 21b demonstrates that it is possible to move jointly the terminal points $\{a_1, a_3\}$ in a specific but opposite direction to the one of the terminals $\{a_2, a_4\}$, resulting in the splitting of the collapsed SP $b$ into two distinct SPs, $b_1$ and $b_2$. Remarkably, this split can be executed while preserving the angles $\theta_1$ and $\theta_2$. Importantly, these angles must correspond to the optimal angles given by equation 14.

where the angles satisfy

$$\cos(\gamma) = \frac{F\left(m_{a_1b} + m_{a_2b}\right)^{2\alpha} - F\left(m_{a_1b}\right)^{2\alpha} - F\left(m_{a_2b}\right)^{2\alpha}}{2F\left(m_{a_1b}\right)^\alpha F\left(m_{a_2b}\right)^\alpha} \tag{20}$$

$$\cos(\theta_1) = \frac{F\left(m_{a_3b} + m_{a_1b}\right)^{2\alpha} + F\left(m_{a_1b}\right)^{2\alpha} - F\left(m_{a_3b}\right)^{2\alpha}}{2F\left(m_{a_3b} + m_{a_1b}\right)^\alpha F\left(m_{a_1b}\right)^\alpha} \tag{21}$$

$$\cos(\theta_2) = \frac{F\left(m_{a_2b} + m_{a_4b}\right)^{2\alpha} + F\left(m_{a_2b}\right)^{2\alpha} - F\left(m_{a_4b}\right)^{2\alpha}}{2F\left(m_{a_2b} + m_{a_4b}\right)^\alpha F\left(m_{a_2b}\right)^\alpha} \tag{22}$$

We can manipulate equation 19 in the following way

$$\gamma = \pi - \theta_1 - \theta_2 \iff \cos(\gamma - \pi) = \cos(-\theta_1 - \theta_2) \iff -\cos(\gamma) = \cos(\theta_1 + \theta_2)$$
$$\underset{*}{\iff} -\cos(\gamma) = \cos(\theta_1)\cos(\theta_2) - \sqrt{\left(1 - \cos(\theta_1)^2\right)\left(1 - \cos(\theta_2)^2\right)} \tag{23}$$

where in (*) we have used the fact that

$$\cos(x + y) = \cos(x)\cos(y) - \sin(x)\sin(y) = \cos(x)\cos(y) - \sqrt{(1 - \cos(x)^2)(1 - \cos(y)^2)}.$$

If we square both sides of 23 and equate to 0 we obtain.

$$\left(\cos(\gamma) + \cos(\theta_1)\cos(\theta_2)\right)^2 - \left(1 - \cos(\theta_1)^2\right)\left(1 - \cos(\theta_2)^2\right) = 0. \tag{24}$$

Equation equation 24 depends on the variables $m_{a_1b}$, $m_{a_2b}$, $m_{a_3b}$, and $\alpha$[10]. Equation equation 24 is generally too complex to be solved analytically. However, with the help of the Mathematica software (Inc., 2022), we have determined that for $\alpha = 1$, the equality does not hold within the constraints of the problem, namely $\sum_{i=1}^{4} m_{a_i,b} = 1$ and $0 < m_{a_i,b} < 1$ for all $i$.

To simplify the notation, let's denote $m_{a_ib}$ as $m_i$. For $\alpha = 1$, when we expand equation equation 24, we find that the numerator of the formula becomes a fourth-degree polynomial with respect to $m_1$. The four roots, $\{m_1^{(j)}\}$ of the polynomial are

$$m_1^{(1)} = \frac{1}{2}\left(1 - m_2 - m_3 - \sqrt{(-1 + m_2 + m_3)^2 - \frac{4}{3}\left(-2m_2 - 2m_3 + 3m_2m_3 - \sqrt{m_2^2 - m_2m_3 + m_3^2}\right)}\right)$$

$$m_1^{(2)} = \frac{1}{2}\left(1 - m_2 - m_3 + \sqrt{(-1 + m_2 + m_3)^2 - \frac{4}{3}\left(-2m_2 - 2m_3 + 3m_2m_3 - 2\sqrt{m_2^2 - m_2m_3 + m_3^2}\right)}\right)$$

$$m_1^{(3)} = \frac{1}{2}\left(1 - m_2 - m_3 - \sqrt{(-1 + m_2 + m_3)^2 - \frac{4}{3}\left(-2m_2 - 2m_3 + 3m_2m_3 + 2\sqrt{m_2^2 - m_2m_3 + m_3^2}\right)}\right)$$

$$m_1^{(4)} = \frac{1}{2}\left(1 - m_2 - m_3 + \sqrt{(-1 + m_2 + m_3)^2 - \frac{4}{3}\left(-2m_2 - 2m_3 + 3m_2m_3 + 2\sqrt{m_2^2 - m_2m_3 + m_3^2}\right)}\right)$$

$$\tag{25}$$

[10]Since $\sum_{i=1}^{4} m_{a_ib} = 1$, $m_{a_4b}$ can be expressed as $1 - m_{a_1b} - m_{a_3b} - m_{a_4b}$.

where we have highlighted the difference between the roots. We will show that $1 < m_1^{(4)} + m_2 + m_3 < m_1^{(2)} + m_2 + m_3$ and $m_1^{(1)} < m_1^{(3)} < 0$, which implies that the problem constraints are not satisfied, and therefore SPs of degree are not possible.

**Claim 1: $1 < m_1^{(4)} + m_2 + m_3 \leq m_1^{(2)} + m_2 + m_3$.** From equation 25 it is clear that $m_1^{(4)} \leq m_1^{(2)}$. Thus, it is enough to prove the inequality for $m_1^{(4)}$:

$$
\begin{aligned}
m_1^{(4)} + m_2 + m_3 &= \frac{1}{2}(1 + m_2 + m_3) + \frac{1}{2}\sqrt{(-1 + m_2 + m_3)^2 - \frac{4}{3}\left(-2m_2 - 2m_3 + 3m_2 m_3 - \sqrt{m_2^2 - m_2 m_3 + m_3^2}\right)} \\
&= \frac{1}{2}(1 + m_2 + m_3) + \frac{1}{2}\sqrt{1 + \frac{2}{3}(m_2 + m_3) + (m_2 - m_3)^2 - \frac{8}{3}\underbrace{\sqrt{m_2^2 - m_2 m_3 + m_3^2}}_{< (m_2 + m_3)^2}} \\
&> \frac{1}{2}(1 + m_2 + m_3) + \frac{1}{2}\sqrt{1 + \frac{2}{3}(m_2 + m_3) + (m_2 - m_3)^2 - \frac{8}{3}(m_2 + m_3)} \\
&= \frac{1}{2}(1 + m_2 + m_3) + \frac{1}{2}\sqrt{1 + (m_2 - m_3)^2 - \frac{1}{2}(m_2 + m_3)} \\
&> \frac{1}{2}\left(\underbrace{1 + m_2 + m_3 + \sqrt{1 - \frac{1}{2}(m_2 + m_3)}}_{>2}\right) > 1
\end{aligned}
$$

For the last inequality, we have used the fact $0 < m2 + m3 < 1$, that the function $g(x) = 1 + x + \sqrt{1 - x/2}$ is increasing in $[0, 1]$ and that $g(0) = 2$.

**Claim 2: $m_1^{(1)} \leq m_1^{(3)} < 0$.** From equation 25 it is clear that $m_1^{(1)} \leq m_1^{(3)}$. Thus, it is enough to prove the inequality for $m_1^{(3)}$.

$$
\begin{aligned}
& m_1^{(3)} < 0 \\
\iff & \frac{1}{2}\left(1 - m_2 - m_3 - \sqrt{(-1 + m_2 + m_3)^2 - \frac{4}{3}\left(-2m_2 - 2m_3 + 3m_2 m_3 + 2\sqrt{m_2^2 - m_2 m_3 + m_3^2}\right)}\right) < 0 \\
\iff & \left(-2m_2 - 2m_3 + 3m_2 m_3 + 2\sqrt{m_2^2 - m_2 m_3 + m_3^2}\right) < 0
\end{aligned}
\tag{26}
$$

Thus, we need to focus on inequality 26. We will differentiate various cases:

- **If $2/3 > m_3 \geq m_2$:**

$$
\left(-2m_2 - 2m_3 + 3m_2 m_3 + 2\sqrt{m_2^2 - m_2 m_3 + m_3^2}\right)
$$

$$
= m_2 \underbrace{(2 - 3m_3)}_{>0 \iff 2/3 > m3} + 2\left(\underbrace{m_3 - \sqrt{m_2^2 - m2m3 + m_3^2}}_{\geq 0 \iff m_3 \geq m_2}\right) > 0
$$

For the second term, we have used the fact that

$$
m_3 \geq \sqrt{m_2^2 - m2m3 + m_3^2} \iff m_3^2 \geq m_2^2 - m_2 m_3 + m_3^2 \iff m_2 m_3 \geq m_2^2
$$
$$
\iff m_3 \geq m_2
$$

- **If $2/3 \geq m_3 \geq m_2$:** Anologous to previous case due to the symmetry of $m_2$ and $m_3$ in equation 26
- **If $\max(m_2, m_3) \geq 2/3$:** W.l.o.g we can assume $m_2 \geq 2/3$ and $m_3 < 1/3$ due to the symmetry between $m_2$ and $m_3$. For this case, we will find the roots with respect to $m_2$ of inequality equation 26 and see that the constraints on $m_2$ do not hold. Indeed,

$$
-2m_2 - 2m_3 + 3m_2 m_3 + 2\sqrt{m_2^2 - m_2 m_3 + m_3^2} = 0 \Rightarrow
$$
$$
(-2m_2 - 2m_3 + 3m_2 m_3)^2 = 4(m_2^2 - m_2 m_3 + m_3^2)
$$

The roots $\{m_2^{(j)}\}$ of the polynomial are

$$m_2^{(1)} = \frac{2m_3}{-5 + 6m_3 + \sqrt{3}\sqrt{7 - 12m_3 + 6m_3^2}} \qquad \& \qquad m_3 \neq \frac{2 - \sqrt{2}}{3},$$

$$m_2^{(2)} = \frac{2m_3}{-5 + 6m_3 - \sqrt{3}\sqrt{7 - 12m_3 + 6m_3^2}} \qquad \& \qquad m_3 \neq \frac{2 - \sqrt{2}}{3}, \qquad (27)$$

$$m_2^{(3)} = \frac{2(-3 + 2\sqrt{2})}{3(-2 + 3\sqrt{2})} \qquad \& \qquad m_3 = \frac{2 - \sqrt{2}}{3}.$$

The denominator of the root $m_2^{(1)}$ has negative sign for $0 < m_3 < 1$, which leads to $m_1^{(2)} < 0$, contradicting the initial constraints. Trivially, $m_2^{(3)}$ is also negative.

The denominator of the root $m_2^{(2)}$ is negative for $0 < m_3 < \frac{2-\sqrt{2}}{3}$, resulting in $m_2^{(2)} < 0$. When $\frac{2-\sqrt{2}}{3} < m_3 < 1/3$, the denominator becomes positive but remains lower than $2m_3$, thus $m_2^{(2)} > 1$, which is also a contradiction.

We have ruled out all possible cases, thus we have proven the next theorem.

**Theorem I.2.** Let $\alpha = 1$. Given a set of terminals which lie in the plane, then the SPs of the optimal solution of the BCST problem will not contain SPs of degree $4$ unless these collapse with a terminal.

### I.3 FEASIBILITY DEGREE-4 STEINER POINT FOR $\alpha \in \ ]0.5, 1[$

Though we have not been able to prove analytically the infeasibility of degree-4 SPs for $\alpha \in ]0.5, 1[$, we strongly believe that the statement still holds. Figure 22, shows the surface plots of the numerator of the equality 24 (once expanded) w.r.t. $m_1$ and $m_2$ for different fixed values of $\alpha$ and $m_3$. Upon analysis, it appears that the numerator exhibits an increasing trend with respect to $\alpha$ within the interval $[0.5, 1]$. This observation leads us to hypothesize that if the equality holds for $\alpha = 0.5$ and $\alpha = 1$, it is likely to hold for intermediate values as well. However, due to the complexity of the formula, it is challenging to verify this hypothesis analytically. In section E.1, we have demonstrated that as $\alpha \to \infty$, the BCST tends to converge to a star graph centered at the geometric median. Consequently, for $\alpha > 1$, there must exist a critical value $\alpha^*$ such that $\forall \alpha > \alpha^*$ a degree-4 SP becomes feasible. It is still an open problem where the phase transition value is.

### I.4 HIGHER DEGREE BRANCHINGS INFEASIBILITY

In Lippmann et al. (2022), it was shown for the BOT problem that if degree-4 SPs are not feasible then higher degree SPs are not possible either. The same reasoning applies for the BCST, since the proof does not depend on the weighting factors. Thus for $\alpha \in [0, 0.5] \cup \{1\}$, only degree-3 SPs are feasible unless they do collapse with a terminal node. Due to the compelling evidence shown in Section I.3, we also believe this is the also the case for $\alpha \in ]0.5, 1[$. Lippmann et al. (2022) also showed that some of the results of the BOT problem obtained on the plane can be extended to other 2-dimensional manifolds. Again, this is also the case for the BCST problem. Among these properties, we emphasize the optimal angles formulae exposed in section H and the infeasibility of degree-4 SPs for appropriate $\alpha$ values. We refer to Appendix F of Lippmann et al. (2022) for more details.

## J ITERATIVELY REWEIGHTED LEAST SQUARE FOR THE GEOMETRIC OPTIMIZATION OF THE STEINER POINTS

In this section we review briefly the iteratively reweighted least square (IRLS) algorithm proposed in Smith (1992). This algorithm was initially developed for the geometric optimization of Steiner points (SPs) and later adapted in Lippmann et al. (2022) for the branched optimal transport (BOT) problem. We will show that the same algorithm can be adapted for the BCST problem, since the algorithm is agnostic to the weighting factors multiplying the distances involved in the BOT and BCST objectives, as defined in equations equation 5 and equation 2, respectively.

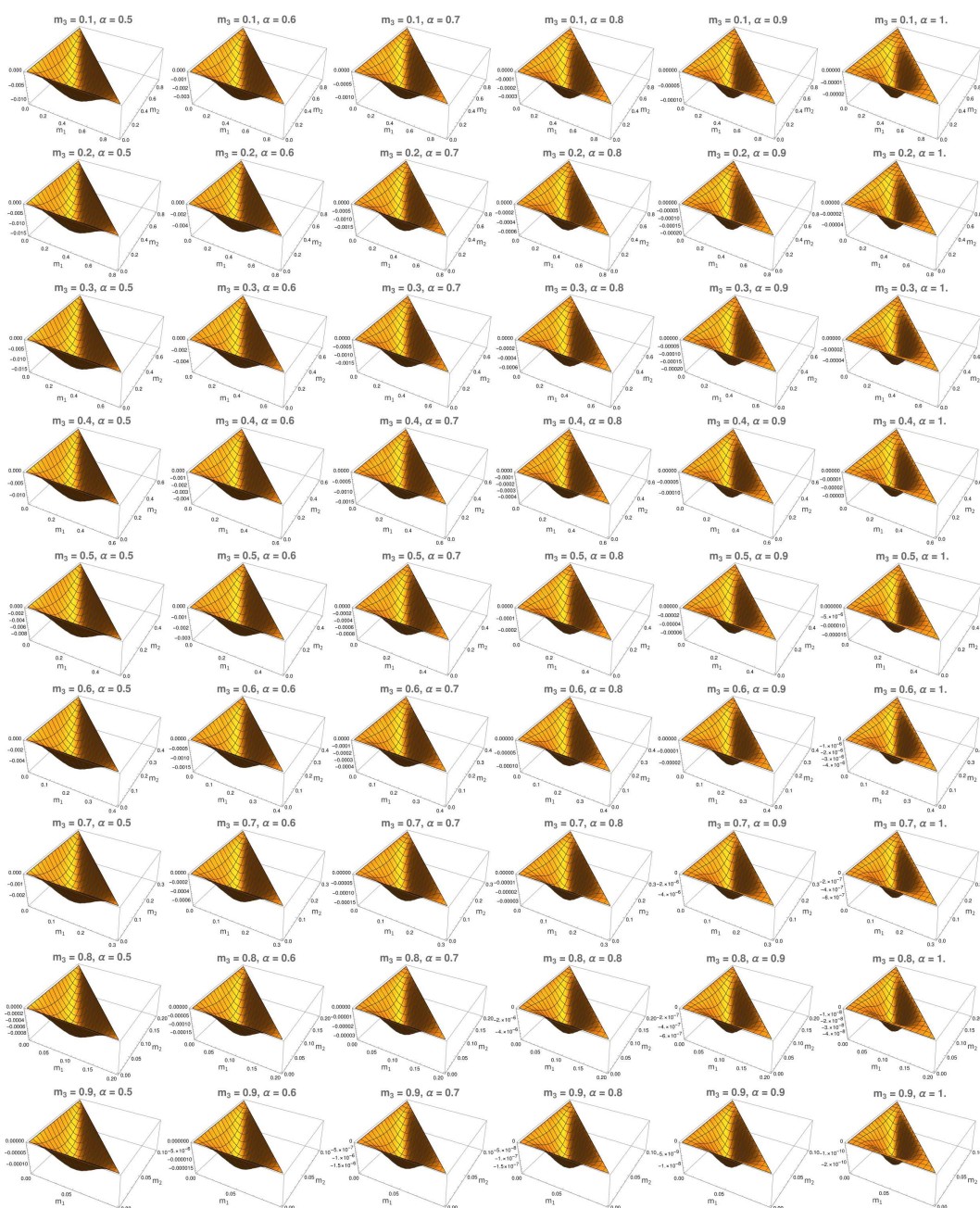

Figure 22: Surface plots are depicted, illustrating the left side of equation equation 24, as a function of $m_1$ and $m_2$, with different fixed values of $\alpha$ and $m_3$. From left to right: $m_3$ is fixed and $\alpha$ ranges over $\{0.5, 0.6, 0.7, 0.8, 0.9, 1.0\}$. From top to bottom: $\alpha$ is fixed and $m_3$ ranges over $\{0.1, 0.2, \ldots, 0.8, 0.9\}$. For $m_3$ fixed, $m_1$ and $m_2$ range over the domain defined by $\{(x, y) : 0 < x + y + m_3 < 1\}$. In all plots, the function values are negative and tend towards 0 as $m_1$, $m_2$, or $m_3$ approaches 0. We can observe that for fixed $m_1$, $m_2$ and $m_3$, the function seems to be increasing with respect to $\alpha$ (from right to left). Since we have previously demonstrated that the left side of equation 24 does not equal zero in the desired domain for $\alpha = 0.5$ and $\alpha = 1$, the plots suggest that this is also the case for $\alpha \in ]0.5, 1[$. Hence, we strongly believe that SPs of degree 4 are not feasible either for $\alpha \in ]0.5, 1[$.

Consider the following minimization problem for a fixed tree topology

$$\min_{X_B} C(X) = \min_{X_B} \sum_{(i,j) \in E} w_{ij} \|x_i - x_j\| \tag{28}$$

where $w_{ij}$ are arbitrary weights, $E$ is the set of edges of the tree, $X_B = \{x_{N+1}, \ldots, x_{2N-2}\}$ are the coordinates of the SPs, which need to be optimized, and $X$ is the set of all coordinates (terminal and SPs). Starting from arbitrary SPs coordinates, denoted as $X^{(0)}$, the algorithm iteratively solves the following linear system of equations.

$$x_i^{(k+1)} = \frac{\displaystyle\sum_{j:(i,j)\in E} w_{ij} \frac{x_j^{(k+1)}}{\|x_i^{(k)} - x_j^{(k)}\|}}{\displaystyle\sum_{j:(i,j)\in E} \frac{w_{ij}}{\|x_i^{(k)} - x_j^{(k)}\|}}, \qquad \forall N+1 \leq i \leq 2N-2. \tag{29}$$

We will show that in each iteration the cost of the objective function decreases, i.e. $C(X^{(k+1)}) < C(X^{(k)})$. As shown in Smith (1992), this implies that the $\lim_{k\to\infty} X^{(k)} = \arg\min C(X)$.

The algorithm can be considered an IRLS approach because it reinterprets the cost function as a quadratic function. Indeed, $C(X)$ can be rewritten as

$$C(X) = \sum_{(i,j) \in E} w_{ij} \|x_i - x_j\| = \sum_{(i,j) \in E} \underbrace{\frac{w_{ij}}{\|x_i - x_j\|}}_{W_{ij}(X)} \|x_i - x_j\|^2 = \sum_{(i,j) \in E} W_{ij}(X) \|x_i - x_j\|^2$$

In concrete, the solution of the linear system equation 29 minimizes the following quadratic function

$$Q^{(k)}(X) = \sum_{(i,j) \in E} W_{ij}(X^{(k)}) \|x_i - x_j\|^2.$$

That is $Q^{(k)}(X) \geq Q^{(k)}(X^{(k+1)}) \, \forall X$. Moreover, note that $C(X^{(k)}) = Q^{(k)}(X^{(k)})$. Now we can show that the cost $C$ decreases at each iteration:

$$
\begin{aligned}
C(X^{(k)}) =& Q^{(k)}(X^{(k)}) \geq Q^{(k)}(X^{(k+1)}) \\
=& \sum_{(i,j) \in E} w_{ij} \frac{\left( \left|x_i^{(k)} - x_j^{(k)}\right| + \left|x_i^{(k+1)} - x_j^{(k+1)}\right| - \left|x_i^{(k)} - x_j^{(k)}\right| \right)^2}{\left|x_i^{(k)} - x_j^{(k)}\right|} \\
=& C(X^{(k)}) + 2\left( C(X^{(k+1)}) - C(X^{(k)}) \right) \\
&+ \sum_{(i,j) \in E} w_{ij} \frac{\left( \left|x_i^{(k+1)} - x_j^{(k+1)}\right| - \left|x_i^{(k)} - x_j^{(k)}\right| \right)^2}{\left|x_i^{(k)} - x_j^{(k)}\right|} \\
\Longleftrightarrow & \\
C(X^{(k+1)}) \leq& C(X^{(k)}) - \underbrace{\sum_{(i,j) \in E} \frac{w_{ij}}{2} \frac{\left( \left|x_i^{(k+1)} - x_j^{(k+1)}\right| - \left|x_i^{(k)} - x_j^{(k)}\right| \right)^2}{\left|x_i^{(k)} - x_j^{(k)}\right|}}_{\geq 0} \\
\leq& C(X^{(k)})
\end{aligned}
\tag{30}
$$

## K  PSEUDOCODE MSTREG HEURISTIC

See Algorithm 1

---

**Algorithm 1:** mSTreg heuristic

---

**Input:** $X$, num_iterations, sampling_frequency, optimize_CST
**Output:** Tree

    /* Define initial topology as mST                                 */
1  $mST_{init}$ =minimum_spanning_tree(X)

    /* transform topology to full tree topology              */
2  $T_{\text{BCST}}$ =transform2fulltopo($mST_{init}$)

    /* compute optimal Steiner point coordinates            */
3  $SP$=compute_SP($T_{init}$)

    /* Set current best BCST cost to $\infty$                  */
4  bestcost_BCST =$\infty$

    /* If optimize_CST is True, set current best CST cost to $\infty$
      */
5  **if** optimize_CST **then**
6      bestcost_CST =$\infty$

7  **while** $it$ <num_iterations **do**
8      **if** sampling_frequency>2 **then**
         /* sample extra points from edges                */
9          Y=sample_from_edge($T_{\text{BCST}}$,$X \cup SP$,sampling_frequency)
10         $SP = SP \cup Y$

11     $mST_{X \cup SP}$=minimum_spanning_tree($X \cup SP$)

     /* transform $mST_{X \cup SP}$ to full tree topology         */
12     $T_{\text{BCST}}$ =transform2fulltopo($mST_{X \cup SP}$)

     /* compute optimal Steiner point coordinates          */
13     $SP$=compute_SP($T_{reg}$)

     /* Store T_BCST if cost is improved              */
14     **if** cost($T_{\text{BCST}}$) <bestcost_BCST **then**
15        bestcost_BCST =cost($T_{\text{BCST}}$)
16        $T_{\text{BCST}best} = T_{\text{BCST}}$

17     **if** optimize_CST **then**
        /* Derive CST topology from BCST topology         */
18        T_CST =remove_SP(T_BCST)

       /* Store T_CST if cost is improved           */
19        **if** cost($T_{\text{CST}}$) <bestcost_CST **then**
20          bestcost_CST =cost($T_{\text{CST}}$)
21          $T_{\text{CST}best} = T_{\text{CST}}$

---

## L  COMPLEXITY MSTREG HEURISTIC

We will delve into the complexity of the two steps of our heuristic.

- **Complexity geometric optimization**: We approximate the optimal SPs coordinates using the IRLS approach presented in Section J. Each iteration of the IRLS requires $\mathcal{O}(nd)$ operations, where $n$ is the number of terminals and $d$ is the data dimensionality. Within each iteration, $d$ linear systems are solved. These can be solved in linear time and in parallel. The number of

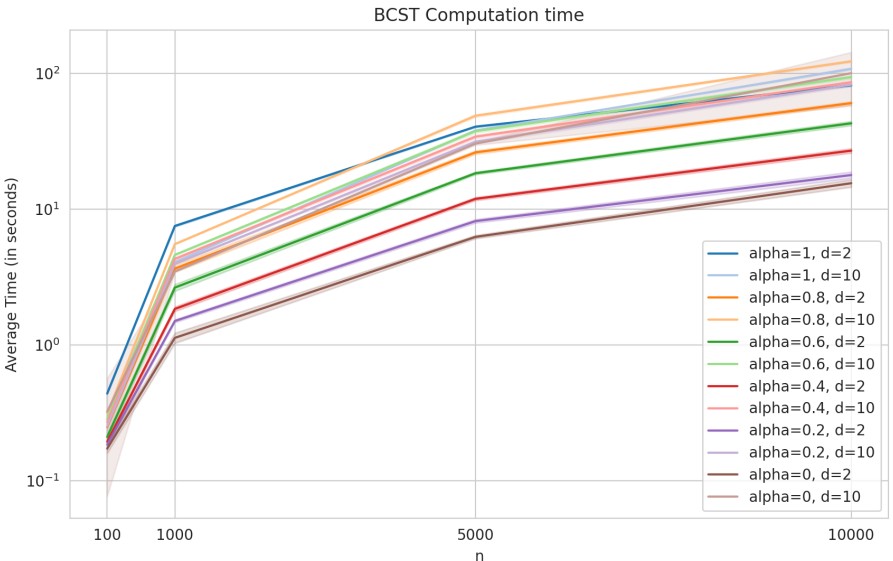

Figure 23: Computation time of 20 iterations of the mSTreg heuristic averaged over 5 distinct instances with different numbers of terminals $n$, data dimensionality $d$, and $\alpha$ values.

iterations needed for the IRLS to converge is not known a priori, however, Lippmann et al. (2022) suggest that this number could scale on average like $\mathcal{O}(\log(n))$. Consequently, each geometric optimization step takes $\mathcal{O}(\log(n)nd)$.

- **Topology optimization step**: In the topology optimization step, we compute the minimum spanning tree (mST) over the terminals and SPs. Given a graph $G = (V, E)$, Kruskal's algorithm takes $\mathcal{O}(|E| \log |V|)$ operations to compute the mST. In a complete graph, this becomes $\mathcal{O}(n^2 \log(n))$. However, in some situations, we may expedite the mST computation by computing the mST over a $k$-nearest neighbor (kNN) graph. Approximating a kNN graph with k-d trees can have a complexity of $\mathcal{O}(dn \log(n)^2)$. In this case, the number of edges in the graph would be $|E| \approx kn$. Hence, the overall mST complexity would be $\mathcal{O}(dn \log(n)^2 + kn \log(n)) \approx \mathcal{O}(dn \log(n)^2)$.

Therefore, the heuristic's per-iteration complexity is approximately $\mathcal{O}(dn \log(n) + n^2 \log(n))$ or $\mathcal{O}(dn \log(n)^2)$ if the mST is computed over a kNN graph. Throughout our experiments, a limit of 20 iterations was set, though practical convergence often demands fewer. In addition, we gauged the computational time of the heuristic by averaging its performance over 20 iterations across 5 distinct instances, varying $n$, $d$, and $\alpha$. Data was generated by sampling $n$ points from a $d$-dimensional unit cube. The performance times are presented in Figure 23. The heuristic was executed on an Intel Xeon Gold 6254 CPU @ 3.10GHz.

## M  EFFECT OF ADDITIONAL INTERMEDIATE POINTS IN THE MSTREG HEURISTIC

In Section 4.2 we have described the mSTreg heuristic as a solution approach for the BCST problem. The algorithm can be summarized in two steps: 1) Optimization of the SPs coordinates given a fixed topology; 2) topology update by computing the mST over the terminals and SPs. The motivation for the topology update is that the optimal positions of the SPs may suggest a more desirable topology, since they may be biased to move closer to other nodes than the ones to which they are connected. Thus, we hope that the new topology, defined by the mST over the SPs and the terminals, interconnects such nodes.

However, there are instances where the SPs may not be sufficiently close to each other, causing the mST to fail in recovering the desired connections. the addition of intermediate nodes along

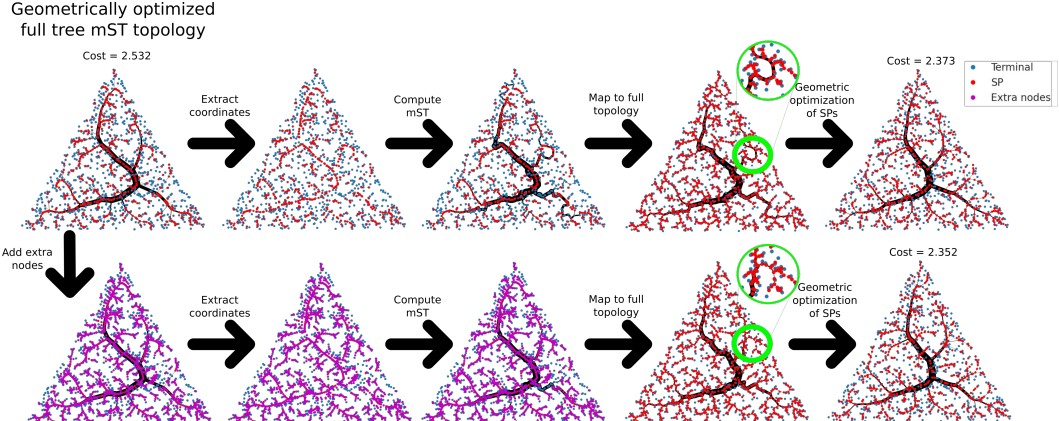

Figure 24: Effect of adding extra points per edge (visualized in violet) in the mST computation step of the mSTreg heuristic. Top left: BCST solution obtained once the mST has been mapped to a full tree topology and its Steiner point coordinates have been optimized. Top row: Next steps of the mSTreg heuristic without adding any extra point. Bottom row: Next steps of the mSTreg heuristic once an extra point has been added at the middle of each edge (shown in violet). The addition of extra points may allow the mST to more reliably follow the edges of the geometry-optimized tree from previous step. We zoom in to highlight an improvement in the topology resulting from the addition of these extra points. In this particular case, the cost obtained with the inclusion of the extra nodes is lower than the cost without them.

the edges may address this problem, allowing the mST to more reliably follow the edges of the geometry-optimized tree from the previous step. An illustrative example highlighting the benefits of this approach can be seen in Figure 24. In general, we have seen that adding between 1 and 3 nodes per edge can often yield improvements. However, the impact on the main backbone is minimal. In Algorithm 1, the number of intermediate points that are added along an edge is regulated by the `sampling\_frequency` variable.

## N    STRATEGIES TO TRANSFORM A FULL TREE TOPOLOGY INTO A CST TOPOLOGY

When using the mSTreg heuristic described in Section 4.2 for solving the CST problem without branching points, we need to map from a full tree topology to a CST topology. As shown in Section F.2, this process is ambiguous and there may be an exponential number of derivable topologies with respect to the number of terminals (see Section F.2). Hence to brute force the one which minimizes the CST cost is out of reach.

In this section, we describe some heuristic rules to transform a full tree topology into a CST topology. In order to transform a full tree topology into a CST topology, we collapse iteratively one SP at a time with one of its neighbors until there are no more SPs to collapse. The first factor to take into account is in which order the SPs are collapsed. We consider two strategies: 1) collapse the SP that is closest to a terminal ("Ordclosestterminal") or 2) collapse the SP with the closest neighbor, i.e. the one that minimizes the distance to one of each neighbors independently of if it is a terminal or a SP ("Ordclosest"). In practice we did not see any big difference, though "Ordclosest" tends to be slightly better.

The second factor to take into account is to which neighbor should an SP collapse. We again compare two different heuristics: 1) collapsing the SP to the neighbor that minimally increases the CST cost ("greedy"); 2) collapsing the SP to the closest neighbor in terms of distance. We found empirically that the greedy approach yields significantly superior results.

Lastly, we conducted tests on updating the position of the collapsed SP. When a SP, denoted as $b_1$, is collapsed with a neighbor $b_2$, then the other neighbors of $b_1$ become neighbors of $b_2$. We observed

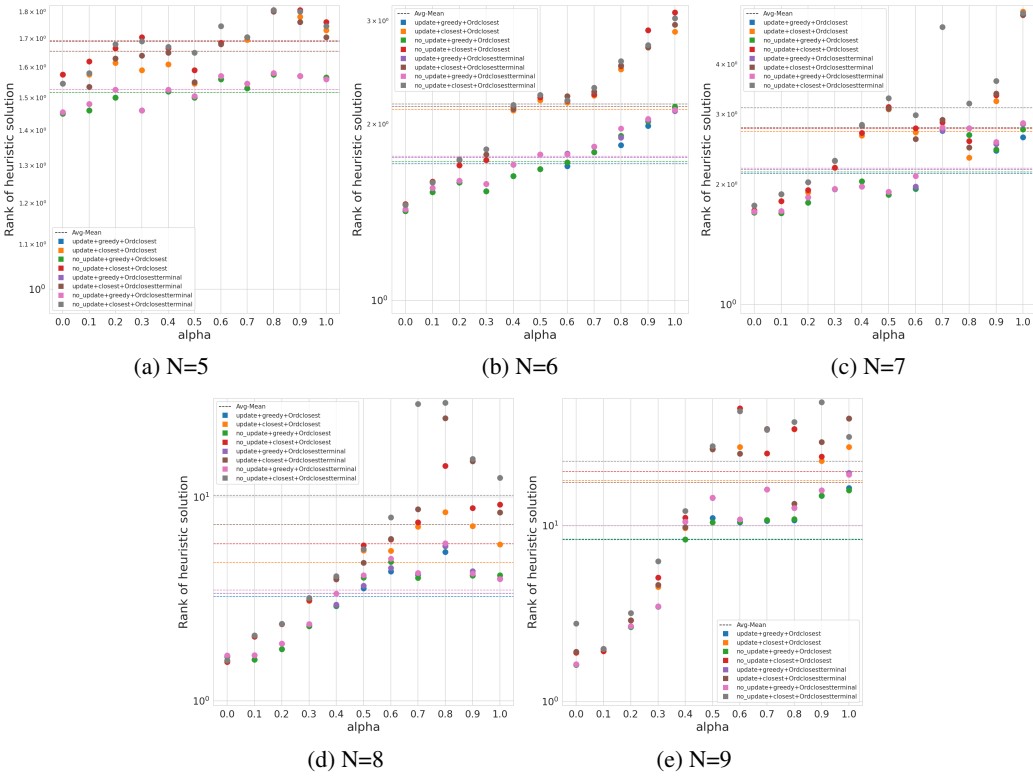

(a) N=5       (b) N=6       (c) N=7

(d) N=8       (e) N=9

Figure 25: Comparison of different collapse strategies to transform a full tree topology into a CST topology. See Section N for a short description of the strategies. We plot the average sorted position of the heuristic for different number of terminals, $N$ and for different $\alpha$ values. We observe that the strategy combinations including the "greedy" collapse approach have significantly better results. The combinations which update the position of the collapsed SP ("update") perform slightly better than the ones that do not ("no_update"). Analogously, the strategies that order the SPs based on the closeness to the neighbors ("Ordclosest") is slightly better than "Ordclosestterminal".

that updating the position of $b_2$ to the weighted geometric median of its neighbors (including those inherited from $b_1$) yielded improved results compared to not updating the coordinates of $b_2$. The coordinates of $b_2$ were only updated when $b_2$ was an SP. If $b_2$ happened to be a terminal, its position was kept fixed. To denote if a strategy updated the position or not we will use the expression "update" and "no_update" respectively.

To evaluate the effectiveness of the strategies, we conducted a series of experiments by sampling 200 problem instances for each $N$ in the set $\{5, 6, 7, 8, 9\}$, where $N$ represents the number of terminals. For each instance, we applied the mSTreg heuristic with different $\alpha$ values and utilized the aforementioned strategies to transform a full tree topology into a CST topology. Figure 25 shows the mean ranking positions obtained by the different strategies, once all feasible solutions have been sorted. The results confirm the observations that we already pointed out. For all of our experiments we used the combination that produced the best results i.e. "update"+"greedy"+"Ordclosest".

## O  FURTHER DETAILS ON THE BRUTE FORCE EXPERIMENT

In this section, we analyze the behavior of the mSTreg heuristic with respect to $\alpha$. To investigate this, we utilize the experiment described in Section 4.3, which compared the cost of the output tree generated by our heuristic with the optimal solution for different numbers of terminal nodes, denoted as N, while specifically examining the influence of $\alpha$.

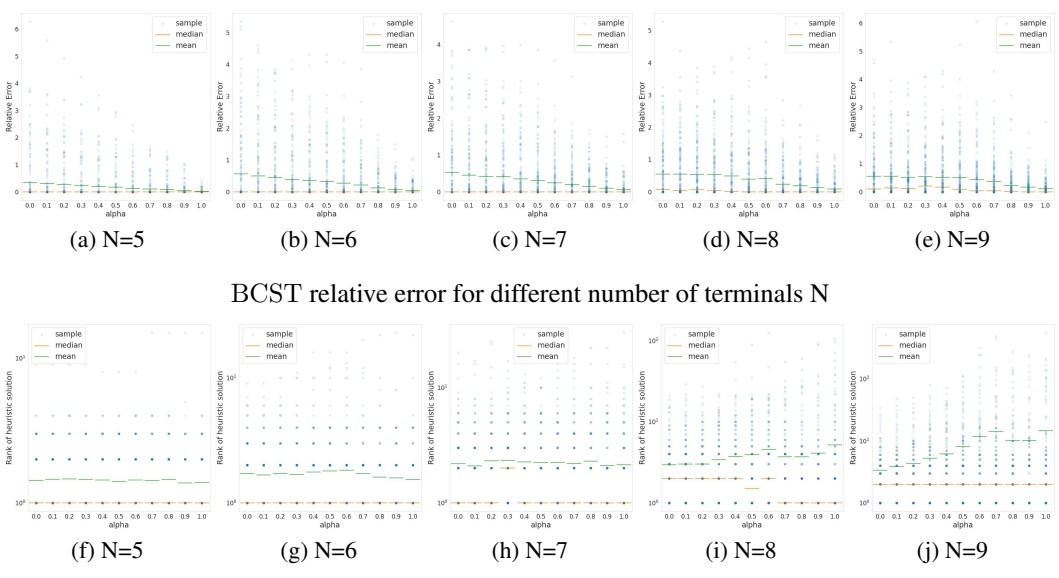

(a) N=5     (b) N=6     (c) N=7     (d) N=8     (e) N=9

BCST relative error for different number of terminals N

(f) N=5     (g) N=6     (h) N=7     (i) N=8     (j) N=9

BCST rank of heuristic for different number of terminals N

Figure 26: Relative cost errors between the $\mathrm{mSTreg}$ heuristic and BCST optimal solutions; and sorted position of the heuristic tree for different number of terminals, $N$. For each $N$ we uniformly sampled 200 different terminal configurations and we solved them for all $\alpha \in \{0.0, 0.1, \ldots, 1.0\}$. Most runs ended up close to the global optimum. There is no clear pattern with respect to the performance of the heuristic with respect to the value of $\alpha$, though for higher number of terminals, it seems that the rank of our solution gets to be worse on average.

For each $N \in \{5, 6, 7, 8, 9\}$, we sample 200 problem instances. We computed the optimal CST and BCST topologies of all problems via brute-force and with the $\mathrm{mSTreg}$ heuristic[11] for all $\alpha \in \{0, 0.1, \ldots, 0.9, 1\}$. Figures 26 and 27 show the relative error and how the heuristic solution ranks, when the costs of all topologies are sorted. When solving the BCST, though there is not a clear trend, we can observe that for higher $N$ the heuristic tends to perform worse for higher $\alpha$ values, since on average the heuristic's solution ranking is higher. When solving the CST problem this pattern can be more clearly seen.

## P   GRASP_PR APPLIED TO GENERAL CST

In this section, we delve into assessing our heuristic's performance on large instances. Utilizing the GRASP_PR algorithm, we compute the CST across various $\alpha$ values, comparing the results with our $\mathrm{mSTreg}$ heuristic on the OR library datasets. Before proceeding, we will briefly outline the GRASP_PR algorithm proposed by Sattari and Didehvar (2015).

### P.1   GRASP_PR BY SATTARI AND DIDEHVAR (2015)

GRASP_PR (Greedy Randomized Adaptive Search Procedure with Path Relinking) is a meta-heuristic for combinatorial optimization. Sattari and Didehvar (2015), adapted the general purpose GRASP_PR to solve the MRCT. The algorithm comprises three phases:

- Construction Phase (Greedy): Acknowledging that the shortest path tree with minimum cost offers a 2-approximation of the optimal MRCT, the authors designate roots as nodes whose shortest path trees incur lower costs. A random tree is then built by initially selecting a root at random.

---

[11]As described in N, we can use different strategies to transform a full tree topology into a CST topology, when solving the CST problem with the $\mathrm{mSTreg}$ heuristic. We used the one that updates the position of SPs and collapses to the closest neighbor.

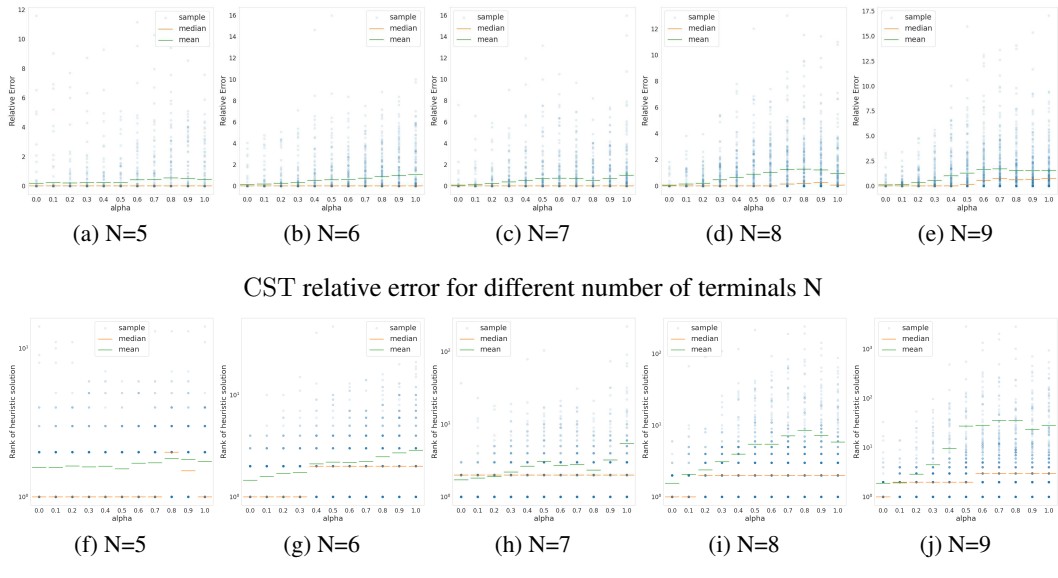

(a) N=5    (b) N=6    (c) N=7    (d) N=8    (e) N=9

CST relative error for different number of terminals N

(f) N=5    (g) N=6    (h) N=7    (i) N=8    (j) N=9

CST rank of heuristic for different number of terminals N

Figure 27: Relative cost errors between the mSTreg heuristic and optimal CST solutions; and sorted position of the heuristic tree for different number of terminals, $N$. For each $N$ we uniformly sampled 200 different terminal configurations and we solved them for all $\alpha \in \{0.0, 0.1, \ldots, 1.0\}$. Most runs ended up close to the global optimum. There is no clear pattern with respect to the performance of the heuristic with respect to the value of $\alpha$, though for higher number of terminals, it seems that the rank of our solution gets to be worse on average.

Following a process akin to Prim's algorithm, vertices are added to the tree by sampling edges, prioritizing shorter edges for higher selection probability.

- Local search: This phase iteratively refines the solution by swapping pairs of edges. Specifically, each iteration targets the pair of edges (one inside and one outside the tree) that most effectively reduces the overall cost. This iterative process continues until it converges to a local minimum, i.e. no further reduction in cost is achievable. Note that each iteration of the local search attempts to swap all pairs of valid edges to identify the one that achieves the maximum reduction in cost. Consequently, in a complete graph, the complexity of each iteration is $\mathcal{O}(n^2)$, where $n$ represents the number of nodes.

- Path Relinking Phase: Explores paths between high-quality solutions by swapping edges between good quality trees until one transforms into the other. The complexity of this step depends on the number of edges that the trees share in common.

## P.2    COMPARISON OF GRASP_PR WITH mSTreg FOR THE GENERAL CST

The previous section provided an overview of the GRASP_PR algorithm by Sattari and Didehvar (2015). It's important to note that the overall complexity is dominated by the local search, scaling quadratically with the number of nodes. In contrast, the mSTreg complexity per iteration is $\mathcal{O}(dn\log(n)^2)$ (Appendix K), rendering mSTreg more efficient.

Additionally, GRASP_PR requires an initial random guess to initiate the local search. The quality of this guess impacts the number of iterations needed for the local search to converge. We will show that initializing the local search with a solution generated by the mSTreg allows for initializations that enhance the performance of GRASP_PR. To ensure variability in the mSTreg output, we initialize it with random topologies instead of the mST. Despite this modification, as the mST initialization yielded favorable results, we opt to sample trees similar to it. To achieve this, we construct a tree by randomly sampling edges, prioritizing shorter ones. In concrete, we sample edge $(i, j)$ with probability proportional to $\exp(-||x_i - x_j||/T)$ for some temperature $T$. This sampling approach is akin to the one used in Karger's algorithm for approximating the minimum cut (Karger, 1993;

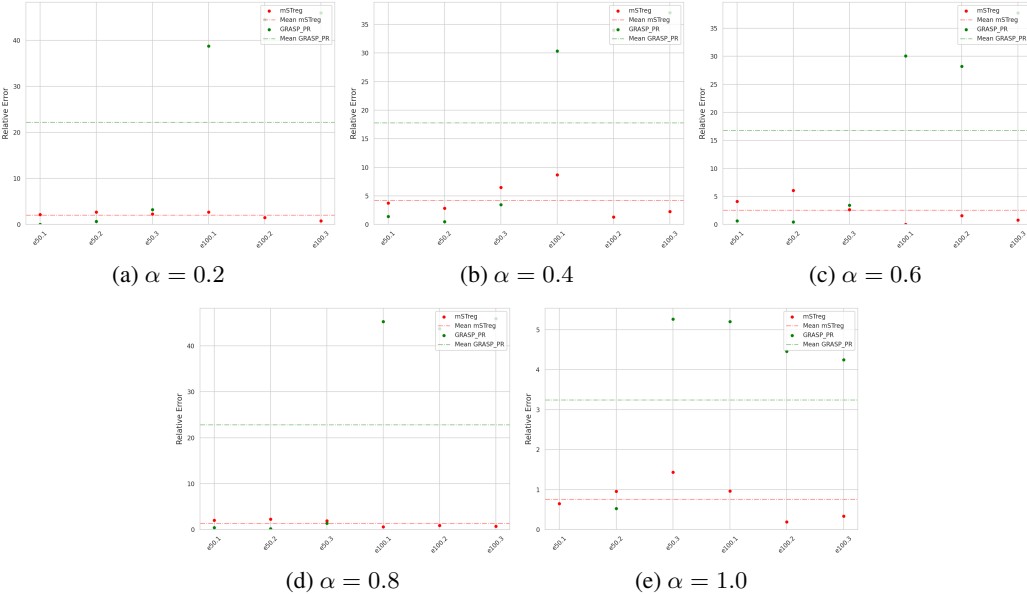

Figure 28: Relative cost error concerning the CST problem at different $\alpha$ values in the set 0.2, 0.4, 0.6, 0.8, 1 for various instances of the OR library dataset (lower is better). The comparison involves mSTreg, GRASP_PR, and the combination of GRASP_PR initialized with the mSTreg solution (referred to as GRASP_PR_mSTreg). The relative error is computed using the cost from GRASP_PR_mSTreg as a reference. The combination of the mSTreg heuristic with GRASP_PR consistently achieved the lowest cost, with all relative costs above 0, demonstrating the enhancement of GRASP_PR performance by mSTreg. To manage time constraints, a threshold of 5 minutes was imposed for methods utilizing GRASP_PR.

Jenner et al., 2021) and differs from the GRASP_PR construction phase in that it doesn't require that one end of the edges belongs to the current tree.

While our main paper relied on GRASP_PR MRCT costs from Sattari and Didehvar (2015), we now validate our claims using our implementation of GRASP_PR. Given that the GRASP_PR is a versatile algorithm, we used it to compute the CST with alternative $\alpha$ values besides 1. Acknowledging that for lower $\alpha$ values, the optimum trees will be more similar to the mST, we adapted the construction phase of GRASP_PR. Specifically, for $\alpha < 0.7$, it generates the random tree based on the Karger's sampling method mentioned above. For $\alpha \geq 0.7$, it uses the construction proposed by Sattari and Didehvar (2015). Due to the relatively slow performance of our Python implementation of GRASP_PR, we imposed a 5-minute time threshold. If exceeded, the algorithm returns the best solution at the end of the path relinking phase.

To assess performance, we conducted tests using GRASP_PR, GRASP_PR initialized with mSTreg (GRASP_PR_mSTreg), and the mSTreg algorithms on OR library datasets for problems with 50 and 100 terminals. Analogously to the plots shown in the main paper, Figure 28 displays the relative errors using GRASP_PR_mSTreg cost as a reference for different $\alpha$ values in the set $\{0.2, 0.4, 0.6, 0.8, 1\}$. Notably, the combination of the mSTreg heuristic with GRASP_PR consistently achieved the lowest cost, as all relative costs are above 0, proving that the mSTreg can enhance GRASP_PR performance.

Across 30 runs (combining 5 $\alpha$ values, 2 problem sizes, and 3 instances), GRASP_PR outperformed mSTreg in only 12 instances. Moreover, achieving a lower cost took minutes for GRASP_PR, while mSTreg run in the order of seconds. Consequently, mSTreg emerges as a favorable alternative, delivering a descent solution quickly.

## Q  SELECTION OF $\alpha$

In this section, we present our practical insights into determining the optimal value for $\alpha$. It is crucial to emphasize that the choice of $\alpha$ is task-dependent and influenced by the desired level of structure preservation. Nevertheless, we share the observations derived from our empirical experiences.

For simpler examples, as the ones illustrated in Appendix A, we have consistently found that $\alpha$ values within the range of $[0.7, 1]$ yield high stability while preserving the primary data structure. In general, an increase in $\alpha$ correlates with a heightened inclination toward a star-shaped tree, in line with the limit case discussed in Appendix E.1. This tendency becomes more pronounced in higher dimensions, where even relatively small $\alpha$ values, approximately $\alpha \lesssim 1$, lead to an almost star-shaped tree, compromising data structure preservation. Figure 12 exemplifies this effect on the Paul dataset

However, even $\alpha \gtrsim 1$ can turn the 2-dimensional toy examples presented in Appendix A into a star-graph, as shown Figures 29 and 30. Due to this early trend toward a star shape, we refrained from further exploring the case of $\alpha > 1$ since practical observations indicated an excessively star-shaped tree, even in low dimensions.

In summary, our empirical experience suggests that intermediate $\alpha$ values (around 0.5) effectively preserve the data structure while maintaining relative stability. This choice holds true for the applications highlighted in the paper. We hope that by sharing our experiences, practitioners can better select an appropriate $\alpha$ for their respective applications.

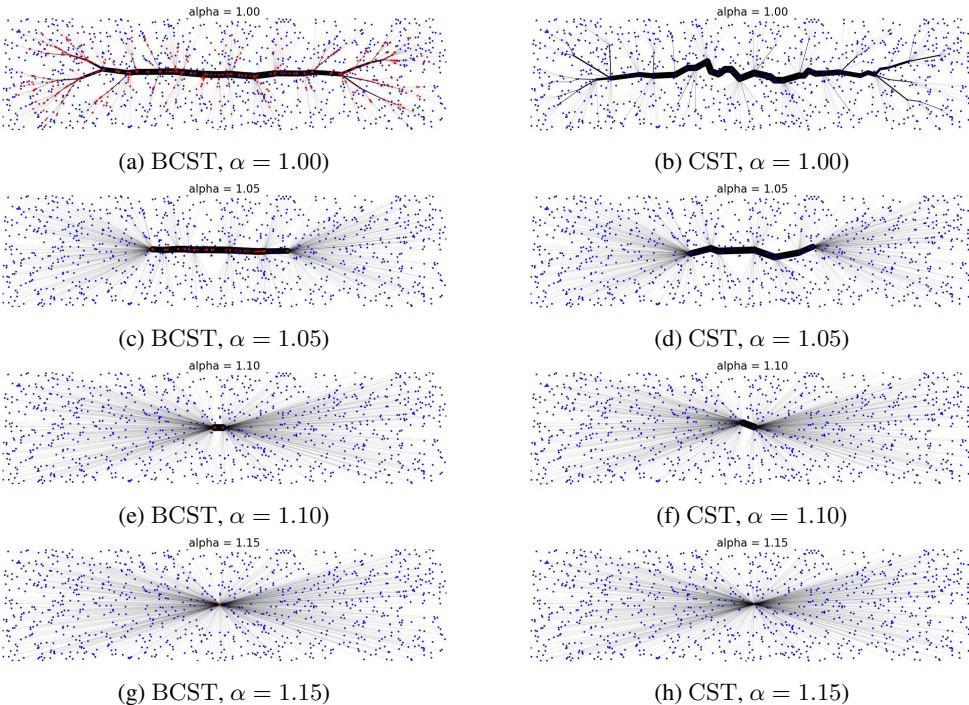

(a) BCST, $\alpha = 1.00$)    (b) CST, $\alpha = 1.00$)

(c) BCST, $\alpha = 1.05$)    (d) CST, $\alpha = 1.05$)

(e) BCST, $\alpha = 1.10$)    (f) CST, $\alpha = 1.10$)

(g) BCST, $\alpha = 1.15$)    (h) CST, $\alpha = 1.15$)

Figure 29: As $\alpha$ increases, both CST and BCST exhibit a transition towards a star graph (see Appendix E.1). This effect may manifest relatively early. Notably, for points uniformly sampled from a rectangle, the value $\alpha = 1.15$ transforms both CST and BCST into a star graph.

## R  IMPLEMENTATION DETAILS

In this section, we explain some implementation details of the mSTreg heuristic and also the parameters used for the different experiments.

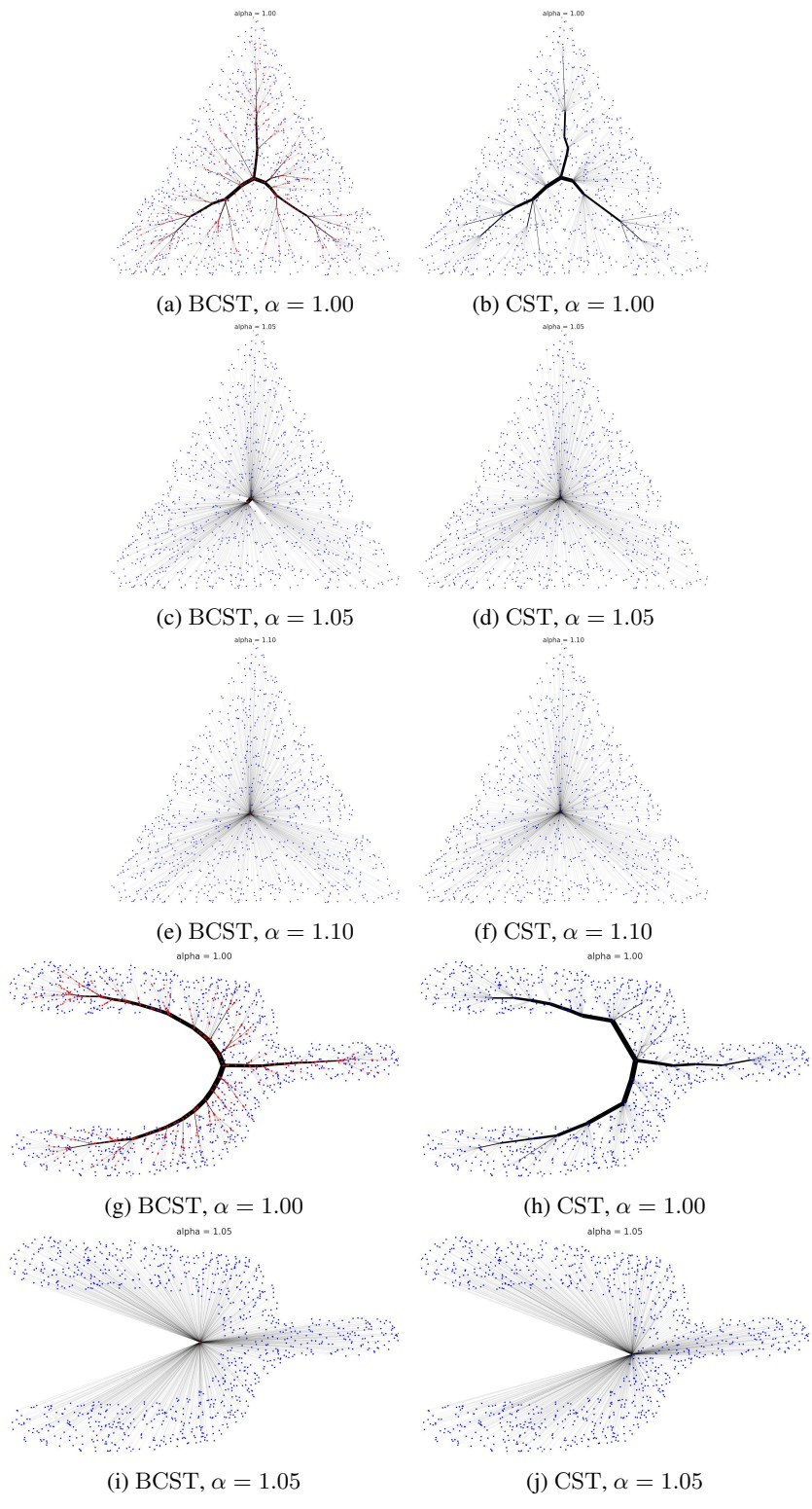

(a) BCST, $\alpha = 1.00$      (b) CST, $\alpha = 1.00$

(c) BCST, $\alpha = 1.05$      (d) CST, $\alpha = 1.05$

(e) BCST, $\alpha = 1.10$      (f) CST, $\alpha = 1.10$

(g) BCST, $\alpha = 1.00$      (h) CST, $\alpha = 1.00$

(i) BCST, $\alpha = 1.05$      (j) CST, $\alpha = 1.05$

Figure 30: As $\alpha$ increases, both CST and BCST exhibit a transition towards a star graph (see Appendix E.1). This effect may manifest relatively early. Notably, for points uniformly sampled from a triangle, the value $\alpha = 1.10$ transforms both CST and BCST into a star graph. Analogously, for a non-convex shape the star-graph realizes at $\alpha = 1.05$.

In each iteration of the mSTreg algorithm, it is necessary to compute the mST. Since we are working with a complete graph, the computational complexity of the mST computation is $O(N^2)$. To reduce this cost, we compute the mST over a k-nearest neighbor (kNN) graph, where we set the value of $k$ to $\log(N)$. Although the resulting mST over the kNN graph may not be guaranteed to be equal to the optimal mST, in practice they do not differ often. It is worth noting that the introduction of additional nodes, as described in Section M, may provide more significant benefits when using the mST computed over a kNN graph.

In Section N we have described different approaches to transform a full tree topology into a CST tree topology. Unless otherwise stated, the strategy used to collapse the SP nodes upon transforming a full tree topology into a CST tree was the one that updates the collapsed SP to the weighted geometric median, collapses greedily the SPs and determines the SP to be collapsed as the one with minimum distance to one of its neighbors ("update+greedy+Ordclosest").

In all experiments, we set the `sampling_frequency` variable of Algorithm 1 equal to 3, and we set the maximum number of iterations of the mSTreg heuristic equal to 20.

Upon publication, we will make the code publicly available.

