# OpenReview forum: "The Central Spanning Tree Problem"
_ICLR.cc/2024/Conference — Submitted to ICLR 2024_

### Official Review · Reviewer_H47r · 2023-10-30

**Soundness:** 3 good
**Presentation:** 2 fair
**Contribution:** 2 fair
**Rating:** 5
**Confidence:** 3

**Summary:**

This paper proposes new classes of spanning trees named central spanning tree (CST) and branched central spanning tree (BCST) problems. CSTs can be considered as a continuous interpolation (with parameter $\alpha$) of well-known minimum spanning trees (mST) ($\alpha=0$) and the traditional minimum routing cost trees (MRCT) ($\alpha=1$) where the sum of distances of all vertex pairs are minimized. BCST is the same as CST except that an additional branching point called Steiner point (SP) can be introduced, and it can be considered as a generalization of Steiner tree problem. This paper empirically reveals the robustness of CSTs against noise in graph data and theoretically investigates the possible topologies for BCSTs. Also, this paper proposes a heuristic algorithm for CST and BCST problem and empirically validated the optimality.

**Strengths:**

Conventionally, the mST problem and the MRCT problem have been considered completely separately, to the extent I know. This paper connects these two spanning tree problems by considering an interpolation, which itself is a theoretically interesting work. Moreover, the structure and topology of CSTs and BCSTs are well investigated.

**Weaknesses:**

The introduced problems, CST and BCST problems, are currently less meaningful in both theoretical or practical senses than the existing problems. To introduce a new problem, it is desired that at least one of the following points are realized:

(I) The newly introduced problem has a strength or offers a trade-off compared to the existing problems.
(II) The newly introduced problem introduces a new better solution/insight on the existing problems.

Regarding (I), it is empirically confirmed that the CST is robust against data perturbations, but the MRCT is also robust against data noise. So, I expected that the CST problem can be solved more easily than the MRCT problem. However, judging from the content, this is not the case: the CST is NP-hard for $\alpha>0$ (Appendix C), and the performance of the proposed heuristics (mSTreg) has no clear correspondence to the $\alpha$ value (Appendix N), meaning that the MRCT problem can be solved to the same extent as the CST/BCST problem. Therefore, I have found no significant advantage of the proposed CST/BCST problems over MRCT problem.

Regarding (II), the heuristics for solving CST/MRCT problems is proposed (mSTreg) and it is applied to the existing problems, Steiner tree problem and MRCT problem. However, for Steiner tree problem, only instances such that the optimal solution is known are used as benchmarks. For MRCT problem, the proposed mSTreg falls short of the existing algorithm GRASP_PR. Although the authors claim that GRASP_PR is time-consuming because of heavy local search and it can be improved by using mSTreg for preprocessing, it is not verified through experiments. Therefore, currently the proposed heuristics, mSTreg, beats the existing methods only for the newly proposed CST/BCST problems.

Minor comment:
Regarding the final point, the computational time is also a key factor for the performance analysis of heuristic algorithms. I have read the per-iteration complexity and the time-measuring experiment in Appendix K, but I think at least a statement for computational time should be described on the main part of the problem. To the extent I understand, the per-iteration complexity is fairly small for graphs with 100 vertices and thus the proposed method runs much faster for the graphs used in the evaluation (Section 4.3). If the consumed time for the evaluation in Section 4.3 can be obtained and can be compared with the existing methods, they should be clearly stated.

**Questions:**

1. Please see "Minor comment" on "Weaknesses".
2. If my understanding on "Minor comment" is correct and the proposed heuristics runs much faster, is there any improvement on the solution quality output by mSTreg by modifying the stopping criterion or increasing the number of iterations?

---

> ### Author Response · Authors · 2023-11-15
>
> We would like to thank the reviewer for pointing out that our work is theoretically interesting. Next, we will address the concerns that were not covered in the main response.
>
> - **Advantages of CST over MRCT**: The consistent performance of mSTreg across diverse $\alpha$ values stands out as a significant strength, aligning with our goal of creating a useful heuristic. Therefore, the point that the CST is as difficult to solve as the MRCT is secondary, as the CST has still valuable advantages over the MRCT. While MRCT's higher star-shape tendency enhances stability, it risks information loss, especially in higher dimensions, as depicted in Figure 12. The parameterized nature of $\alpha$ in CST mitigates this, allowing for a tailored balance between robustness and data fidelity. This flexibility is particularly crucial in higher dimensions, where the CST's interpolation surpasses the limitations of MRCT. These observations underscore the superiority of the more general CST and its branched version, emphasizing their real-world applicability, specifically in higher dimensions.
>
> - **Increased number of iterations**: Adding more iterations to mSTreg may lower the tree cost. While our heuristic converges quickly, its inherent randomness permits additional exploration. As explained in Section 3, the same CST "type" topology can yield multiple full tree topologies. Consequently, when transforming the mST into a full tree topology, variations can emerge. Despite convergence, the randomness in this step may explore new topologies, further reducing the overall cost.

---

> > ### Comment · Reviewer_H47r · 2023-11-23
> >
> > Thank you for detailed responses of authors.
> >
> > Regarding the first point (I), I certainly overlooked the advantage of CST/BCST with moderate $\alpha$ values that their information loss is smaller than MRCT. Thank you for pointing out this, and I will raise the score. However, it is still questionable that the CST/BCST problems can easily be solved than the MRCT problem; I think this is important since, judging from the main contents of this manuscript (without Appendix), the main focus of this manuscript is on the robustness of the spanning tree.
> >
> > Regarding the second point (II), thank you for conducting additional experiments that verify the performance of GRASP_PR initialized with the proposed mSTreg heuristics. I look the experimental results somewhat convincing, but it is still questionable that the long-time running results of GRASP_PR will be. In the additional experiments, the time limit is 5 minutes was set for the running time of GRASP_PR, and the results of GRASP_PR was worse than those of single mSTreg for many instances. Thus, I cannot imagine the results of GRASP_PR and GRASP_PR_mSTreg when more running time is given.

---

> ### Author Response · Authors · 2023-11-23
>
> We express our gratitude to the reviewer for their feedback.
>
> Concerning the level of difficulty in solving the more general CST/BCST compared to the MRCT, it is important to clarify that we are not asserting that our proposal is inherently easier to solve. Rather, our argument centers on the observation that the parameterized CST/BCST exhibits more appealing properties than the MRCT. As a result, opting for the CST/BCST might be justified even if both are equally challenging to solve.
>
> Regarding the 5-minute time limit, we aimed to demonstrate that mSTreg can outperform the MRCT in terms of efficiency, while yielding satisfactory solutions. Notably, mSTreg, executed with just 20 iterations, produced a solution within seconds. In contrast, GRASP_PR did not consistently enhance this solution within the specified 5-minute timeframe. Extending the runtime of GRASP_PR would likely result in further improvements, as it would explore the solution space more extensively. It is important to emphasize that GRASP_PR, initialized with mSTreg, also adhered to the 5-minute time limit, surpassing the other two alternatives. This demonstrates the clear benefit of the combined approach and highlights how initializing with mSTreg can enhance GRASP_PR.

---

### Official Review · Reviewer_2UJx · 2023-10-31

**Soundness:** 2 fair
**Presentation:** 1 poor
**Contribution:** 2 fair
**Rating:** 3
**Confidence:** 4

**Summary:**

The problem under consideration in the present paper is the estimation of the underlying tree structure of a point cloud. To this end, a new model, called central spanning tree, is developed and investigated. It is defined as the solution of an optimization problem involving combinatorial and geometric aspects.

Parameterized by $\alpha$ between $0$ and $1$, central spanning tree interpolates models from the literature, in particular the minimum spanning tree ($\alpha=0$). The main interest of this new model is that its robustness to noise increases with $\alpha$, making it more robust than minimum spanning tree.

A heuristic is proposed to solve the optimization problem of interest, which turns out to be NP-hard. Numerical simulations support the claims of the paper on robustness of the model and accuracy of the estimation algorithm.

**Strengths:**

This article proposes a new method for extracting the tree structure of a point cloud, with emphasis on robustness to noise, which is a major challenge in real applications.

The paper focuses on the theoretical and numerical properties of the model, on both synthetic and real data, and proposes a heuristic for solving the optimization problem at stake.

**Weaknesses:**

The main weakness of this article is its presentation. From my point of view, it is not in the right format for a conference like ICLR. The main paper refers extensively to the appendices, which are 30 pages long. Some parts of the main paper are difficult to read without calling up the long version, particularly the optimization algorithm (section 4). Mention should also be made of the link with the minimum concave cost network flow problem, which helps to show that the problem is NP-hard: some elements are given in the main body, but the link can only be understood by carefully reading the appendix.

The main advantage of the proposed model is its robustness to noise. However, it is only compared with the minimum spanning tree (and the Steiner tree in the case where nodes can be added), which is known not to be robust to noise. However, competing algorithms are mentioned: some are even used as benchmarks to measure the accuracy of the heuristic. The paper's main motivation (robustness) should be more thoroughly documented.

On the same aspect, it would be relevant for the authors to cite and compare with the following papers:
> Kasperski and Zielinski (TCS, 2010). On the approximability of robust spanning tree problems.

> Bezrukov et al. (LNCS, 1996). On central spanning trees of a graph.

The parameter $\alpha$ serves as a trade-off between noise robustness and data fidelity, but how do the authors choose it? For example, why was $\alpha=0.8$ chosen for the simulations in Figure 2? The model may lose some of its interest if practitioners do not know how to select $\alpha$ in their applications.

**Questions:**

In addition to the comments above, here are a few questions for the authors.

If I understand the definition of Steiner points correctly (page 3), the data are only used for the terminal nodes of the tree, while all the internal nodes are added (they are not part of the data) and their position is to be optimized. Is this really the case? If it is, I have two questions:

(i) First, is optimization problem (2) well formulated (compared to equation (1))?

(ii) Then, is this model really relevant to extract the tree structure of a point cloud?

I think this point should be better discussed throughout the paper.

Why do the authors exclude $\alpha>1$ from their investigations? Did they study the complexity of the optimization problem (which is proved to be NP-hard only for $0<\alpha\leq 1$) in this case?

Why do the authors do not consider the central spanning tree (but only the branched version) in the application to 3D plant skeletonization?

page 2: Is it obvious that the model turns into the minimum routing cost tree for $\alpha=1$?

page 6: Why does the optimization problem (2) is not everywhere differentiable?

I can understand the theoretical interest behind the study of branching angles and degree of Steiner points, but what is the link with the rest of the paper? Could this be of any use in addressing the paper's main questions?

Typos:

page 4: The ability

page 6: convergence

parametrize (page 9)/parameterize (page 1)

---

> ### Author Response · Authors · 2023-11-15
> **Part 1**
>
> We would like to express our gratitude to the reviewer for their conscientious review.
>
> - **Presentation**: While we acknowledge the length of our Appendix, we believe it serves the purpose of providing comprehensive details for readers interested in a deeper understanding of our work. Our intention is to strike a balance between clarity in the main paper and the availability of detailed information in the Appendix. We contend that our manuscript does its best at conveying the most relevant information in the main part. Regarding the comprehension of our heuristic, we would value specific feedback on the aspects that may have been challenging, enabling us to make targeted improvements in the clarity of the section.
>
> - **Robustness discussion**: To the best of our knowledge, we are the first ones to delve into the robustness of the geometrical structure of a spanning tree in response to data perturbations, such as noise. We thank the reviewer for providing additional references. However, it's crucial to highlight that the robustness treated by these papers differs from ours. Bezrukov et al.'s paper, although related by name, focuses on computing a Frechet mean among all spanning trees of a graph using the symmetric difference of edge sets as a metric. The paper by Kasperski and Zielinski studies the mST with uncertain edge costs. Thus, they aim to find a tree whose cost is robust under different realizations of the edge costs. Thus, both papers neglect the geometrical stability of the tree. Consequently, our comparison only included the mST and the MRCT, as these are natural limits of our CST proposal. Note that, though the MRCT already existed previously in the literature, its robustness, as explained here, had not been mentioned before. Moreover, as discussed in  "Advantages of CST over MRCT"  (see response to H47r), the MRCT's robustness may cause an information loss in contrast to the parameterized CST. In light of this, we contend that our study effectively addresses the specific robustness considerations within our scope. Simultaneously, we acknowledge that our related work did not mention alternative robustness criteria, and we will amend this in the final version.
>
> - **Role of Steiner points (SPs)**: Indeed, your understanding of the SPs is right. Regarding the formulation of the optimization problem in (2), we posit that it is appropriately formulated. While (1) only minimizes over the set of spanning trees, (2) jointly minimizes both the trees and the SPs' positions, what from our perspective does not pose a conflict. (2) is equivalent to minimizing the SPs' positions given a fixed topology and subsequently selecting the topology that achieves the minimum cost. If there are specific aspects of (2) that are unclear or concerning, we would appreciate additional details to address them comprehensively.
>
>   Regarding the relevance of the Steiner points to model the structure, we acknowledge your valid point. Although SPs are not inherent to the original data, there are scenarios, where abstracting the data requires points that are not necessarily part of the raw data. For example, in many single-cell trajectory inference methods, it is common to define cluster centers (which are not part of the actual data) to describe the trajectory. Also in skeletonization the SPs become relevant as argued below in response "CST and Skeleton". We will emphasize these aspects in the final manuscript to provide a more comprehensive understanding of the applicability of the SPs.
>
> - **CST and Skeleton**: We exclusively utilized the BCST for skeletonization, primarily because it is often desired that the skeleton goes through the "middle" of the volume. The SPs in the BCST offer the flexibility to be situated in the volume's center, ensuring that the BCST skeleton traverses this central region. In contrast, the CST lacks this flexibility and would, at best, zigzag over surface points. Hence, the CST was not considered for this particular application.
>
> - **Case $\alpha>1$**: Apart from the limit case detailed in Appendix D.1, we refrained from further exploration due to practical observations indicating an excessive star-shaped tree, even in low dimensions. For instance, the toy dataset in Figure 2 became a star tree for $\alpha=1.15$. Regarding NP-hardness, it remains unproven in this regime as the strictly concavity argument applicable for $0<\alpha\leq1$ loses its validity. We hypothesize that in practice the problem might be generally easier to solve for higher $\alpha$ values, especially considering that, for the BCST problem, the SPs tend to collapse into a single point.

---

> > ### Author Response · Authors · 2023-11-15
> > **Part 2**
> >
> > - **Equivalence MRCT and CST with $\alpha=1$**: As argued in the paper, the term $m_{ij}(1-m_{ij})$ is proportional to the betweenness centrality of the edge $(i, j)$. This centrality quantifies the number of shortest paths that traverse the given edge. Thus, the multiplication of the length of each edge by its frequency in a shortest path is a rearrangement of the sum over all shortest path costs, i.e. the MRCT cost. Formally,
> >
> > $\displaystyle\sum_{i,j\in V\times V}d_T(i,j)=\sum_{i,j\in V\times V}\sum_{(u,v)\in P_{ij}} ||x_u-x_v||=\sum_{(u,v)\in T} \sum_{i,j\in V\times V}I_{(u,v)\in P_{ij}}||x_u-x_v||\propto \sum_{(u,v)\in T}m_{uv}(1-m_{uv})||x_u-x_v||$
> >
> >   where $P_{ij}$ is the path from $i$ to $j$ in tree $T$ and $I_{(u,v)\in P_{ij}}$ is an indicator set.
> >
> > - **Differentiability of (2)**: Equation (2) contains the term $||x_i-x_j||$. The Euclidean norm is not differentiable at 0, thus when two Steiner points $x_i$ and $x_j$ share coordinates, the function is not differentiable.
> >
> >
> > - **Branching angles study**: The study of the angles has practical applications on our algorithms. Having an analytical expression of the branching angles may serve as a criterion of the convergence of the iterative reweighted least square approach used to compute the SPs positions. Moreover, the infeasability of degree four SPs can ease the detection of non-optimal solutions and identify areas where the tree can be further improved, potentially leading to better results.

---

> > > ### Comment · Reviewer_2UJx · 2023-11-21
> > > **Response to authors**
> > >
> > > I would like to thank the authors for their response, which is quite convincing on a number of points I raised in my report.
> > >
> > > Nevertheless, I remain sceptical about the format of the article.
> > >
> > > As for Steiner's points, it is now clearer to me. Identifying applications where their use is relevant or not seems to be a difficult problem. I thank the authors for pointing out these interesting examples where extracting the data structure may require introducing new points.
> > >
> > > If I am not mistaken, the authors have not answered the question about the selection of the  $\alpha$-parameter, in general and in the numerical simulations discussed in the paper.

---

> > > > ### Author Response · Authors · 2023-11-21
> > > > **Choice fof $\alpha$**
> > > >
> > > > Thank you for your feedback. Regarding the point concerning the "selection of $\alpha$" we would like to include the response that we unintentionally omitted in the previous message. This oversight occurred due to the splitting of responses into two parts.
> > > >
> > > > - **Selection of $\alpha$**: The choice of the specific $\alpha$ depends on the task and the desired level of structure preservation. We share here our empirical experience. In Figure 2, the choice of $\alpha=0.8$ is not rooted in a specific rationale but rather in the visual inspection of tree stability. For the toy examples, we found that $\alpha$ values in the range of $[0.7,1]$ provide high stability while preserving the primary data structure. The trees in Figure 2 illustrate the case for $\alpha \in \{0, 0.8\}$, but a more comprehensive view is presented in Appendix A, where various toy datasets showcase the impact of different $\alpha$ values. Generally, increasing $\alpha$ results in a higher tendency towards a star-shaped tree (see Appendix D.1). We have observed that this effect is more pronounced in higher dimensions, where $\alpha\lesssim 1$ already results in almost a star tree compromising the data structure preservation. Figure 12 exemplifies this effect on the Paul dataset. Yet, even a slightly higher $\alpha$ (e.g., 1.15) can turn the tree of the rectangle toy dataset into a star. Our empirical experience suggests that intermediate $\alpha$ values (~0.5) effectively preserve the data structure while maintaining relative stability. This choice holds true for the applications shown in the paper.

---

> > > > > ### Comment · Reviewer_2UJx · 2023-11-21
> > > > >
> > > > > Thanks for the additional comment on the choice of $\alpha$, which I think would be worth including in the paper.

---

> > > > > > ### Author Response · Authors · 2023-11-21
> > > > > >
> > > > > > Following your advice, we have incorporated a dedicated section in the appendix presenting our empirical insights on the choice of $\alpha$. Additionally, a comprehensive table of contents for the appendix has been included, offering a concise summary of each section. We hope that this addition enhances the accessibility and comprehensiveness of the appendix contents.

---

### Official Review · Reviewer_BdLt · 2023-10-31

**Soundness:** 3 good
**Presentation:** 3 good
**Contribution:** 2 fair
**Rating:** 5
**Confidence:** 4

**Summary:**

In this paper, the authors propose a parameterized family of spanning tree, which encompasses the minimum spanning, the Steiner and minimum routing cost trees as limiting case. Moreover, they propose a heuristic algorithm to compute the corresponding trees. At last, they conduct experiments to evaluate the effectiveness of the proposed algorithms.

**Strengths:**

S1. New family of spanning trees are proposed.
S2. New algorithms are proposed to compute the corresponding trees.
S2. Experiments are conducted to evaluate the proposed algorithms.

**Weaknesses:**

W1. The motivation of the proposed tree is not strong.
W2. Only one real example regarding the superiority of the proposed model is provided.
W3. The efficiency of the proposed algorithms are not well evaluated.

**Questions:**

Q1. The motivation to proposed a new tree seem weak. The paper argue that small perturbations of the original data set  often lead to drastic changes in the existing spanning trees, but the scenarios that the robustness really matters are not provided. The necessity of a new family of trees seems unconvincing.

Q2. In Section 2, a single cell gene expression measurement example is provided. However, it seems that CST also cannot detect the trajectory bifurcation. Moreover, as mST cannot detect the trajectory bifurcation even on the original data, the example seems hard to support the superiority of proposed model regarding the robustness.


Q3. The efficiency of the proposed algorithms are not well evaluated in the experiments. As mST can be computed efficiently in practice, it is unfair to propose a more complex tree model without considering the efficiency. It is better if the efficiency of the proposed algorithm could be compared with existing algorithms.


Q4. Is there any theoretical guarantee regarding the returned results of the heuristic algorithms proposed in section 4?

---

> ### Author Response · Authors · 2023-11-15
>
> We would like to thank the reviewer for the thorough examination of our work. Next, we will address the concerns not covered in the main response.
>
> - **(Q2) CST on single cell data**: While CST may not fully capture the bifurcation, it does exhibit a discernible split, indicated by two wide edges. The fact that the mST is not able to detect the bifurcation, does not diminish the value of the robustness argument as the mST undergoes drastic changes post-perturbation, in contrast to the CST. This highlights the CST's superiority in terms of robustness, even if not optimal. Nevertheless, we acknowledge that BCST might be more suitable for such applications due to better trajectory representation and preservation. We remark, that this is not the only example we show; the Appendix illustrates another instance leading to the same conclusions.
>
> - **(Q4) Guarantees of the heuristic**: Currently, our heuristic lacks theoretical guarantees regarding the cost of the solutions it provides. However, through empirical evaluation, we have demonstrated that our heuristic consistently produces competitive solutions. It is worth noting that many algorithms employed for this class of problems rely on genetic algorithms or local search strategies, neither of which inherently provide any formal guarantee. While some algorithms may offer worst-case scenario bounds, they often struggle to scale effectively for large instance problems [1,2].
>
> [1] Ravelo et al., "A PTAS for the metric case of the optimum weighted source–destination communication spanning tree problem", (2019).
>
> [2] Wu et al., "Approximation algorithms for some optimum communication spanning tree problems", (2000).

---

> > ### Comment · Reviewer_BdLt · 2023-11-23
> >
> > Thanks for your response, I keep my score

---

### Official Review · Reviewer_JfkS · 2023-11-01

**Soundness:** 3 good
**Presentation:** 2 fair
**Contribution:** 3 good
**Rating:** 6
**Confidence:** 3

**Summary:**

The submission at hand proposes a novel notion of spanning trees - called central spanning trees - that interpolate between concisely representing the structure of shortest connections between a set of data points in a Euclidean space and being robust to slight pertubations of these points. This comes in the form of a parameterized framework where one weights the cost of an edge in the spanning tree according to its centrality in the spanning tree. The parameter (alpha) determines how strong this weighting is.
The motivating aspect that central spanning trees are more robust than spanning trees is supported by an empirical analysis.
For alpha=0 the problem coincides with the minimum spanning tree and for alpha=1 the problem coincides with the minimum routing cost tree.
For all alpha > 0 determining the minimum cost of a central spanning tree is NP-hard and hence the manuscript also includes a heuristic for it.
All these contributions are also extended to the setting in which non-terminal points are allowed in the tree (leading to analogues of the Steiner tree problem).

**Strengths:**

I believe this contribution is a decent topical fit for ICLR. The suggested notion of central spanning trees is very natural and interesting, also from a theoretic standpoint.
Overall the presentation of the results is good (with the exception of me not completely understanding the setup of the experiments in 2.1).

**Weaknesses:**

In my opinion, the benefit of considering CST over MRST is not sufficiently motivated in the current manuscript.
Also, the quality of the heuristic seems to only be considered for alpha=1 on large instances. I can understand why, but it is still a weakness of the evaluation of the heuristic.

A minor error on the level of a typo is that on page 2, 0 should not be included in the range of alpha for which concavity is claimed.

**Questions:**

Have you also thought about extending central spanning trees to more general graphs (not complete Euclidean metric ones)?

What exactly is the setup for Section 2.1? How many random instances do you start with and then perturb?

---

> ### Author Response · Authors · 2023-11-15
>
> We would like to thank the reviewer for pointing out that our work is very natural and interesting.
>
> - **Section 2.1**: This section aimed to empirically assess the stability of the (B)CST against noise. We started with an initial sample, denoted as $P_0$, consisting of n=1000 points uniformly drawn from a rectangle. To investigate the impact of noise, we generated five perturbed samples $P_i$ by introducing zero-centered Gaussian noise, defined as $P_i = P_0 + \Sigma_i$, where $i$ ranges from 1 to 5, and $\Sigma_i \in \mathbb{R}^{1000\times2}$ is a matrix with entries sampled independently from $\mathcal{N}(0,0.005)$. Using the Frobenius norm $||\cdot||_F$, we assessed robustness by comparing the shortest path distance matrices between trees computed by (B)CST at different $\alpha$ values over original and perturbed samples. In other words, for each $i\in {1,...,5}$, we computed the norm $||D_0^{\alpha}-D_i^{\alpha}||_F$, where $D_i^{\alpha}$ is the shortest path distance matrix of the (B)CST applied to $P_i$ at a certain $\alpha$ value. Figure 2a) presents the average Frobenius norm across $\alpha$ along with the average total length of the trees. The plot illustrates a decrease in the Frobenius norm, indicating increased similarity and greater robustness to perturbations with rising $\alpha$, albeit at the cost of longer edges. We hope this clarifies the experiment's setup.
> - **(B)CST for more general graphs**: The CST easily generalizes to non-complete Euclidean graphs by considering their spanning trees. Analogously, the BCST could be generalized by considering full topologies derivable from the non-complete graph spanning trees. Yet, our heuristic faces challenges in this extension, as the mST step might yield a spanning tree, which is not a subgraph of the original graph. Ensuring the topology's validity in each iteration is non-trivial and therefore is left for future work. Additionally, our work briefly explores extending the (B)CST to Riemannian manifolds, addressing in Appendix H.4 the degree four Steiner points infeasibility on 2-dimensional manifolds. We recognize the heuristic's potential extension to arbitrary Riemannian manifolds pending efficient computation of Steiner points. Notably, CST, unlike BCST, can be defined over weighted graphs withtout requiring embedding in a metric space. However, our heuristic is irrelevant in this context, as the relation with BCST cannot be exploited.

---

> > ### Comment · Reviewer_JfkS · 2023-11-22
> >
> > Thank you for your response. My question about extending central spanning trees to non-metric or non-clique graphs is answered to my satisfaction, and so is that about the setup in the toy data set. I am somewhat surprised that only one starting point set was chosen which I think could have been increased to a higher number for a more reliable interpretation of the experiment's outcome but as not too much focus is put on this experiment and more real-data pointsets are included, I do not see this as a major concern but something that could be improved in the future.

---

> > > ### Author Response · Authors · 2023-11-22
> > >
> > > Thank you for your feedback. We appreciate your positive comments on our responses.
> > >
> > > Regarding your mention of only one "starting point set being chosen," we want to clarify that we are uncertain about the specific reference. If you are referring to the selection of a single original dataset, $P_0$, generated by points sampled uniformly from a rectangle, we would like to state that we conducted the same experiment on the 2-dimensional datasets depicted in Appendix A.  The corresponding plots, akin to the one illustrated in Figure 2a, exhibited similar trends for these alternative datasets. We omitted them from the paper as we deemed them redundant. Instead, we chose to exclusively display the trees of these datasets at various $\alpha$ values.

---

> > > > ### Comment · Reviewer_JfkS · 2023-11-22
> > > >
> > > > There only being one set $P_0$ is indeed what I was referring to. What I am saying is essentially that I think it would be better to have repeated the same experiment many times. Of course the figures you make to show the influence of alpha make sense to depict for one specific $P_0$ but in terms of experimental setup one would want to exclude that the initial point set was just a lucky pick. The same applies to each distribution in Appendix A.

---

> > > > > ### Author Response · Authors · 2023-11-22
> > > > >
> > > > > For the final manuscript, we will replicate the experiment multiple times to validate the consistency and reliability of our findings.

---

> > > > > > ### Author Response · Authors · 2023-11-23
> > > > > > **Repetition experiment with more original samples**
> > > > > >
> > > > > > We replicated the experiment with three distinct original samples, each comprising 1000 points sampled from a rectangle. Each set was perturbed five times using zero-centered Gaussian noise. Figure 2a in the main paper has been updated accordingly, demonstrating a consistent pattern in comparison to the experiment with a single original sample set.

---

### Author Response · Authors · 2023-11-15
**Motivation**

We appreciate the time invested in carefully reading our paper and the very helpful and detailed comments.

- **Motivation**: We would like to further emphasize the significance of the proposed CST and BCST methods. The (B)CST, parameterized by $\alpha$, strikes a unique balance, ensuring both fine detail capture and global geometric stability. While robustness in spanning trees has been explored previously (see "Robustness discussion" response to 2UJx) , to our knowledge, we are the pioneers approaching this type of geometric stability, a valuable contribution in its own right, even in absence of application. However, Section 2 highlights potential applications where the (B)CST holds relevance, and we now expand on this list.
    - *Biological and Medical Machine Learning*: In applications like bioinformatics and medical diagnostics, a robust tree ensures consistent relationships between biological entities across varying conditions, vital for tasks such as single-cell transcriptomics and phylogenetic tree construction.
    - *3D Modeling and Reconstruction in Computer Graphics*: In 3D modeling, especially in reconstructing shapes from point clouds, a robust tree can assist in creating accurate and stable representations. This is particularly relevant in noisy environments or when dealing with incomplete data sets.
    - *Operations research and network design*: Designing networks to fortify terminal communication while favoring shorter edges and maintaining robustness against node location uncertainty can be achieved using the (B)CST. The extent of the desired effect is governed by the parameter \alpha.

    In summary, a stable spanning tree, such as the (B)CST, can find widespread utility across domains, from biomedical research to computational geometry among others. Our paper aims to raise awareness of the (B)CST problem, showcasing its potential across domains and encouraging its application in diverse fields.

---

> ### Author Response · Authors · 2023-11-15
> **Efficiency and comparison of the proposed mSTreg heuristic**
>
> - **mSTreg vs GRASP_PR**: In Section 4, we evaluated the effectiveness of the proposed mSTreg heuristic on solving large problems of the (B)CST particular problem instances with $\alpha=0$ and $\alpha=1$, namely the Steiner and Minimum Routing Cost Tree (MRCT) problems. It's important to note that our goal wasn't to achieve state-of-the-art performance, but to asses how competent can the mSTreg be. The Steiner problem, being an established problem, has specialized algorithms that may outperform our heuristic. Therefore, we only compared against the optimum cost. For the MRCT problem, lacking access to the optimum solution, we compared against other methods, including the GRASP_PR by Sattari et al. While the GRASP_PR method yielded a lower cost than our heuristic, we argued that mSTreg is more efficient and can enhance GRASP_PR's performance. In the following, we will clarify our claims:
>     - **Efficiency**: GRASP_PR follows a two-phase approach. In the first one, it constructs a random solution and enhances it through local search. Each iteration of the local search replaces an edge of the tree, by the one that most decreases the cost. The second phase, explores paths between high-quality solutions to enhance the search. The overall complexity is dominated by the local search. A single iteration of it has a complexity of $\mathcal{O}(n^2)$, with $n$ being the number of nodes. In contrast, the mSTreg complexity per iteration is $\mathcal{O}(dn\log(n)^2$) (see Appendix K), where $d$ is the dimensionality of the data. Thus, mSTreg is more efficient.
>     - **mSTreg can enhance GRASP_PR performance**: The effectiveness of GRASP_PR is influenced by the quality of its initial random guess, as it impacts the number of iterations needed for the the local search to converge. Next, we show that initializing the local search with a solution generated by mSTreg allows for initializations that enhance the performance of GRASP\_PR.
>     In our original report, we relied on the reported costs from the original GRASP_PR's paper, as we couldn't find an existing implementation. Now, we implemented it to validate our claims. Given that GRASP_PR is a versatile algorithm, we have used it also to compute the CST with other $\alpha$ values in addition to the $\alpha=1$ case reported in the main paper. Due to the slowness of the GRASP_PR algorithm we set a 5-minute time threshold. If exceeded, the algorithm returns the current best solution at the end of its second phase. We conducted tests using GRASP_PR, GRASP_PR initialized with mSTreg (GRASP_PR_mSTreg), and the mSTreg algorithms on the OR library datasets for problems with 50 and 100 nodes. The table below displays the average cost over three instances for different $\alpha$ values in the set {0.2, 0.4, 0.6, 0.8, 1}. Notably, the combination of the mSTreg heuristic with GRASP_PR consistently achieved the lowest cost. Across 30 runs (combining 5 $\alpha$ values, 2 problem sizes, and 3 instances), GRASP_PR outperformed mSTreg in only 12 instances. Moreover, achieving a lower cost took minutes for GRASP_PR, while mSTreg run in the order of seconds. Consequently, mSTreg emerges as a favorable alternative, delivering a decent solution quickly.
>
> |Problem size and $\alpha$| |GRASP_PR|mSTreg|GRASP_PR_mSTreg|
> |------------------------------------|-|----------------|----------|-------------------------------|
> |n=50, $\alpha$=0.2|| 14.3| 14.5| **14.2**|
> |n=50, $\alpha$=0.4|| 40.7| 41.8| **40.0**|
> |n=50, $\alpha$=0.6|| 115.9| 119.1| **114.2**|
> |n=50, $\alpha$=0.8|| 331.9| 336.5| **329.7**|
> |n=50, $\alpha$=1.0|| 942.7| 935.6| **926.3**|
> |n=100, $\alpha$=0.2|| 33.2| 23.6| **23.2**|
> |n=100, $\alpha$=0.4|| 106.4| 82.7| **79.5**|
> |n=100, $\alpha$=0.6|| 370.7| 282.8| **280.6**|
> |n=100, $\alpha$=0.8|| 1442.5| 1002.4| **994.9**|
> |n=100, $\alpha$=1.0|| 3651.9| 3507.6| **3490.0**|

---

> > ### Author Response · Authors · 2023-11-20
> > **Submission update**
> >
> > We have updated our manuscript, aiming to further motivate our proposal and introducing related work on alternative robustness approaches. Additionally, we have included a section in the Appendix (Appendix P) detailing the experiments conducted during the rebuttal, comparing GRASP_PR with and without mSTreg initialization for various $\alpha$ values.

---

### Meta-Review · Area_Chair_WKaQ · 2023-12-04

**Metareview:**

The paper introduces a new spanning tree problem.  The idea of the problem is to construct a definition that is robust to nosie in the data where current trees like MST and Steiner tree fail.

Challenges with the problem formulation are that (1) it is not clear what the tradeoffs are between this formalization and the current state-of-the-art formalizations, (2) the efficiency of computing the problem and (3) there is a body of work on robust spanning trees (MST, Steiner tree) that were not discussed.

This paper needs to theoretically establish the formal guarantees of the proposed problem over prior work.  The paper needs to better compare with past work on robust trees.

**Justification For Why Not Higher Score:**

I would like to have seen more comparison with prior work before publication.

**Justification For Why Not Lower Score:**

N/A

---

### Decision · Program_Chairs · 2024-01-16

Reject